# Divide, Harmonize, Then Conquer It: Shooting Multi-Commodity Flow Problems with Multi-modal Language Models[*]

**Xinyu Yuan**[△]**, Yan Qiao**[▽]**, Zonghui Wang**[△,✉] **& Wenzhi Chen**[△,✉]
[△]Zhejiang University   [▽]Hefei University of Technology   [✉]Co-corresponding authors
`yxy5315@gmail.com, qiaoyan@hfut.edu.cn, {zhwang,chenwz}@zju.edu.cn`

## Abstract

The multi-commodity flow (MCF) problem is a fundamental topic in network flow and combinatorial optimization, with broad applications in transportation, communication, and logistics, etc. Nowadays, the rapid expansion of allocation systems has posed challenges for existing optimization engines in balancing optimality and tractability. In this paper, we present PRAM, the *first* ML-based method that leverages the reasoning power of multimodal language models (MLMs) for addressing the trade-off dilemma—a great need of service providers. As part of our proposal, PRAM (i) quickly computes high-quality allocations by *dividing* the original problem into local subproblems, which are then resolved by an MLM-powered "agent", and (ii) ensures global consistency by *harmonizing* these subproblems via a multi-agent reinforcement learning algorithm. Theoretically, we show that PRAM, which learns to perform gradient descent in context, provably converges to the optimum within the family of MCF problems. Empirically, on real-world datasets and public topologies, PRAM achieves performance comparable to, and in some cases even surpassing, linear programming solvers (very close to the optimal solution), and substantially lower runtimes (1∼2 orders of magnitude faster). Moreover, PRAM exhibits strong robustness (<10% performance degradation under link failures or flow bursts), demonstrating MLM's generalization ability to unforeseen events. PRAM is objective-agnostic and seamlessly integrates with mainstream allocation systems, providing a practical and scalable solution for future networks.

## 1 Introduction

Suppose we want to send multiple commodities from their sources to their respective destinations along the arcs of an underlying network, with the objectives of achieving low link utilization, high throughput, and fairness among commodities. This scenario results in what we call a multi-commodity flow (MCF) problem (Assad, 1978). Its importance has been underscored by wide-ranging applications in transportation, communication, logistics, energy, and cloud computing (Schrijver, 2002; Wang & Wang, 1999; Balcik et al., 2014; Blaauwbroek et al., 2015; Chang et al., 2010). Over the past decades, optimization-based algorithms built on linear programming (LP) played an important role in solving such problems, which is guaranteed for computing near-optimal solutions (Khachiyan, 1980; Karmarkar, 1984; Chen & Ye, 2024). However, the conventional wisdom in the community is that solving these LPs becomes time-consuming as the solution space scales up (Cohen et al., 2021). This bottleneck is further magnified in modern systems, which often comprise thousands of nodes or/and links, and must accommodate millions of commodities with unpredictable demands (Applegate & Cohen, 2003; Wang et al., 2006). More recently, the advancements of machine learning (ML) have prompted extensive research into ML-based solutions (Valadarsky et al., 2017; Bernárdez et al., 2021). While faster, these methods are still limited by the problem scale (§ 3.1), as the number of variables to be optimized remains unchanged. Additionally, they inherit the common shortcomings of ML models, e.g., sensitivity to unseen environments and painstaking hyperparameter sweeping.

In response to the central barrier of scalability, we take a different perspective. Rather than monolithic optimization, our primary idea is to handle the growing network size by decomposing the original

---

[*]Our implementation is publicly available at ⛓ `https://github.com/Y-debug-sys/PRAM`.

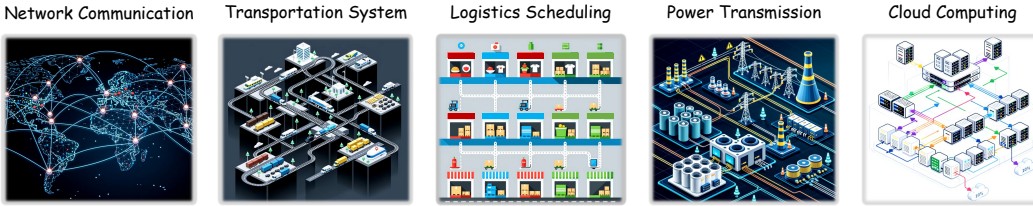

Figure 1: **Various real-world examples of multi-commodity flow.** From left to right: wide-area network traffic engineering, urban mobility management, delivery route optimization, regional power dispatch, and tenant-aware flow control, all of which involve a very large solution space today.

problem into smaller subproblems through the partition of topology and demands (Abuzaid et al., 2021; Cohen et al., 2021), and given historical data, directly solving them in parallel through deep neural networks (DNNs). Building on this idea, we propose **PRAM**: **P**artitioned **R**esource **A**llocation with **M**ultimodal Language Models (MLMs). First, we divide the MCF problem by commodity flow sources, then allocate flows for each subset individually using a shared "agent" model. We here choose MLMs as the agent backbone for two major reasons: ① they exhibit emergent abilities that were not explicitly programmed into them during pre-training on massive data, such as mathematical problem reasoning and generalization to unforeseen conditions (Zhang et al., 2024; Wu et al., 2024a); and ② they mitigate the costs of retraining and handcrafting specialized DNNs for complex inputs, by processing topology and demand data as images and text tokens, respectively. Second, we present a novel multi-agent reinforcement learning (MARL) algorithm for fine-tuning the agent model using counterfactual policy gradients (Foerster et al., 2018), with lightweight communication enabled by trainable low-rank matrices and prefix context. This allows each logical agent to exchange and estimate its individual contribution to the team's success. Further, through case studies, we show that since the MCF objective typically satisfies suitable convexity/concavity properties, once adapted, PRAM provably attains the optimum by internally simulating the gradient descent (GD) procedure. Hence, the effectiveness of PRAM rests on a solid theoretical foundation. To this end, the paper makes three key contributions which can be summarized as follows:

• In a remarkable departure from conventional centralized consensus, we propose PRAM (§ 3) to divide and optimize MCF problems in a distributed manner, thereby providing efficiency at scale.

• PRAM is the first end-to-end MCF solver built on off-the-shelf MLMs (§ 3.2), with no redundant manual design overhead. It employs a lightweight and interpretable adaptation framework (§ 3.3) that counters partition effects and fully taps the model's reasoning for different flow allocation tasks.

• A suite of theoretical results (§ 4) demonstrates that, once adapted, PRAM can internally approximate near-optimal solutions, offering performance guarantees largely absent in prior works.

Our evaluation (§ 5) encompasses topologies of different scales, heterogeneous demand distributions, and multiple MCF objectives. In comparison with a variety of solutions, including RL, LP and heuristics, the main results show that: ① PRAM *achieves near-optimal performance.* It consistently outperforms previous RL-based allocation schemes, with an average performance gap of less than 8% from the optimal solution. ② PRAM *accelerates flow allocation at scale.* It speeds $10\times$ to $100\times$ faster than solving LPs on large-scale topologies. ③ PRAM *generalizes well to new environment.* It exhibits strong robustness to demand distributions, flow dynamics, and network failures. Through ablation studies and visualizations, we delve deeper into the factors contributing to PRAM's effectiveness.

In conclusion (§ 6), we regard our exploration of MLM-powered MCF optimization as an initial yet groundbreaking step, and discuss the limitations of PRAM that we hope future research can address.

## 2 PROBLEM FORMULATION

We formulate the multi-commodity flow (MCF) problem on a directed graph or topology $\mathcal{G}(\mathcal{V}, \mathcal{E}, c)$, where $\mathcal{V}$ is the set of vertices or nodes, $\mathcal{E}$ the set of directed edges or links, and $c : \mathcal{E} \to \mathbb{R}^+$ specifies link capacities. Each source-destination pair $s, t$ is associated with a *predefined* set of candidate paths $P_{s,t}$ and *historical* demand $\mathcal{D}_{s,t}$. A configuration $\pi$ distributes each commodity flow using candidate path weights $r_p$. The goal is to periodically determine the optimal $\pi$ under different objectives, e.g., minimizing maximum link utilization for resilience, maximizing total flow for profit, or concurrent flow for fairness (see Appendix B.1 for objective details). This problem is known to be

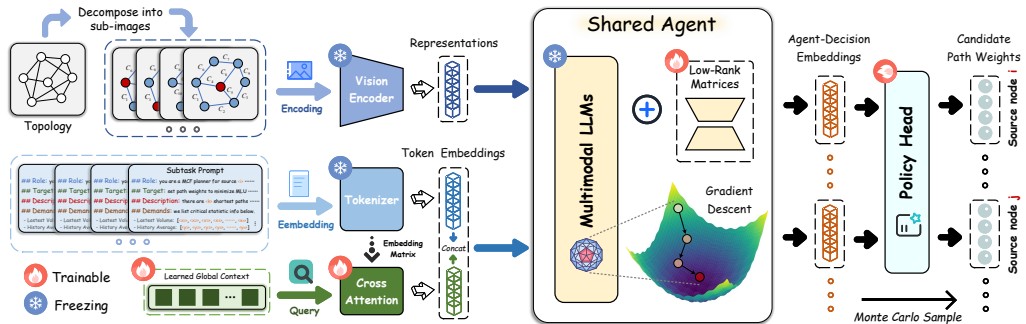

Figure 3: **Overview of PRAM.** It consists of three core components: *partition* module to divide task into smaller sub-tasks, *MLM-based agent* module to to generate sub-task-specific answers, and *adaptation* module to efficiently learn global knowledge for MCF optimization.

NP-complete (Even et al., 1975) and finds many important applications across various domains (e.g., communication, transportation, logistics and so on) as listed in Figure 1.

# 3 PRAM: PARTITIONED RESOURCE ALLOCATION WITH MLMS

Complex MCF problems are often intractable as a whole but can be decomposed into subproblems defined over subsets of commodities and links. We propose to leverage MLMs for solving these subproblems in parallel, exploiting their mathematical reasoning capacity to yield high-quality allocations. We call this framework Partitioned Resource Allocation with MLMs (or PRAM for short). In the rest of this section, we describe the motivations, benefits, and designs of PRAM.

## 3.1 MOTIVATIONS BEHIND PRAM

**Achilles' Heel of LP-based Methods**    The MCF problem is typically formulated as a mathematical program, and in theory, an optimal solution can always be obtained (or infeasibility certified) using LPs. In practice, however, solving these programs can be prohibitively expensive: the worst-case complexity of LP with $d$ variables is about $\mathcal{O}(d^{2.3729})$ (Lee & Sidford, 2015; Cohen et al., 2021; Narayanan et al., 2021), and modern systems may involve millions of variables (e.g., at least one per source–destination pair), resulting in extremely long runtimes (e.g., in hours). Even worse, the optimal solution assumes full knowledge of all current commodities, yet real-time measurement at scale is often impractical. As a result, inputs are typically derived from historical flows or their predictions. But given the uncertain and unpredictable nature of data such as network traffic (Wang et al., 2006; Perry et al., 2023), the final performance is often highly suboptimal.

**Where ML-based Methods Fall Short**    Recent studies have sought to bypass the iterative optimization process using machine learning, where deep neural networks (DNNs) directly determine routing decisions. Although promising, existing ML-based methods fall short in several key limitations, as we list in the following. ① *High engineering costs.* Their success is heavily dependent on retraining, particularly for RL, which relies on fragile optimization (Henderson et al., 2018), and engineering DNN models, such as graph neural networks (GNNs) (Wu et al., 2020; Bernárdez et al., 2021), for the target scenarios, which, however, can be labor-intensive due to the complex structures. ② *Poor generalization.* DNNs trained on specific environments may not perform well on unseen ones (Yuan et al., 2026). ③ *Curse of dimensionality.* Since representing the flow allocation needs $\mathcal{O}\left(|\mathcal{V}|^2\right)$ path weights, the scaling challenge remains. For instance, in a topology of $1,000$ nodes with $4$ candidate paths, the output alone would require $4$ million dimensions.

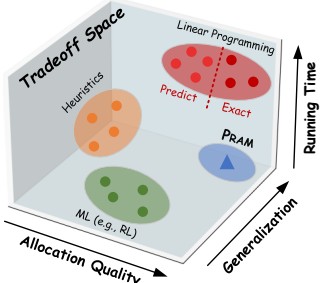

Figure 2: Tradeoff space.

Faced with the above dilemma, an intuitive way of accelerating allocation and reducing complexity is to decompose MCF optimization into sub-tasks, applying powerful solver(s) simultaneously in each subproblem and merging their results at the end. For example, dividing commodities and demands evenly among $k$ sub-problems will reduce the number of variables in each sub-problem by $k^2$ (Cohen

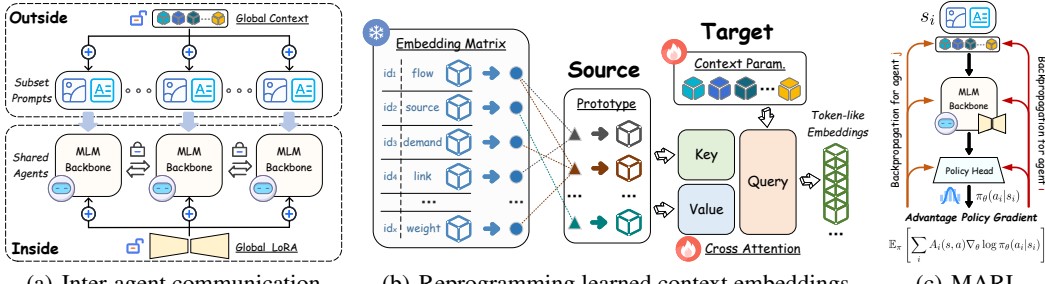

(a) Inter-agent communication      (b) Reprogramming learned context embeddings      (c) MARL

Figure 4: **Illustration of PRAM's adaptation framework.** In (a), PRAM builds inter-agent communication through LoRA and reprogramming context using cross attention in (b). In (c), policy gradient flow is computed from each agent's difference reward to estimate the contribution of its actions to the team's global reward.

et al., 2021). Holding the premise, we argue that with adequate generalization and expressiveness, without frequent retraining, ML can also have the potential to be a principled alternative that enables near-optimal and rapid decision making for general MCF problems. This is exemplified by the recent popular pretrained (multimodal) large language models, opening new opportunities that allow PRAM to be effective, high-accuracy, but faster as seen in Figure 2.

## 3.2 MULTIMODAL PROBLEM PARTITION

To translate our motivation into practice, the first step of PRAM is a partition procedure. We aim to maximize parallelism without excessive decomposition, since the latter can impede convergence and even increase solving time at scale. Revisiting the earlier example, decomposing at the granularity of individual commodity flows may generate millions of subproblems, requiring numerous batching rounds even on modern GPUs. So taking a step back, PRAM focuses on solutions at a node level by treating commodities from the same source as one subset, reducing the model complexity from $\mathcal{O}(|\mathcal{V}|^2)$ to generally tractable $\mathcal{O}(|\mathcal{V}|)$. No need for any specially designed modules such as GNNs or RNNs, thanks to the off-the-shelf interface of MLMs, our design can be realized in a simple and convenient manner. As illustrated on the left of Figure 3, the procedure amounts to instructing the MLM with subgraphs (in image form) together with the commodity information (in text form). Concretely, we plot all routing links from the source node to every other node once to obtain a visual representation, which is then fed to MLMs through a vision encoder (e.g., CLIP (Radford et al., 2021)). At the same time, we introduce subtask-aware prompting, in which each recent demand in the commodity subset is explicitly paired with the subtask's key descriptions (e.g., source node specification) and statistics (e.g., historical rolling average demand). A prompt example is in Appendix C.1. The MLM "agent" is then expected to jointly process and reason over both modalities to yield high-quality MCF configurations—an issue we address next with our adaptation framework.

## 3.3 LIGHTWEIGHT MULTI-AGENT ADAPTATION

In the implementation of PRAM, we promote its efficiency by sharing the MLM backbone among the subproblem agents. These logical agents can still behave differently because they receive different observations and thus evolve different hidden states. However, since the model is not inherently specialized for this task, the agents are unable to perceive each other's presence. This necessitates the introduction of additional (communication) parameters and raises the challenge of fine-tuning them.

**Communication** Our scheme of inter-agent communication is in Figure 4(a). The global (trainable) parameters are added in two ways: (i) Inside the model, we introduce low-rank matrices for MLM's attention weights to approximate the changes needed in the backbone parameters using the LoRA technique (Hu et al., 2022). (ii) Outside the model, inspired by LM's in-context learning (ICL) mechanics (Brown et al., 2020; Lin et al., 2024): an ability to flexibly adjust their prediction based on additional data given in context (i.e., in the input sequence itself), we create a set number of learnable "global context" embeddings as prefix of input prompts using the reprogramming technique (Li et al., 2023; Jin et al., 2024a). As illustrated in Figure 4(b), we connect the context to a frozen input embedding matrix of the tokenizer associated with the MLM, and we perform alignment with a multi-head cross attention layer. Specifically, we model text prototypes as keys and values by linearly aggregating original token embeddings, while the context parameters act as queries to extract

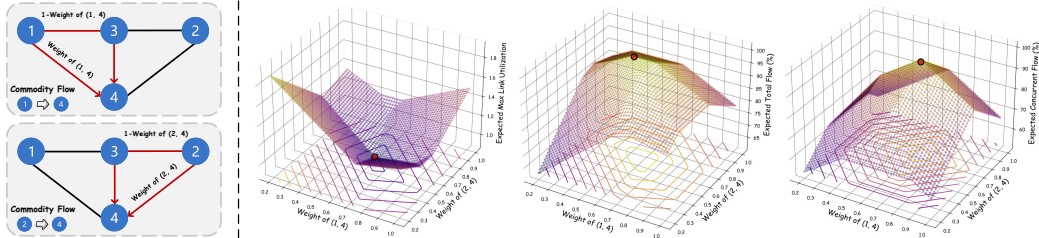

Figure 5: **A case study on the multi-commodity flow problem.** *(left)* In a simple network, both ① and ② need to transmit commodities to ④, each having two candidate paths. *(right)* The (expected) objective functions are plotted against the path weights, with their optima indicated by red dots. The curves exhibit clear convexity/concavity, lending theoretical support to the soundness of PRAM.

global information or guidance the agent requires and understands. Upon packing the prompt and context embeddings, as shown in Figure 3, we feed them through the MLM backbone to produce the agent-decision embeddings. As our evaluation results in § 5.4 show, these designs impose limited parameter overhead; in other words, PRAM is lightweight overall.

**Adaptation** While independent adaptation of different agents via gradients from the global objective (loss) is straightforward, the lack of information exchange during fine-tuning limits the ability to learn coordinated strategies or to assess an individual agent's contribution to the system. To overcome this, we adopt multi-agent reinforcement learning (MARL) algorithms (Busoniu et al., 2008; Kraemer & Banerjee, 2016; Foerster et al., 2018; Lyu et al., 2021; 2023). In particular, PRAM leverages *counterfactual reasoning* (Foerster et al., 2018; Xu et al., 2023), asking: *how would the global objective change if only one subset's flows were reallocated while others remained fixed?* The reward difference defines the single agent's advantage relative to a counterfactual baseline, thereby quantifying its contribution to the joint outcome.

To be more specific, when (history) demands arrive, PRAM provides the token embeddings of each subset as the local state $s_i$ to the corresponding RL agent $i$, which then outputs an action $a_i$ from a policy head network, i.e., a vector of path weights for its managed demands. All agents share a policy $\pi_\theta$ parameterized by $\theta$, trained via policy gradient. We then denote by $s$ the central state composed of all local states $s_i$, and by $a$ the joint action of all agents' actions $a_i$. After all decisions are made, a reward $R(s, a)$ is obtained. To compute the advantage $A_i(s, a)$, PRAM exploits the one-step nature of the MCF: since actions (flow allocations) do not affect future states (demands or/and paths), the expected return reduces to the immediate reward $R(s, a)$. Moreover, if agent $i$ changes its action to $a_i'$ while others keep theirs fixed, the new joint action $(a_{-i}, a_i')$ can be directly evaluated by simulating its effect on the objective, i.e., we compute the MCF objective obtained if the new joint action were to be used. Putting it together, PRAM computes the advantage for agent $i$ as

$$A_i(s, a) = R(s, a) - \sum_{a_i'} \pi_\theta(a_i'|s_i) R\left(s, (a_{-i}, a_i')\right), \quad g = \mathbb{E}_\pi\left[\sum_i A_i(s, a)\nabla_\theta \log \pi_\theta(a_i|s_i)\right],$$

where $g$ is the policy gradient, and the counterfactual baseline is approximated via Monte Carlo sampling, i.e., drawing random actions $a_i' \sim \pi_\theta(\cdot|s_i)$. As shown in Figure 4(c), we fine-tune PRAM in an end-to-end manner, with $\theta$ denoting all tunable parameters, so that gradients are backpropagated seamlessly from the policy head network. We include more details of the algorithm in Appendix C.3.

## 4    UNDERSTANDING PRAM: CASE STUDY AND THEORY

**Through the Looking Glass of MCF** To delve into the MCF problem, we conduct an illustrative case study in Figure 5. This study is based on a toy topology consisting of 4 nodes and 5 links, each with a capacity of 1. Now, suppose nodes ① and ② need to send commodity flows to ④. With equal probability, their demands is either $\frac{3}{2}, \frac{6}{7}$ or $\frac{7}{6}, \frac{3}{2}$. The candidate paths from ① to ④ are $(1, 4)$ and $(1, 3, 4)$, while those from node ② to ④ are $(2, 4)$ and $(2, 3, 4)$. In this case, the configuration $\pi$ is fully determined once the weights on links $(1, 4)$ and $(2, 4)$ are specified. In Figure 5 (right), we plot the curves of all three (expected) objective functions as the two weights vary, and mark the optimal values with red dots. We observe that the objective possesses a favorable

property—*convexity/concavity* with respect to the path weights. such a property ensures that simple first-order methods, i.e., gradient descent (GD), can effectively solve the problem. In particular, by iteratively updating the configuration in the direction of the steepest descent, one can guarantee convergence to the global optimum. We formalize this insight in the following theorem:

**Theorem 1.** *(Solving MCF with GD) Consider a GD algorithm with update rule $\pi^{(t+1)} = \pi^{(t)} - \eta v_t$. Then, there exists a step size $\eta > 0$ and a finite number of iterations $T$ such that $\mathcal{L}(\pi)$, the MCF objective function, attains the optimum up to an arbitrarily small error.*

Appendix D.3 presents the proof of Theorem 1, which hinges on the convexity/concavity of the three objective functions. In our simple example, executing GD requires precise knowledge of the distribution of demands. However, It is almost impossible to estimate the expected objective without exact prediction of the future demands. To address this, a more practical approach is to implicitly model the probability distribution using a DNN trained on extensive empirical data (like PRAM).

**What Makes PRAM Tick** Given the above case study, to approximate the optimal mapping from recent observations to configurations, the theoretical analysis of PRAM's effectiveness is intrinsically aligned with the GD algorithm. Before proceeding, we first present the convergence guarantee of PRAM's multi-agent adaptation:

**Lemma 1.** *Assume that both $R(s,a)$ and its Hessian $\nabla_\theta^2 R(s,a)$ are bounded. Let $\{\alpha_k\}_{k=0}^\infty$ be any step-size sequence satisfying $\lim_{k\to\infty} \alpha_k = 0$, $\sum_k \alpha_k = \infty$; and $\theta^{(k+1)} = \theta^{(k)} + \alpha_k v_k$, where $v_k$ is in the direction of the gradient. Then the policy iteration converges such that the expected gradient $\lim_{k\to\infty} \mathbb{E}_\pi \left[ \sum_i A_i(s,a) \nabla_{\theta^{(k)}} \log \pi_{\theta^{(k)}}(a_i \mid s_i) \right] = 0$.*

The proof of Lemma 1 (see Appendix D.4) refers to the approximation results from previous actor-critic algorithms (Bertsekas & Tsitsiklis, 1996; Sutton et al., 1999). Lemma 1 establishes the convergence of PRAM to a locally optimal agent, which implies that, once the fine-tuning is complete, our model can be regarded as a global mapping from observed demands to allocation schemes. Now back to MCF problems, our intuition that MLMs are capable of running the GD algorithm is again in the light of ICL. Concretely, it can approximate gradient-based few-shot learning within its forward pass (Fu et al., 2023), thereby "reasoning" its way to the solution. We formalize this intuition in Theorem 2, where we assume the MCF problem in the token space of PRAM as a linear regression.

**Theorem 2.** *(PRAM Learns to Implement GD) Let the learned context and the problem be well-defined under Assumption 2 and 3 in the Appendix. Then there exists an adapted MLM with constant depth and constant width that can simulate multiple steps of GD updates on the problem objective.*

The proof is deferred to Appendix D.5, which follows a simple weight construction introduced by Ahn et al. (2023). Taken together, the convexity of MCF problems (Theorem 1) and the ability of MLMs to simulate gradient descent (Theorem 2) establish that PRAM, once adapted, internally acts as an optimizer on the distance between initial configurations (input token embeddings) and near-optimal configurations (agent-decision embeddings)—consistent with our empirical results (§ 5).

## 5 MAIN RESULTS

We evaluate PRAM on open-source multimodal language models, with ⬡ Qwen2.5-VL-7B-Instruct (Bai et al., 2025) as a standard. Unless otherwise noted, we truncate and fine-tune the first *8* layers of its LM as the backbone. Implementation specifics are in Appendix C, and key experimental setups are summarized below with further details in Appendix B. This section sequentially includes comparative experiments on real-world and large-scale generated datasets, generalization analysis, and ablation studies. We also provide additional supplementary experiments (e.g., w.r.t number of layers, used MLMs, and demand distributions) in the Appendix E.

**Datasets.** Our evaluation is conducted on two groups of datasets. First, we leverage five real-world datasets, namely *Meta DB, Meta WEB, Abilene, CERNET*, and *GÉANT*. These datasets are relatively small in scale, each containing fewer than 30 nodes. To complement it, we also incorporate five large-scale topologies with synthetic data, including *GtsCe, Colt, UsCarrier, Cogentco*, and *Kdl*. The sizes of these topologies range from 100 to 800 nodes. In § 5.2, we employ a gravity model (Roughan et al., 2002) to generate synthetic demands. Additionally, we select the *12* most recently observed demands as historical information.

**Baselines.** We compare PRAM with six representative baselines: linear programming (*LP* (Gurobi Optimization, LLC, 2024)), partition-based optimization (*POP* (Cohen et al., 2021)), heuristic

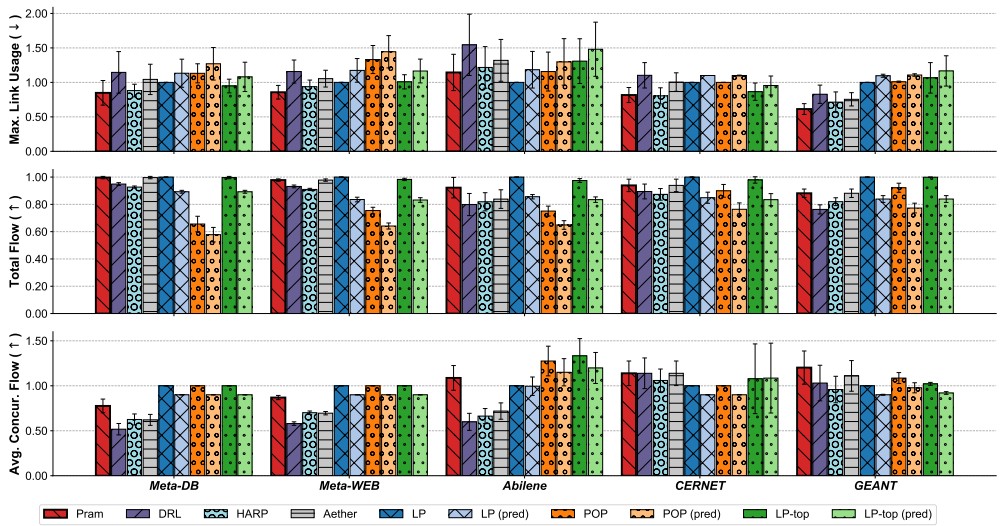

Figure 6: Real-world evaluation results of maximum link utilization (lower the better), allocated total flow (higher the better), and concurrent flow (higher the better). PRAM achieves top-tier performance.

approach (*LP-top* (Namyar et al., 2022)), reinforcement learning (*DRL* (Valadarsky et al., 2017)), and GNN-based machine learning (*HARP* (AlQiam et al., 2024), *Aether* (Fan et al., 2025)). Since some methods rely on the ground-truth demand, we also evaluate their predicted variants *(pred)* using the default moving-average forecasting, which we found to be simple yet effective (see Appendix E.3).

**Metrics.** We consider three standard objectives for the MCF problem: minimizing *maximum link utilization* (MLU), maximizing *total flow*, and maximizing *concurrent flow*. To enable consistent comparison across datasets, all reported results are *normalized*, which is the ratio of the result to the (near-)optimal result obtained using the LP optimization solver. We also measure the *computation time* of each approach on the same machine to assess their efficiency.

## 5.1 EXPERIMENT ON REAL-WORLD DATASETS

Our first experiment aims to answer: **how does PRAM perform in practice?** We present the results of the experiment in Figure 6, which evaluates the performance of our proposed PRAM over publicly available real-world datasets. In total, we run three independent sub-experiments, one for each MCF allocation objective. First, the results reveal substantial room for improvement when flows are allocated by the prior ML-based approach (i.e., DRL), likely due to the well-known training challenges of RL with randomly initialized networks. Second, the gap between the optimal solution and the partitioned solution produced by POP is evident, making POP the second-worst method. Third, while LP-top often performs comparably to LP, it exhibits less stability because it focuses exclusively on optimizing large flows. From the figure, we can also see that relative to their exact-demand counterparts, all predictive variants suffer performance degradation (no less than 10%), with the note that the ML-based methods use historical demand as input. Finally, we find PRAM achieves the second-best average performance across the three objectives, ranking just behind the LP solver that has a perfect future prediction, indicating it is able to achieve near-optimal by leveraging the reasoning power of MLMs. One surprising observation is that, in the case of minimizing link usage, PRAM's output configurations are even better than those of LPs (in particular, about 21% lower in CERNET and 45% in GÉANT). This may be because MLU exhibits stronger convexity properties—consistent with our theoretical results—and demonstrates more stable variation ranges, making it especially amenable to fine-tuning with MLMs. HARP performs second only to ours in MLU because it is specifically optimized for this objective. However, its performance on the other two objectives is subpar. In a clear example, on the CERNET and GÉANT datasets, its average allocated concurrent flow is even inferior to the simple DRL algorithm. While Aether shows an improvement over DRL, it fails to overcome the inherent RL limitations, resulting in a overall trend that is still inferior to PRAM and classical method. In conclusion, these findings suggest PRAM's good suitability for real-world deployment.

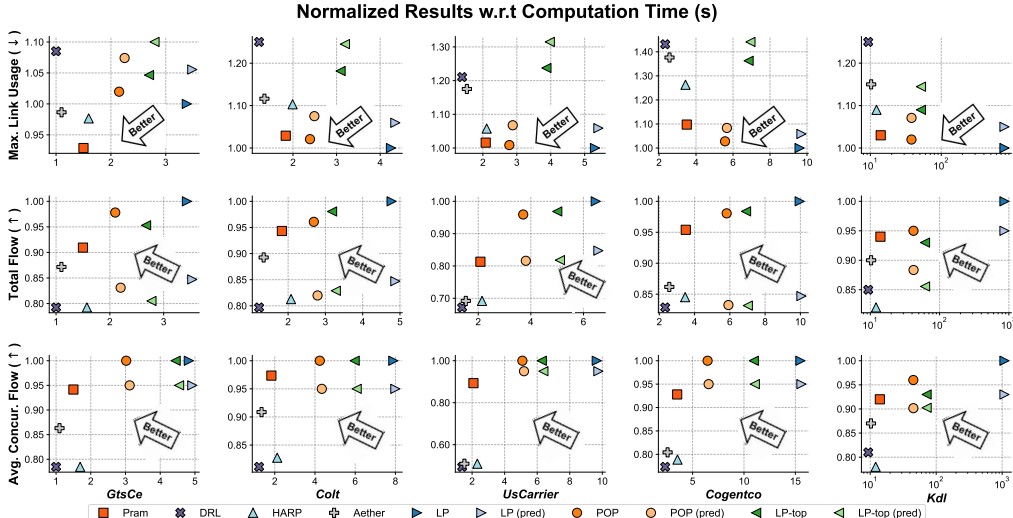

Figure 7: Comparison of PRAM with baselines on large-scale topologies using synthetic data. As the solution space expands, PRAM demonstrates scalable performance, achieving comparable or superior results with reduced computation time.

## 5.2 EXPERIMENT ON LARGE-SCALE DATASETS

Next, we experiment on larger network topologies to explore the following question: **how does PRAM work at scale?** Analogously, Figure 7 compares online testing results of different methods together with their running time. As the network size grows, PRAM demonstrates the intended scalability due to our dividing operation, delivering high-quality allocations with lower computation time. In particular, on the largest Kdl topology (754 nodes, 1,790 links, and over 2 million candidate path weights), PRAM completes each flow allocation in under 25 seconds on average—5× faster than POP, 7× faster than LP-top, and 100× faster than LP. For demand prediction, our non-parametric approach introduces negligible overhead. DRL, benefiting from a small policy network without MLM's backbone, achieves the fastest inference, but fails to approach near-optimal allocation quality. This is because the simple network also limits its expressive capability and lacks effective modeling of complex structures. In contrast, PRAM satisfies > 90% of the performance of the best-performing LP scheme that is close enough to the optimal. Additionally, we outperform HARP and Aether on average by 6.1% and 17.2% on MLU, by 16.6% and 7.3% on total flow, and by 24.8% and 13.5% on concurrent flow. HARP's recurrent optimization and invariant architecture limits its expressive capability, while Aether's link-path graph lacks effective modeling of large-scale topology structures and correlations. Since we inject Gaussian noise to mimic temporal demand variations, prediction-based schemes achieve slightly better results than with real-world traces, yet still lag behind PRAM. In summary, ML-based methods are fast but inaccurate, while LP-based methods perform well but are inefficient. By striking a balance, PRAM offers both efficiency and accuracy, making it a compelling choice for production MCF systems.

## 5.3 GENERALIZATION TO UNSEEN SITUATIONS

In this subsection, we assess our model's performance on unforeseen conditions, i.e., asking: **can PRAM generalize well?** Due to space constraints, we hereafter report results primarily in terms of link utilization. We consider two most representative situations, where unforeseen changes affect the two inputs of the MCF problem: topology (link failure) and commodity demand (flow fluctuation).

(1) *Coping with link failures.* PRAM addresses link failures by first reassigning the affected commodity flows using simple heuristics. Take the case of three candidate paths having weights of $(\frac{1}{2}, \frac{1}{5}, \frac{3}{10})$. If the first path experiences a failure, the adjusted weights would be proportionally reset as $(0, \frac{2}{5}, \frac{3}{5})$. Such reassignment requires minimal computation time, and the entire process operates at the millisecond-level scale, comparable to a standard inference step. In addition, the capacity of failed links will be marked as zero, and it will be depicted in the sub-image, which is then fed into PRAM. Figure 8(a) compares the performance of PRAM with prediction-based methods (access to

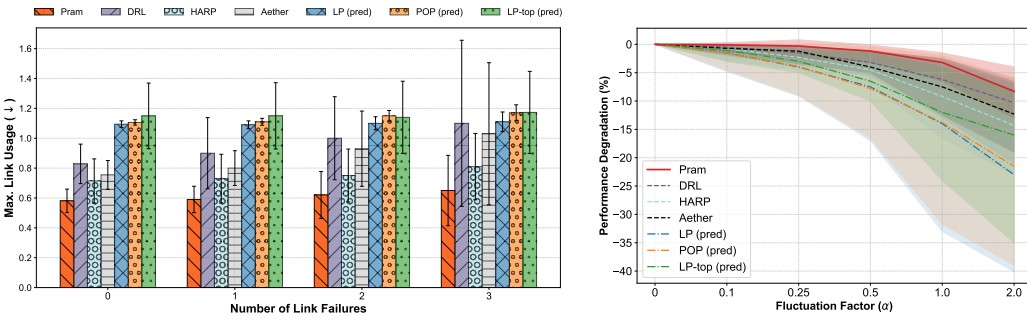

(a) MLU change with random link failures                    (b) Performance decline with fluctuation

Figure 8: Handling different unforeseen conditions on GÉANT; The right plot shows the mean and deviation from 10th to 90th percentile. The performance variations of PRAM are not significant.

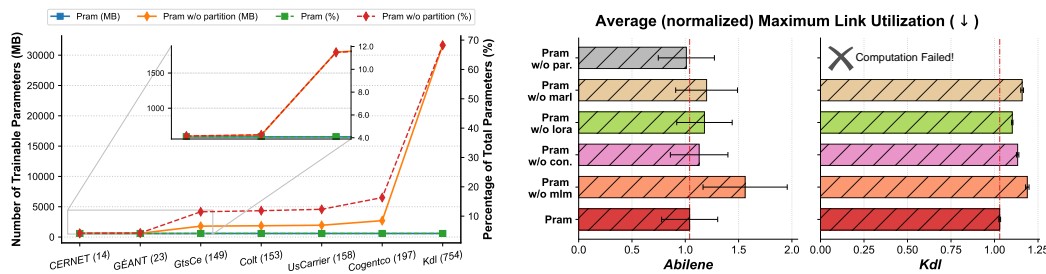

(a) # of trainable parameters with(out) partition        (b) Ablation study of PRAM's key components

Figure 9: Scalability and link utilization performance comparison between PRAM and its variants.

failures), in terms of link utilization, when different numbers of links fail in the GÉANT topology. For a fair comparison, we applied the same correction on all ML-based baselines. It can be seen that, except for HARP, they all have showed a significant decline, with an approximately 7% to 24% more MLU upon increased link failures. Our results show that PRAM still consistently outperforms all baselines without re-adaptation, indicating its generality across (transient) failures.

(2) *Reacting to flow fluctuation.* Our approach for increasing flow variability is as follows: we introduce independent noise into each source-destination pair by sampling from a Gaussian distribution $\mathcal{N}(\mu, \sigma_{s,d}^2)$, where $\mu = 0$ and $\sigma_{s,d}$ is the standard deviation of the demand $\mathcal{D}_{s,d}$. These noises are then multiplied by a factor $\alpha$ to control the fluctuation intensity. Also, figure 8(b) shows that the performance degradation of PRAM is no higher than 15% even for $\alpha = 2$ in the 90th percentile. Almost all the evaluated ML-based baselines handle small fluctuations effectively, but their performance declines noticeably as the fluctuations escalate to $\alpha > 0.5$. Meanwhile, the three classical methods struggle in this scenario, with a drop more than twice that of PRAM.

### 5.4 ABLATION STUDY AND VISUALIZATION

At the end of this section, we would like to investigate the final question: **how is PRAM effective?** We perform an ablation study to assess the impact of PRAM's key features on its overall performance.

(1) *Impact of Problem Partition.* Figure 9(a) compares the number of tunable parameters in PRAM with its non-partitioned variant, i.e., PRAM w/o partition (or w/o par. in short), across topologies of different sizes. Without partitioning, the parameter scale grows uncontrollably; for example, on the KDL topology, the number of trainable parameters is about 31,600 MB, which nearly matches that of full-parameter MLM adaptation. In contrast, PRAM localizes its parameters within LoRA and context modules—both inherently lightweight as shown in the figure—so its model size hardly increases with network scale. These results underscore the critical role of problem partition in handling MCF optimizations with large solution spaces.

(2) *Effect of Adaptation Choices.* To assess the importance of PRAM's core components, we retrain five variants on the Abilene and KDL topologies: ① PRAM w/o mlm: the MLM backbone is removed, with sub-topologies handled by a GNN and demands by FC layers; ② PRAM w/o con.: the global context is excluded from the input prompt; ③ PRAM w/o lora: all low-rank adapters are removed from

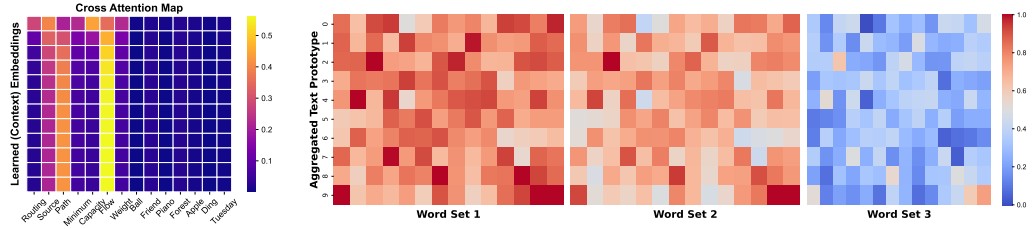

(a) Cross-attention map      (b) Visualization of ten different learned text prototypes

Figure 11: Learned representation interpretation of the context embeddings. In (a), each row represents a context embedding, while columns correspond to selected words. In (b), each row represents a linear aggregated prototype, while columns correspond to words from different sets.

the backbone; ④ PRAM w/o marl: the model is directly fine-tuned with the objective function end to end; ⑤ PRAM w/o par.: identical to PRAM w/o partition. The results are presented in Figure 9(b).

We can see that using MLM significantly improves performance in minimizing MLU, especially for Abilene. Moreover, all three designs (i.e., LoRA, learned context and multi-agent RL) used in our adaptation contribute to the optimization of the objective function. Although PRAM w/o partition achieves a lower MLU on the smaller network, it is too "gigantic" to make decisions on large Kdl. As a generality test, moreover, we evaluate the distinct single-failure scenarios and and artificially inject the 24 most influential failures to the GÉANT network. Figure 10 compares the performance of PRAM with its variant excluding MLM. Across all failure sce-

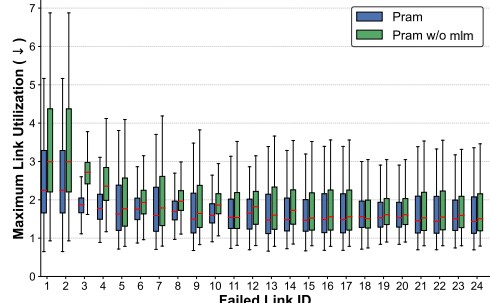

Figure 10: (un-normalized) MLU under different single-link failure scenarios for GÉANT.

narios, PRAM's median MLU ranges from 1.4 to 2.3, while that of PRAM w/o MLM ranges from 1.5 to 3 which is clearly not as effective under link failures because rescaling is not as effective as recomputing routes. It confirms PRAM's superiority to handle topology change not observed in the training data, by leveraging the pre-trained knowledge of MLMs.

(3) *Visualization of Learned Representation.* We present an interpretation of the learned context module on the Abilene dataset in Figure 11. As shown in Figure 11(a), we visualize the cross-attention map of a single head by replacing text prototypes with token embeddings of words either related or unrelated to the task. Each cell reflects the relevance between a row and a column (with brighter values indicating stronger correlation). To further examine how context is represented through different prototype combinations, we randomly select ten prototypes and visualize them in Figure 11(b). For this analysis, we consider three word sets — word set 1 (commodity):{"Flow", "Demand", . . . }, word set 2 (topology): {"Capacity", "Node", . . . }, and word set 3 (others): {"Monday", "February", . . . }. From the heatmaps, we make two key observations: first, prototypes exhibit strong relevance to words describing MCF problems; and second, PRAM successfully aligns its learned context with task-relevant word embeddings. So we can reasonably conclude that context satisfies our assumption.

# 6 CONCLUSION

In this paper, we present PRAM, a novel MCF solution that takes advantage of MLMs to optimize practical flow allocation performance. The MLM is equipped with a partition method to "divide" complexity and transform the original problem into a multi-agent decision-making problem. Then we design an adaptation (and communication) framework to "harmonize" the learning of these agents. We also theoretically study the effectiveness of PRAM in well-studied theoretical models. Our evaluations demonstrate that the PRAM is accurate, fast and robust, highlighting the potential of leveraging MLMs for "shooting" MCF problems. However, fine-tuning PRAM remains resource-intensive, even after truncating the backbone model. In addition, we acknowledge that the visual encoding scheme may introduce an inevitable bias, as discussed in Appendix E.5 & E.6. Due to space constraints, we defer discussion of several important limitations regarding PRAM and potential future directions to Appendix F.

## 7 ETHICS STATEMENT

We did not use any non-public data, unauthorized software, or API in our paper. In particular, the measured demand matrices, used in our evaluation, are aggregate counters between pairs of at the granularity of minutes (or coarser). They do not contain user IP addresses or packet contents. Thus, there are no privacy or other related ethical concerns.

## 8 REPRODUCIBILITY STATEMENT

The paper fully discloses all the information needed to reproduce the main experimental results of the paper to the extent that it affects the main claims and conclusions of the paper. Specifically, we detailed all the experimental setups and implementation details in § 5 and Appendix B & C. Moreover, our experimental codebase is publicly available at https://github.com/Y-debug-sys/PRAM.

## 9 ACKNOWLEDGMENT

This work is supported in part by the Key Research and Development Program of Zhejiang Province under Grant No. 2025C02103, in part by the National Natural Science Foundation of China under Grant No. 92373205 and No. 62572168, in part by the National Key Research and Development Program of China No. 2023YFB4404401, and in part by the Anhui Provincial Natural Science Fund (2508085MF151).

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

# Appendix

## TABLE OF CONTENTS

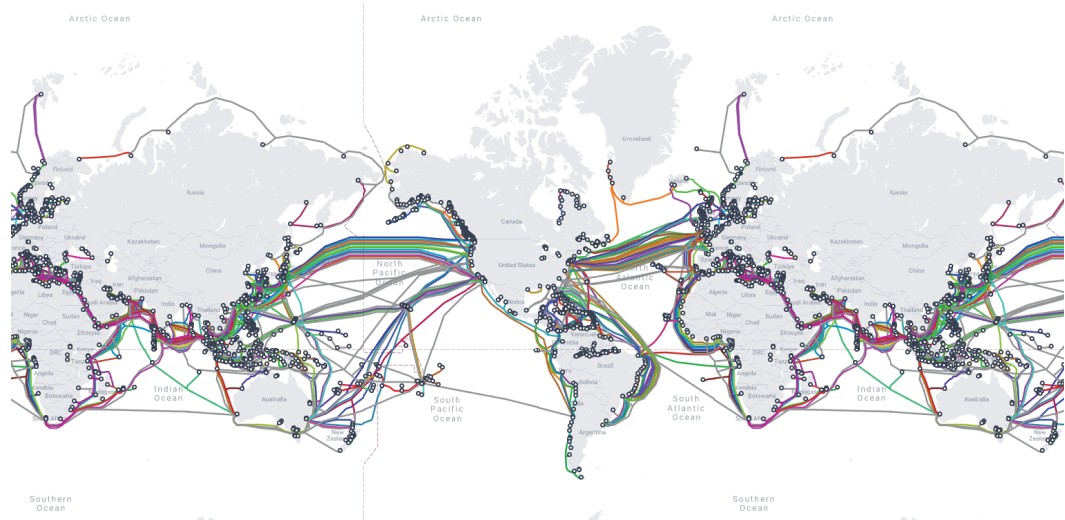

Figure 12: Submarine cables serve as choke points in large-scale topologies for network communication, providing a critical yet expensive infrastructure; the figure is drawn from this URL.

## A   RELATED WORKS

**Solving Multi-Commodity Flow Problems**   The multi-commodity flow (MCF) is a very classic resource allocation problem widely found in various fields such as network (Wang & Wang, 1999; Applegate & Cohen, 2003; Jain et al., 2013; Kumar et al., 2018; Abuzaid et al., 2021; Krishnaswamy et al., 2022), power (Blaauwbroek et al., 2015; Chalvatzis et al., 2019), transportation (Weiner, 1987; Schrijver, 2002; Wey & Wu, 2007; Noor-A-Rahim et al., 2020), logistics (Mandell, 1991; Balcik et al., 2014; Yan et al., 2022), could services (Chang et al., 2010; Akhter & Othman, 2016) and so on, playing a critical role in service and provider infrastructures. This part will primarily discuss works related to network traffic engineering (TE), which is the most extensively studied MCF problem. For the adaptive TE, LP solvers such as Gurobi are faster in practice, perhaps because they take larger steps towards the optimal allocation, but are still slow for the problem sizes considered here. Unlike adaptive TE, oblivious routing often optimizes worst-case link utilization across all possible demand matrices (DMs) (Racke, 2002; Applegate & Cohen, 2003; Azar et al., 2003). To fit current demands and react to failures, SMORE (Kumar et al., 2018) proposed semi-oblivious routing by dynamically adapting sending rates. An improved work is COPE (Wang et al., 2006), which achieves optimization within the space spanned by historical DMs without compromising worst-case performance bounds. However, this combined approach of generalizing oblivious routing and performing multi-matrix optimization incurs substantial computational overhead (especially for large-scale systems shown in Figure 12). Although NCFlow (Abuzaid et al., 2021) and POP (Narayanan et al., 2021) significantly speed up multi-commodity flow computations via problem decomposition and parallelization, their methods inherit the limitations of prediction-dependent approaches. Nowadays, researchers have also explored ML-based allocation (Xiao et al., 2021). These methods typically use demand-prediction and RL approaches (Valadarsky et al., 2017; Xu et al., 2018; Zhang et al., 2020; Hope & Yoneki, 2021; Bernárdez et al., 2021). For example, CFR-RL (Zhang et al., 2020) learns a policy to select critical flows for each given DM automatically. To explore the feasibility of combining GNNs with DRL, GDDR (Hope & Yoneki, 2021) and MARL-GNN (Bernárdez et al., 2021) transformed TE into online optimization problems over graphs to handle topologies of various structures. They represent the state-of-the-art recently, yet still exhibit obvious limitations—it often comes with high computational complexity and sensitivity to parameter settings. More recently, LMTE (Yuan et al., 2026) leverages large language models to solve TE problems, but it was not intentionally designed or optimized for large-scale systems. Compared to the above methods, PRAM is not only easy to train, but also several orders of magnitude faster than them when solving large-scale MCF problems.

**Investigating Reasoning Ability of Language Models**   Recent advancements in large language models have exhibited exceptional proficiency across diverse NLP tasks. Commercial LLMs such as ChatGPT (Achiam et al., 2023) and Gemini (Team et al., 2024) currently embody the state-

of-the-art in reasoning capabilities. Webb et al. (2023) first claimed that LLMs like GPT-3 have acquired an emergent ability to find zero-shot solutions to a broad range of analogy problems. González & Nori (2024) extended it beyond mathematics to vision domains. And Lin et al. (2024) investigated the application of in-context learning (ICL) in pre-trained models for decision-making tasks, while Shi et al. (2024) explored its efficacy in game-playing scenarios. For theoretical analysis of LMs' reasoning capabilities, Wang et al. (2024) shows that Transformers learn planning via path-finding by encoding adjacency/reachability matrices. Subsequently, Sanford et al. (2024) sharply separates Transformers from other architectures and provides width and depth separations via graph algorithms. Moreover, Li et al. (2024) demonstrates that chain-of-thought (CoT) enables bounded-depth transformers to solve serial computation problems, and Chen & Zou (2024) studied the capabilities of the transformer architecture with varying depth. Huang et al. (2025) proved that a transformer with CoT prompting can learn to perform multi-step gradient descent autoregressively. Several works have also established the existence of deep transformers capable of implementing gradient descent across different domains (Akyürek et al., 2023; Ahn et al., 2023; Mahankali et al., 2024; Nichani et al., 2024; Shen et al., 2024). In the theoretical part of our study, we also leverage the MLMs' advanced reasoning power to guide and assist in resource allocation.

**Adapting Language Models for TS or Graph** Here, we review previous work that employs LLMs for time series (TS) and graph applications. The first category is to boost time series related tasks (Jin et al., 2024b), such as forecasting, imputation, and anomaly detection. To adapt LLMs for time series forecasting, Chang et al. (2023) employed a two-stage approach: fine-tuning GPT-2's transformer module and redesigning positional encoding. Besides, Zhou et al. (2023) propose a parameter-efficient adaptation method for pre-trained language models, named "One Fits All", that preserves their original self-attention and feedforward architectures. Time-LLM (Jin et al., 2024a) further proposed an embedding reprogramming technique to align the language model's word embedding space with time-series representations. Moreover, Zhou & Yu (2025) conducted a comprehensive investigation into LLMs' understanding of time series data and their anomaly detection capabilities. The second category encompasses graph tasks, which typically include graph learning and graph algorithms. For node classification tasks in text-attributed graphs, GraphGPT (Tang et al., 2024) processed the input graph through a GNN encoder for tokenization before LLM integration. Following this paradigm, MolCA (Liu et al., 2023) applied similar GNN-based encoders to handle molecular structures when predicting their properties. To enable LLMs to reason over graph-structured data, NLGraph (Wang et al., 2023) and Talk like a Graph (Fatemi et al., 2024) introduced a natural language translation step, followed by few-shot prompting or CoT techniques for effective inference. LLaGA (Chen et al., 2024) reformatted the center node and its neighborhood into a structure-aware textual representation, and Wu et al. (2024b) conducted a pioneering investigation into graph-learning methodologies for task planning in language agents. Unlike existing approaches, this paper advances MCF solutions as a novel framework to bridge the gap between these two research trajectories.

# B    DETAILED EXPERIMENTAL SETUP

In this section, we provide the detailed experimental setup to ensure clarity of our paper. We first outline the objectives of the experiments, followed by the datasets used in our study. We then introduce the baseline methods for comparison and, finally, describe the path selection strategy.

## B.1    MCF OBJECTIVES

Using the notations introduced in § 2, we define three optimization objectives as follows.

(1) **Maximum Link Utilization (MLU)**: It refers to the maximum value of all link utilization ratios in the topology, which is a classical allocation objective. A lower link utilization suggests greater resilience. Equation (1) summarizes the formulation.

$$
\begin{aligned}
\textbf{minimize} \quad & \alpha = \max_{e \in \mathcal{E}} \frac{f_e}{c_e} \\
\textbf{subject to} \quad & f_e = \sum_{s,t \in V} \sum_{p \in P_{s,t}} \sum_{e \ni p} \mathcal{D}_{s,t} \cdot r_p, \quad \forall e \in \mathcal{E} \\
& r_p \geq 0, \quad \forall s,t \in \mathcal{V}, \, \forall p \in P_{s,t} \\
& \sum_{p \in P_{s,t}} r_p = 1.0, \quad \forall s,t \in \mathcal{V}
\end{aligned}
\tag{1}
$$

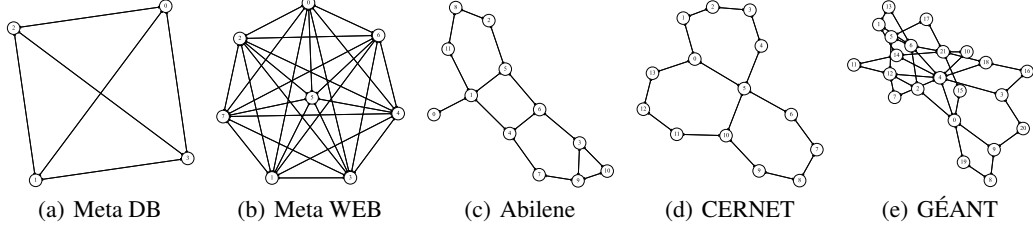

| (a) Meta DB | (b) Meta WEB | (c) Abilene | (d) CERNET | (e) GÉANT |

Figure 13: Five real-world network topologies of various systems.

(2) **Maximum Total Flow (MTF)**: The objective computes a policy that satisfies the demand and capacity constraints while maximizing permissible flow. We here additionally define a pair-wise capacity $\omega_p$ that represents the maximum permissible flow between $s$ to $t$ along the tunnel $p$. $\omega_p$ is also an optimizable component of the configuration. Then the optimization problem is formulated as follows.

$$
\begin{aligned}
\textbf{maximize} \quad & \alpha = \sum_{s,t \in \mathcal{V}} \sum_{p \in P_{s,t}} f_p \\
\textbf{subject to} \quad & f_p \leq \mathcal{D}_{s,t} \cdot r_p, \quad \forall s,t \in \mathcal{V}, \ \forall p \in P_{s,t} \\
& f_p \leq \omega_p, \quad \forall s,t \in V, \ \forall p \in P_{s,t} \\
& r_p \geq 0, \ \omega_p \geq 0, \ f_p \geq 0, \quad \forall s,t \in \mathcal{V}, \ \forall p \in P_{s,t} \\
& \sum_{s,t \in V} \sum_{p \in P_{s,t}} \sum_{e \ni p} \omega_p \leq c_e, \quad \forall e \in \mathcal{E} \\
& r_p \geq 0, \quad \forall s,t \in \mathcal{V}, \ \forall p \in P_{s,t}
\end{aligned}
\tag{2}
$$

(3) **Maximum Concurrent Flow (MCF)**: In maximum-concurrent-flow problems, the objective is to compute a configuration that maximizes the total network throughput. Specifically, it depends on a surrogate bound $\alpha$, which is an $\alpha$-fraction of each $\mathcal{D}_{s,t}$ is routed concurrently. By defining $f_{s,t}^{(p)}$ as the actual commodity flow on the tunnel $p$ between $s$ and $t$, the surrogate problem can be formally expressed as

$$
\begin{aligned}
\textbf{maximize} \quad & \alpha \in [0,1] \\
\textbf{subject to} \quad & \sum_{p \in P_{s,t}} f_{s,t}^{(p)} \geq \alpha \cdot \mathcal{D}_{s,t}, \quad \forall s,t \in \mathcal{V} \\
& \sum_{s,t \in V} \sum_{p \in P_{s,t}} \sum_{e \ni p} f_{s,t}^{(p)} \leq c(e), \quad \forall e \in \mathcal{E} \\
& f_{s,t}^{(p)} \geq 0, \quad \forall s,t \in \mathcal{V}, \ \forall p \in P_{s,t} \\
& f_{s,t}^{(p)} \leq \mathcal{D}_{s,t} \cdot r_p, \quad \forall s,t \in \mathcal{V}, \ \forall p \in P_{s,t} \\
& r_p \geq 0, \quad \forall s,t \in \mathcal{V}, \ \forall p \in P_{s,t}
\end{aligned}
\tag{3}
$$

However, since in many cases the minimum concurrent flow is either very close to zero or exactly zero and thus not representative, in the experiments we instead use the average satisfaction ratio across all flows to capture the intended fairness effect.

## B.2 EXPERIMENTAL DATASETS

In this subsection, we provide detailed information on the datasets used in all experiments. Note that all these datasets and topologies are publicly available, and their corresponding access links can be found in the references.

### B.2.1 REAL-WORLD DATASETS

We first consider two small-scale social network datasets, **Meta DB** and **Meta WEB** (Roy et al., 2015). The Meta DB cluster consists of MySQL servers that store user data and process SQL queries,

| Topology | # Nodes | # Links | # Total DMs | Granularity | Collection Time |
|----------|---------|---------|-------------|-------------|-----------------|
| Meta DB | 4 | 12 | 10,000 | \ | \ |
| Meta WEB | 8 | 56 | 10,000 | \ | \ |
| Abilene | 12 | 30 | 48,384 | 5 minutes | March $\sim$ September 2004 |
| CERNET | 14 | 32 | 10,000 | 5 minutes | February $\sim$ March 2013 |
| GÉANT | 23 | 74 | 10,733 | 15 minutes | January $\sim$ April 2004 |

Table 1: Real-world network topology statistics

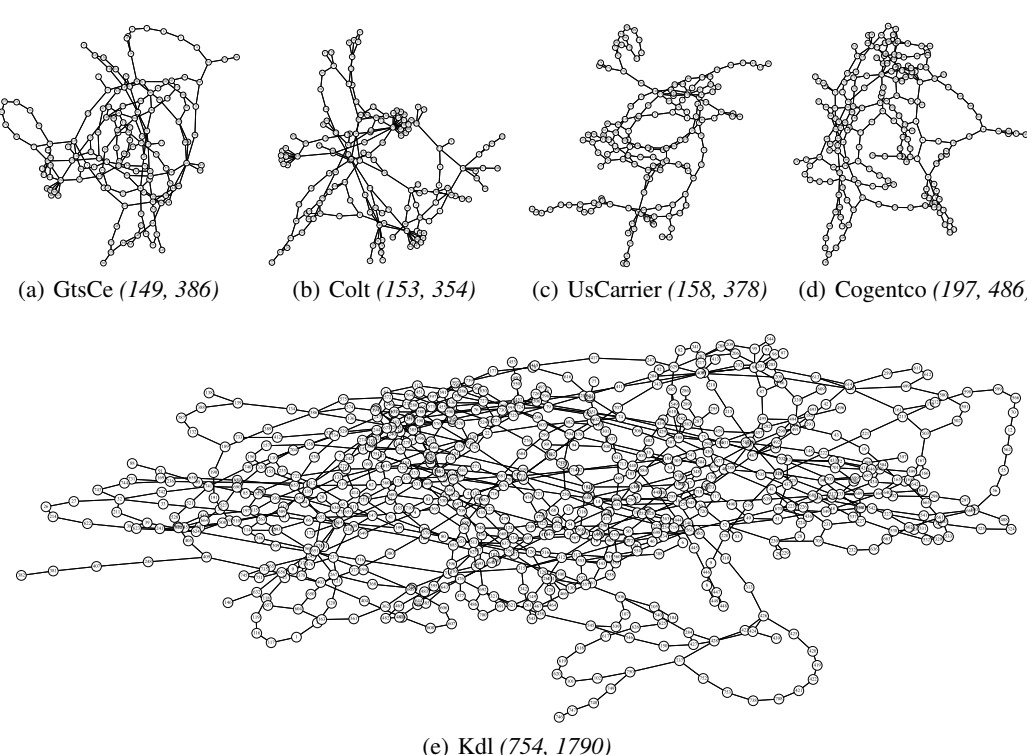

(a) GtsCe *(149, 386)*   (b) Colt *(153, 354)*   (c) UsCarrier *(158, 378)*   (d) Cogentco *(197, 486)*

(e) Kdl *(754, 1790)*

Figure 14: Five large-scale topologies used in our evaluation.

while the Meta WEB cluster handles web traffic. Both clusters operate at the Point of Delivery (PoD) level, which is fully connected. We also use three real-world WAN datasets: US Internet2 Network (**Abilene**, Zhang et al. (2005)), Pan-European Research Network (**GÉANT**, Uhlig et al. (2006)) and China Education and Research Network (**CERNET**, Li et al. (2011)), each with public DMs captured at different snapshots. Table 1 presents key statistics (i.e., number of nodes, links, and samples) for these datasets. Besides, we visualize their topologies in Figure 13. All DMs are normalized by ten times the maximum link capacity of their topologies for DNN training and testing (hereinafter the same).

### B.2.2 SYNTHETIC DATASETS

To evaluate PRAM's ability to scale to larger systems, we also extracted several topologies from the Internet Topology Zoo (Knight et al., 2011). They are **GtsCe** with 149 nodes and 386 edges, **Colt** with 153 nodes and 354 edges, **UsCarrier** with 158 nodes and 378 edges, **Cogentco** with 197 nodes and 486 edges, and **Kdl** with 754 nodes and 1790 edges. Figure 14 shows visualizations for the five used topologies; note that we set *1,000* for all link capacities. Also note that we only take the largest strongly connected component in these topologies for our evaluations.

We synthesize DMs for the above large topologies based on three typical types of distribution or method[1]. All datasets except for Kdl, the number of generated samples is *3,000* while that of Kdl is *500* due to its super-high dimension. We present their details as follows:

• *Gravity* (Roughan et al., 2002): The distribution has been an empirical success in that it accurately predicts trade flows between countries for many goods and services. Its key idea is that the total demand leaving a node is proportional to the total capacity on the node's outgoing links; this demand is divided among other nodes proportional to the total capacity on their incoming links. We present the Python pseudocode for the corresponding synthetic data in Algorithm 1. Specifically, the algorithm first computes the total inbound and outbound capacities for all nodes. It then derives the DMs by sampling from a Gaussian distribution centered at the expected fraction, with variance proportional to the same value. Finally, the resulting demand matrix is rescaled by a factor $r$ to match the desired total resource level.

---

**Algorithm 1** Gravity-based Demands Generation in a Python style

---

```python
# G: graph or topology
# r: scaling factor

num_nodes = len(G.nodes) # number of nodes
in_cap, out_cap = {}, {}
A = zeros(num_nodes, num_nodes) # initialize DMs

for u in G.nodes:
    in_cap[u] = sum(cap(v, u) for v in predecessors(u))
    out_cap[u] = sum(cap(u, v) for v in successors(u))

for u in G.nodes:
    norm_u = out_cap[u] / sum(out_cap)
    for v in G.nodes:
        frac = norm_u * in_cap[v] / (sum(in_cap) - in_cap[u])

        # allocate resource with randomness
        A[u, v] = max(Gaussian(frac, frac/4), 0)

A = A * r # scale by constant factor
```

---

• *Poisson* (Tebaldi & West, 1998): This model can be formally expressed as $P(\lambda, \delta)$, where the demand between nodes $s$ and $t$ follows a Poisson random variable with mean $\lambda\delta^{d_{s,t}}$. Here, $d_{s,t}$ denotes the hop distance of the shortest path between $s$ and $t$, and $\delta \in [0, 1]$ is a decay factor controlling demand concentration. Choosing $\delta$ close to 0 yields highly concentrated demands (favoring nearby nodes), while values close to 1 generate more uniform demands across the topology. The corresponding Python-style pseudocode is given in Algorithm 2. The algorithm first computes the all-pairs shortest path distances. For each source–destination pair, the expected demand is set to $\lambda\delta^{d_{s,t}}$, which is then sampled from a Poisson distribution to incorporate randomness. The diagonal entries are zeroed to exclude self-demands. Finally, the entire demand matrix is scaled by a factor $r$ to adjust the total resource level.

• *Bimodal* (Applegate & Cohen, 2003): In the bimodal model, a fraction $p$ of source–destination pairs (selected uniformly at random) are assigned demands drawn from Uniform$(b, c)$, while the remaining pairs receive demands from Uniform$(0, a)$. This construction creates a heterogeneous demand matrix where some pairs experience significantly higher traffic than others. Algorithm 3 provides Python-style pseudocode for generating synthetic data under this model. The algorithm first initializes an empty demand matrix. It then uses a random assignment to decide the group for each entry. Depending on the group, entries are sampled from the corresponding uniform distribution. Finally, the diagonal entries are set to zero. This yields a demand matrix with a bimodal distribution of traffic intensities.

---

[1] The code is mainly based on this Github repository: https://github.com/netcontract/ncflow.

---

**Algorithm 2** Poisson-based Demands Generation in a Python style

---

```python
# G: graph or topology
# lam: base Poisson parameter
# decay: decay factor (<=1)
# r: scaling factor

num_nodes = len(G.nodes) # number of nodes

# compute all-pairs shortest distances
distances = zeros(num_nodes, num_nodes)
for src, dist_dict in shortest_path_length(G):
    for target, dist in dist_dict.items():
        distances[src, target] = dist

# generate Poisson-based demand matrix
A = array([[poisson(lam * (decay ** dist)) for dist in row]
          for row in distances], dtype=float)
np.fill_diagonal(A, 0.0) # no demand for self

A = A * r # scale by constant factor
```

---

**Algorithm 3** Bimodal-based Demands Generation in a Python style

---

```python
# G: graph or topology
# fraction: fraction of low-range allocations
# low_range: [min, max] for low allocations
# high_range: [min, max] for high allocations

num_nodes = len(G.nodes) # number of nodes

# initialize demand matrix
A = zeros(num_nodes, num_nodes)

p=[fraction, 1 - fraction]

# randomly choose low/high demand for each entry
inds = random_choice_0_1(num_nodes, num_nodes) with a probablity of p

# low allocations
A[inds] = uniform(low_range[0], low_range[1], sum(inds))

# high allocations
A[~inds] = uniform(high_range[0], high_range[1], sum(~inds))

fill_diagonal(A, 0.0) # no demand for self
```

---

### B.3 BASELINES

In this paper, we compare PRAM against the following state-of-the-art MCF solutions[2]:

• **DRL** (Valadarsky et al., 2017): It leverages deep reinforcement learning to replace explicit demand prediction with end-to-end optimization, mapping recent DMs to MCF configurations. Specifically, we train a policy neural network of the same size as PRAM's trainable parameters, using OpenAI's proximal policy optimization (PPO) algorithms (Schulman et al., 2017).

• **HARP** (AlQiam et al., 2024): It is a GNN-based scheme which ensures invariances to natural input transformations (e.g., permutations of node ids, tunnel reordering), and has a neural architecture

---

[2]Implementations of all non-open-source baseline methods can be found in our codebase. For HARP and Aether, we follow the settings in their papers and experiments.

| Approach | Abilene | GÉANT | UsCarrier | Cogentco |
|:---:|:---:|:---:|:---:|:---:|
| **LP-w** | $19.67 \pm 8.26$ | $1.60 \pm 0.54$ | $67.88 \pm 7.25$ | $73.39 \pm 2.87$ |
| **LP-f** | $22.56 \pm 6.47$ | $2.87 \pm 1.09$ | $10.08 \pm 0.03$ | $17.08 \pm 0.06$ |

Table 2: Comparing different optimization scheme for minimizing MLU objective.

aligned to optimization models. HARP has focused on the MLU objective, so we extended the other two objective functions according to our framework. Due to its end-to-end nature, this extension is totally fine. Also note that different from HRAP's original implementation which uses exact demands as input of models, we here modified it with historical information.

• **Aether** (Fan et al., 2025): It is a most recently proposed algorithm, which excels in generalizing across different networks and different amounts of demands. In addition to a graph Transformer, Aether also employs a differentiated traffic strategy based on multi-agent RL to focus on larger flows for better learning while handling small flows with rules.

• **LP** (Gurobi Optimization, LLC, 2024): It solves the MCF optimization problem for all demands using linear programming (LP), in which Gurobi (version 11.0.3) optimization solver is employed. The code for the different MCF objectives is listed in Algorithm 4 to Algorithm 6. We include two variants for minimizing MLU: one that treats path weights as the decision variables (LP-w) and another that optimizes over path flows (LP-f). As shown in Table 2, the two formulations yield solutions of noticeably different quality. LP-w performs reasonably well on small topologies, but its effectiveness drops sharply at larger scales. Given the stability and consistency of the results, we adopt LP-f as the default baseline throughout this paper.

• **POP** (Narayanan et al., 2021): It also decomposes large-scale granular flow problems into subproblems that are solved in parallel using an LP solver. Specifically, the entire topology is replicated $k$ times, with each replica assigned $\frac{1}{k}$ of the original link capacities. Commodity demands are then *randomly* distributed across these replicas, and each subproblem is solved independently in parallel. We set $k$ according to the topology size: $k = 2$ for the five real-world datasets, $k = 12$ for four large-scale topologies other than KDL, and $k = 128$ for KDL.

• **LP-top** (Namyar et al., 2022): It implements a simple yet effective heuristic algorithm that is recently revealed as "demand pinning". It allocates the top 10% of demands using an LP solver and assigns the remaining demands to the shortest paths.

• ∼ **(pred)**: Since most of the non-learning method (i.e., LP, POP, and LP-top) optimizes the solution with perfect knowledge of future demands, we consider their practical implementation based on predicted future demands. For simplicity and efficiency, we primarily use a non-parametric weighted moving average for prediction. Experiments on other prediction schemes are presented in Appendix E.3.

---

**Algorithm 4** Optimize weights for MLU in a Gurobi style

```
# E: set of edges, each with capacity C[e]
# PathToEdge[p, e] = 1 if path p uses edge e

# create optimization model
m = Model(); mlu = m.addVar(lb=0)
for each pair and path k in P(s,d): add variable w[s,d,k] in [0,1]

# add constraints
for each (s, d): sum_k w[s,d,k] == 1
for each demand D and each edge e in E:
    load = sum over (s,d,k): w[s,d,k] * D[s,d] * PathToEdge[(s,d,k), e]
    load <= mlu * C[e]

# minimize maximum link utilization
m.setObjective(mlu); m.optimize()
```

---

---

**Algorithm 5** Optimize flows for MLU in a Gurobi style

---

```
# E: set of edges, each with capacity C[e]
# PathToEdge[p, e] = 1 if path p uses edge e

# create optimization model
m = Model(); mlu = m.addVar(lb=0)
f = {p: m.addVar(lb=0) for p in all_paths}

# add constraints
for each edge e:
    load = sum over p: f[p] * PathToEdge[p, e]
    load <= mlu * C[e]
for each commodity k: sum over p in P[k]: f[p] == D[k]

# minimize maximum link utilization
m.setObjective(mlu); m.optimize()
```

---

---

**Algorithm 6** Optimize flows for MTF & MCF in a Gurobi style

---

```
# E: set of edges, each with capacity C[e]
# PathToEdge[p, e] = 1 if path p uses edge e

# create optimization model
m = Model(); f = {p: m.addVar(lb=0) for p in all_paths}

if objective == "MTF":
    # maximize total delivered flow
    obj = sum over all p: f[p]
else:
    # maximize concurrent flow factor alpha
    alpha = m.addVar(lb=0, ub=1)
    for each commodity k: sum_{p in P[k]} f[p] >= alpha * D[k]
    obj = alpha

# edge capacity constraints
for each edge e:
    load = sum over p: f[p] * PathToEdge[p, e]
    load <= C[e]

# per-commodity flow cannot exceed demand
for each commodity k: sum_{p in P[k]} f[p] <= d[k]

# optimize
m.setObjective(obj, maximize=True); m.optimize()
```

---

### B.4 CANDIDATE PATH SELECTION

For each topology, unless otherwise specified, we use Yen's algorithm (Yen, 1971) to precompute up to **4 shortest paths** between every pair of nodes, which serve as candidate paths for multi-commodity flow allocation. If fewer than 4 paths exist, the last available path is repeated to ensure a total of 4.

## C PRAM IMPLEMENTATION

We implement PRAM using PyTorch (Paszke et al., 2019) and HuggingFace Transformers (Wolf et al., 2019). Most experiments are conducted on a Linux server equipped with two NVIDIA A100 GPUs (80GB memory each). And we adopt a 7:1:2 ratio for training, validation, and testing across all datasets. For model initialization, we employ Kaiming uniform initialization for the low-rank matrices. Additional implementation details are provided below.

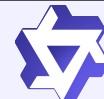

> # Subtask Prompt Structure
>
> **## Role :** You are an expert in solving multi-commodity flow problems.
>
> **## Task :** Based on `<|vision_start|><|image_pad|><|vision_end|>` the all traversed links from a source node (marked in red) to others in the image and other information, determine candidate path weights for each source-destination pair in the near future. ** **Current Source node need to optimize is ``** **.
>
> **## Goal :** ** **The objective is to minimize `<maximum link utilization>`** ** . Find a set of candidate path weights (split ratios) that minimizes the highest utilization across all links in the network.
>
> **## Description :** There are `<num_nodes>` nodes in the topology and `<num_paths>` shortest paths between each node pair are used as candidate paths. Note that the sum of the candidate path weights should equal 1 . All commodity demands mu-st be routed (conservation of flow), while weights must be positive floats. ······ (other descriptions)
>
> **## Demands :** The most recent demands from (``, 0) to (``, `<num_nodes>` - 1) are as follows.\n `<x_1, x_2, ···, x_n>`; and history mean demands from (``, 0) to (``, `<num_nodes>` - 1) are as follows. \n `<m_1, m_2, ···, m_n>`. ······ (other statistics)
>
> **## Format :** Make sure to follow the answer template. [source-0-path1-weight, ···, source-0-pathx-weight], [source-1-path1-weight, ···source-1-pathx-weight],···; And no allocation is needed for the source node to itself.

Figure 15: Example subtask prompt structure for maximum link utilization optimization.

## C.1 PROMPT STRUCTURE

We show the overall prompt structure used by PRAM for minimizing maximum link utilization in Figure 15. Our design can be roughly divided into three parts: task instructions, domain knowledge, and commodity information, as illustrated below. (i) Task instructions: The role section positions the MLM as an expert in multi-commodity flow problems, ensuring that subsequent reasoning is guided by a subtask-oriented perspective. The task and goal sections clearly specify the optimization target—minimizing maximum link utilization—while constraining the solution to path weights that obey feasibility requirements (e.g., non-negativity and normalization); (ii) Domain knowledge: The description encodes domain knowledge, such as the number of nodes, candidate paths, and flow conservation principles, thereby narrowing the search space toward admissible allocations; (iii) Commodity information: The demands section conveys the actual commodity information, including most recent traffic volumes and their historical statistics, enabling the model to reason about both instantaneous and smoothed demand dynamics. Finally, the format clause enforces a deterministic answer template, which facilitates automatic parsing and validation of the policy head network. The template for the maximum flow and concurrent flow is essentially the same as the one outlined above.

**Algorithm 7** Draw sub-images per source in a Python style

```
# G: network graph

# group by source
grouped_paths = {s: all paths starting from s}

for s, paths in grouped_paths.items():
    subG = combine all edges in paths
    layout(subG)      # compute layout positions and scales
    draw all nodes and highlight source node s
    draw all edges, add arrows if G is directed
    for edge in subG.edges: draw edge label = G[edge]['capacity']
```

## C.2 SUB-IMAGE PLOTTING

We utilize Matplotlib (Hunter, 2007) and NetworkX (Hagberg et al., 2008) to generate visual representations of the sub-graph data. We draw all candidate paths grouped by source node. Each

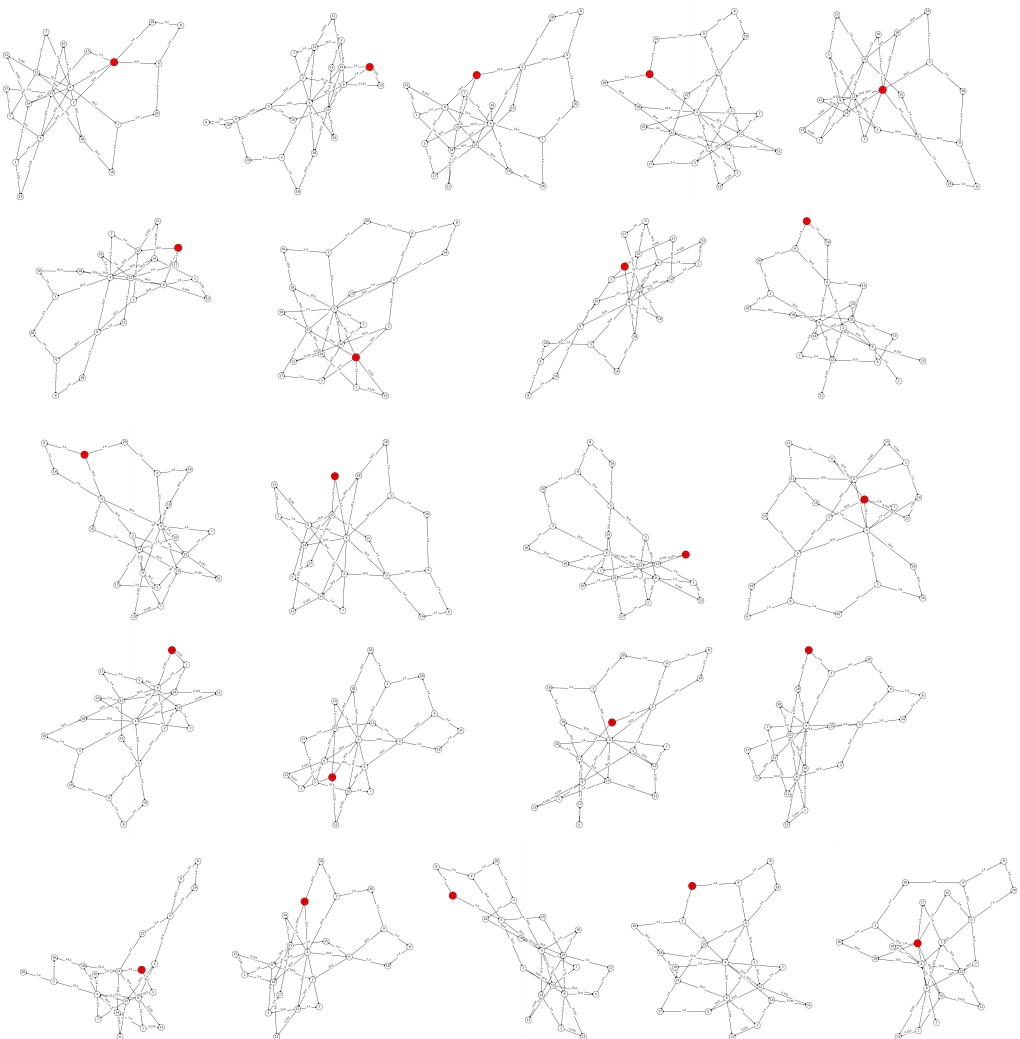

Figure 16: Examples of input sub-images on GÉANT topology. In each image, the starting point of the commodity flow is marked in red, only the links that exist in the candidate paths are drawn, and the link capacities are directly indicated on the links.

source gets one figure containing all its candidate paths (to all destinations). The pseudo-code is presented in Algorithm 7. These visualizations are then provided to multimodal LLMs capable of processing image inputs (.png file with 240 dpi). Figure 16 shows all sub-image examples from the GÉANT topology, where its source node is marked in red and the link capacities are explicitly indicated in the figure.

## C.3    ADAPTATION ALGORITHM

We next provide implementation details that complement the main text. We first outline how objectives (i.e., MLU, MTF and MCF) are computed efficiently as rewards using parallel matrix operations, followed by the multi-agent RL procedure adopted to fine-tune PRAM.

### C.3.1    OBJECTIVE (REWARD) COMPUTATION

For MLU, we train PRAM to output all path weights (with Sigmoid non-linear as its final layer). The mapping between these weights and the resulting MLU can be represented using Algorithm 8. It is

noteworthy that our reward computation relies on real samples (demand at future time steps), whereas the model only has access to historical information. No need for any loop operations, this mapping can be established through simple matrix operations in parallel. Regarding MTF and MCF, we force PRAM to directly predict allocated demand $w_p$ for each candidate path (with ReLU non-linear as its final layer). As seen in Algorithm 9, we then scale $w_p = \frac{w_p}{\gamma}$, where $\gamma = \max\left(\max_{e \in \mathcal{E}} \sum_{p:e \in p} \frac{w_p}{c(e)}, 1\right)$. In this way, PRAM guarantees that no link capacity can be exceeded, and the computation is also fast.

---

**Algorithm 8** Compute MLU in a PyTorch style

---

```
# w: path weights (split ratios)
# d: commodity demand vector
# C: link capacities
# PathtoEdge: path-to-edge incidence (1 if edge j in path i else 0)
# SDtoPath: SD-to-path incidence (1 if path j serves commodity i else 0)

# normalize path weights
total_w = SDtoPath @ w
w = w * (SDtoPath.T @ (1.0 / total_w))

# demand on paths and flow on edges
demand_on_paths = (SDtoPath.T @ d.T) * w
flow_on_edges = PathtoEdge.T @ demand_on_paths

# maximum link utilization
congestion = flow_on_edges / C
MLU = max(congestion)
```

---

**Algorithm 9** Compute MTF & MCF in a PyTorch style

---

```
# w_p: predicted demand vector
# d: commodity demand vector
# C: link capacities
# PathtoEdge: path-to-edge incidence (1 if edge j in path i else 0)
# SDtoPath: SD-to-path incidence (1 if path j serves SD i else 0)

# scale planned demands on each path
w_p_scaled = w_p / max((PathtoEdge.T @ w_p) / C)

# compute flow per SD pair, limited by demand
flow = min((SDtoPath @ w_p_scaled).T, d)

# total flow
MTF = sum(flow)

# maximum concurrent ratio
MCF = min(flow[d != 0] / d[d != 0])
```

---

### C.3.2 FINE-TUNING DETAILS

As detailed in the main text (§ 3.3), PRAM opts for employing the adaptation framework of multi-agent RL. Since MCF allocations are deterministic, we adopt the standard approach of modeling $\pi_\theta(a_i|s_i)$ as a Gaussian distribution: during training, actions are sampled from this distribution, while at deployment the mean is used as the deterministic allocation. To be more specific, we employ a separate one-layer MLP to predict the mean, while keeping the standard deviation fixed[3]. After sampling from the corresponding Gaussian distribution, the log probability is calculated as: $\log \pi_\theta(a_i|s_i) = -\frac{1}{2}\left(\log(2\pi\sigma_i^2) + \frac{(a_i - \mu_i)^2}{\sigma_i^2}\right)$, which is implemented by

---

[3]We also experimented with learnable standard deviations in our early trials, but the results were suboptimal.

torch.distributions.normal.log_prob. Algorithm 10 presents the pseudo-code for fine-tuning PRAM. It leverages the observation that allocations in MCF problems computed for a single time interval do not influence subsequent intervals (e.g., the demand matrices). This domain-specific insight enables us to simplify training by reducing the long-term return to a one-step return. Furthermore, note that the subproblem decomposition and the .png image transformation are performed only once at the beginning of the task, ensuring that they do not compromise the efficiency of model fine-tuning.

---

**Algorithm 10** Fine-tuning PRAM in a PyTorch style

---

```
# Model: policy network (based on MLM)
# Optimizer: optimizer for Model
# episodes: sequence of training episodes

for observation in loader:
    # observation: pre-processed sub-images and text prompts
    loss = 0
    for step in episode:
        action, log_probability = Model(observation)     # select action
        advantage = compute_objective(action)     # get multi-agent reward

        # accumulate policy gradient
        loss += (- log_probability * advantage).mean()

    Optimizer.zero_grad()
    loss.backward()
    Optimizer.step()

    if EarlyStop and converged(): break
```

---

### C.4 FRAMEWORK DETAILS

In this subsection, we first outline the architectural information of PRAM and provide an overview of our hyperparameter choices.

#### C.4.1 ARCHITECTURE

As shown in Figure 17, given a Qwen2.5-VL-style multimodal language model, images are inserted at the position marked by "$< |\text{vision\_start}| >< |\text{image\_pad}| >< |\text{vision\_end}| >$" (see Figure 15). Text and images are arranged in the order they appear in the original prompt. The context is prepended to the input as a prefix. We apply LoRA fine-tuning exclusively to the attention blocks of the language model, specifically restricted to the QV projections. As described in the preceding subsection, the final stage of PRAM's architecture samples actions (i.e., path weights) from a Gaussian distribution parameterized by the predicted mean and a fixed standard deviation.

#### C.4.2 HYPERPARAMETER

Since PRAM primarily relies on the multimodal interface of the pre-trained model, most components (including token embedding dimensions) are fixed, leaving only a limited set of parameters to tune, as detailed below. First, we set the length of the learnable context proportional to the number of nodes in the topology, with a minimum of 48 tokens. Second, in the cross-attention module, the embedding dimensions of the Q, K, and V matrices are chosen from $[64, 128, 256, 512]$. Third, we take the final hidden state of the language model as input to the policy head, which consists of a single fully connected input layer, followed by two output heads (for mean and standard deviation), each implemented as a single fully connected layer (with output size $\approx$ number of paths $\times$ number of nodes). To speed up processing, we crop pre-saved sub-images to $128 \times 128 \sim 512 \times 512$ pixels. For the sampling batch size, we select from $[8, 16, 32, 64]$. The sampling granularity is defined at the source-node level, meaning that allocations for larger topologies require multiple batches. For example, allocating resources for a 150-node topology requires at least three batches. However, we can also increase the partition granularity by aggregating multiple nodes into larger groups,

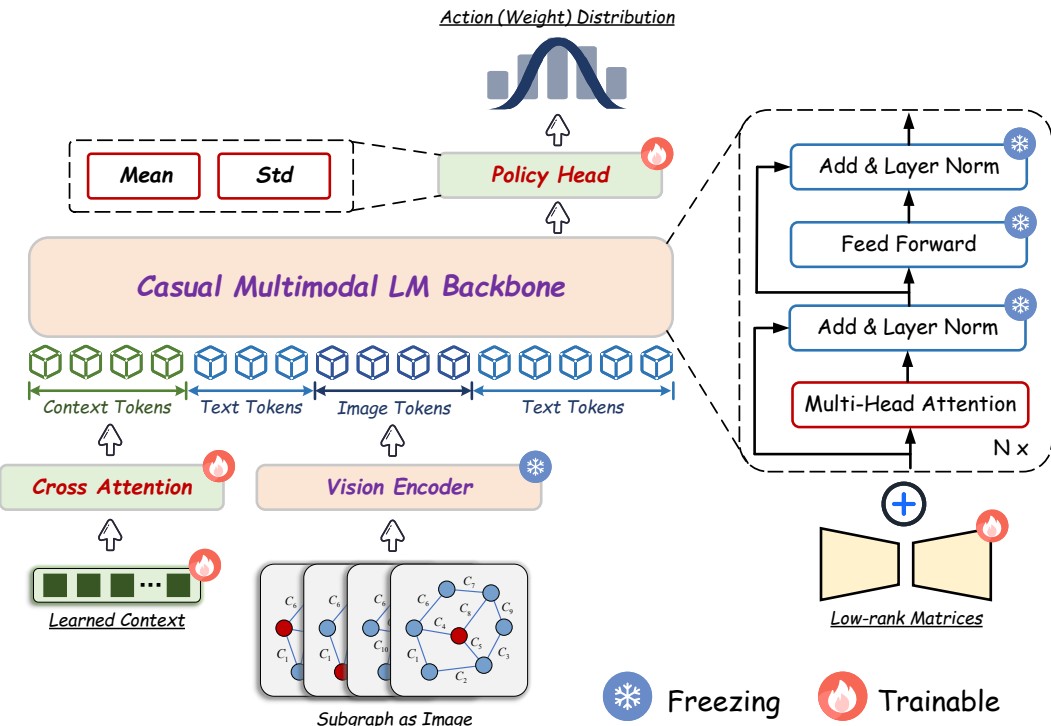

Figure 17: Model architecture of PRAM. In the figure, we use a Qwen2.5-VL-style MLM, which integrates of a vision encoder and a language model decoder to process multimodal inputs.

thereby reducing the number of subgraphs and the overall computational and memory footprint (to enable non-cross-batch operations). Finally, the learning rate for fine-tuning is selected from $[10^{-3}, 5 \times 10^{-4}, 10^{-4}]$, and the adaptation of PRAM is limited to a maximum of 10 epochs with early stopping. Moreover, by default, the Adam optimizer (Kingma, 2014) is employed throughout all experiments.

### C.4.3 POST-SOLUTION REFINEMENT

Although our MLM-powered method already produces high-quality allocation results, there is still room for further improvement under certain test conditions. We refer to this fine-grained adjustment as *post-solution refinement*. When partial demand information or other known conditions are available, post-solution refinement can be carried out in roughly three ways: (i) running gradient descent with ADMM (Xu et al., 2023), (ii) recurrent adjustment with RNNs (AlQiam et al., 2024), and (iii) LP-based partial solution refinement (Liu et al., 2025). Herein we use the idea of (iii) for our refinement. Specifically, we can select $\mathcal{O}(|\mathcal{V}|)$ flows with the highest (Gaussian) variation by the policy head as the manageable demands for solution refinement. This approach still reduces the complexity and scale of the MCF problem, making it tractable for large-scale scenarios while maintaining optimal or near-optimal performance.

# D  THEORETICAL RESULTS

In this section, we provide the detailed theoretical proofs that underpin the main results presented in the paper. Our goal is to establish rigorous guarantees for the proposed method, including its correctness, convergence, and properties. To this end, we first formalize the necessary assumptions and notations, and then proceed to derive the main lemmas and theorems step by step. In particular, our arguments borrow insights from classical results on convex optimization and more recent analyses of LMs. Some key inspirations were excerpted from prior works of Bertsekas & Tsitsiklis (1996); Sutton et al. (1999); Foerster et al. (2018); Perry et al. (2023); Ahn et al. (2023).

## D.1  NOTATIONS & DEFINITIONS

**Notations**  Below, we list our notational conventions used in this section.

- $\|\cdot\|$ is the Euclidean norm.
- $n$ denotes the total number of nodes, i.e., $|\mathcal{V}|$.
- $c_{min}$ denotes the minimum link capacity.
- $c_{max}$ denotes the maximum link capacity.
- $\mathcal{D}_{max}$ is an upper bound on the maximum demand between a source-destination pair.
- $\mathcal{D}_{min}$ is a lower bound on the maximum demand between a source-destination pair. Although it can be 0, to ensure the validity of the results, we replace 0 with a value arbitrarily close to 0.
- $p$ denotes the maximum number of candidate paths interconnecting a source-destination pair.
- $\pi$ denotes an allocation configuration, which can be represented as a vector with $n^2 \times p$ components. It is noteworthy that each source-destination element in $\pi$ is itself a vector whose components sum to $\leq 1$, specifying its weights $r_{(\cdot)}$ across at most $p$ candidate paths.
- $\mathcal{L}_l$ denotes the maximum link utilization objective function, which, in its simplest form, is $\mathcal{L}_l(\pi) = \max_{e \in \mathcal{E}} \frac{f_e(\pi)}{c_e}$.
- $\mathcal{L}_t$ denotes the maximum total flow objective function. For convenience, we reformulate it as follows: Let the allocated flow on each link $x_e(\pi) = \sum_{e \ni p} x_p(\pi)$ where $x_p(\pi)$ is the allocated flow on path $p$. Then the actual flow on $p$ is $f_p(\pi) = x_p(\pi) \cdot \min\left(\min_e \frac{c_e}{x_e(\pi)}, 1\right)$, and the actual flow successfully transmitted between $s$ and $t$ is $f_{s.t}(\pi) = \sum_{p \in P_{s,t}} f_p(\pi)$. Now, we can define $\mathcal{L}_t(\pi) = \sum_{s,t} f_{s,t}(\pi)$ or $\sum_{s,t} \min(\mathcal{D}_{s,t}, f_{s,t}(\pi))$.
- $\mathcal{L}_c$ denotes the maximum concurrent flow objective function. Similar to $\mathcal{L}_t$, we can also reformulate it as follows: $\mathcal{L}_c(\pi) = \min\left(\left\{\frac{f_{s,t}(\pi)}{\mathcal{D}_{s,t}} \mid s, t \in \mathcal{V}, \mathcal{D}_{s,t} > 0\right\}\right)$ or $\min\left(\left\{\frac{f_{s,t}(\pi)}{\mathcal{D}_{s,t}} \mid s, t \in \mathcal{V}, \mathcal{D}_{s,t} > 0\right\} \cup \{1\}\right)$.
- $\bar{R}(\cdot)$ denotes the long-term expected reward per step. Given a policy $\pi$, it is defined as $\bar{R}(\pi) = \mathbb{E}\left[\sum_{t=1}^{\infty} \gamma^{t-1} r_t \mid s_0, \pi\right]$, where $\gamma$, $s_0$ and $r_t$ denote discount rate, start state, and reward at time $t$ respectively.
- $Q^{(\cdot)}(\cdot, \cdot)$ is an action-value function. Given a policy $\pi$, it is defined as $Q^\pi(s, a) = \mathbb{E}\left[\sum_{k=1}^{\infty} \gamma^{k-1} r_{t+k} \mid s_t = s, a_t = a, \pi\right]$, where $a_t$ is the action at time $t$.
- $d^{(\cdot)}(\cdot)$ is the stationary distribution of states. Given a policy $\pi$, it is defined as $d^\pi(s) = \mathbb{E}\left[\sum_{t=1}^{\infty} \gamma^t \Pr\{s_t = s \mid s_0, \pi\}\right]$.

**Definition 1.** *(Convexity) A set $C$ in a vector space is convex if for any two vectors $u, v \in C$ and $\alpha \in [0, 1]$, we have that $\alpha u + (1 - \alpha) v \in C$. A function $f : C \to \mathbb{R}$ is convex if $f(\alpha u + (1 - \alpha) v) \leq \alpha f(u) + (1 - \alpha) f(v)$.*

**Definition 2.** *(Lipschitzness) Let $C \subset \mathbb{R}^d$. A function $f : \mathbb{R}^d \to \mathbb{R}^k$ is $\rho$-Lipschitz over $C$ if for every $w_1, w_2 \in C$ we have that $\|f(w_1) - f(w_2)\| \leq \rho\|w_1 - w_2\|$.*

**Definition 3.** *(Subgradients) Let $S$ be an open convex set, and $f : S \to \mathbb{R}$ be a convex function. A vector $v$ that satisfies $f(w) + \langle u - w, v \rangle \leq f(u)$ for any $u \in S$, is called a subgradient of $f$ at $w$. The set of subgradients of $f$ at $w$ is called the differential set and denoted $\partial f(w)$.*

**Definition 4.** *(Gradient Descent, GD) Gradient descent is an iterative optimization method that, at each step, updates the solution by moving in the direction of the negative gradient of the objective function evaluated at the current point, i.e., $\theta^{(t)} = \theta^{(t-1)} - \Delta \cdot \nabla_\theta f$.*

**Definition 5.** *(Markov Decision Process, MDP) A MDP is defined by the tuple $\mathcal{M} = (\mathcal{S}, \mathcal{A}, P, R, \gamma)$. The state, action, and reward at each time $t$ are denoted $s_t \in \mathcal{S}$, $a_t \in \mathcal{A}$ and $r_t \in \mathbb{R}$ respectively. $P(s', s, a) = \Pr\{s_{t+1} = s' \mid s_t = s, a_t = a\}$ denotes the transition kernel, $R(s, a)$ the expected rewards $\mathbb{E}[r_{t+1} \mid s_t = s, a_t = a]$, and $\gamma \in [0, 1)$ the discount factor. The agent's decision-making procedure at each time is characterized by a policy, $\pi_\theta(a \mid s) = \Pr\{a_t = a \mid s_t = s, \theta\}$, where $\theta$ is a parameter vector. We also write just $\pi(s, a)$ for $\pi_\theta(a \mid s)$.*

**Definition 6.** *(Quasi-Convexity, Hazan et al. (2015)) A function $f : \mathbb{R}^d \to \mathbb{R}$ is quasi-convex if, for all $\mathbf{x}, \mathbf{y} \in \mathbb{R}^d$ with $f(\mathbf{y}) \leq f(\mathbf{x})$, $\langle \nabla f(\mathbf{x}), , \mathbf{y} - \mathbf{x} \rangle \leq 0$; or it satisfies that its level sets, $\mathcal{L}_\alpha(f) = \{x \mid f(x) \leq \alpha\}$, are convex sets for all $\alpha \in \mathbb{R}$.*

### D.2 ASSUMPTIONS & USEFUL LEMMAS

**Assumption 1.** *(Convex or Concave Objective) All three objective functions, i.e., maximum link utilization, maximum total flow, and maximum concurrent flow, are convex/concave with respect to the input candidate path weights.*

**Assumption 2.** *In each attention block, there exist token embeddings of observation–decision pairs $(x_i, y_i)$ from the training set that are aligned with the embeddings of the learned context.*

**Assumption 3.** *In each attention block, the token embeddings of the observation–decision pairs $(x_i, y_i)$ approximately satisfy a linear relation[4] $y_i = W x_i$, where $W \in \mathbb{R}^{N_y \times N_x}$ is a weight matrix.*

Note that the aforementioned Assumption 1 can actually be relaxed slightly; as long as non-smooth quasi-convexity (see Figure 18 on the right) is satisfied, it does not affect the related GD-convergence results in this paper, by using the study of Konnov (2003) and Hazan et al. (2015) (Theorem 5.1). Next, we explicitly prove the core property for all three MCF objective functions in Proposition 1 to support our insight.

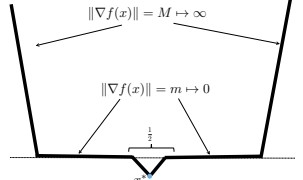

**Lemma 2.** $\frac{ax+(1-x)c}{bx+(1-x)d} \geq \min\left(\frac{a}{b}, \frac{c}{d}\right)$, $x \in [0, 1]$, *holds for any $a, b, c, d > 0$.*

Figure 18: A quasi-convex Locally-Lipschitz function.

*Proof.* Let $f(x) = \frac{ax+(1-x)c}{bx+(1-x)d}$, then we have its derivative:

$$f'(x) = \frac{(a-c)(bx-(1-x)d) - (b-d)(ax+(1-x)c)}{(bx+(1-x)d)^2} = \frac{ad - bc}{(bx+(1-x)d)}$$

Now, since $f(0) = \frac{c}{d}$, $f(1) = \frac{a}{b}$ and $f'(x)$ is monotonic on the interval $[0, 1]$, $f(x) = \frac{ax+(1-x)c}{bx+(1-x)d} \geq \min\left(\frac{a}{b}, \frac{c}{d}\right)$. $\qquad\square$

**Proposition 1.** *The objective function $\mathcal{L}_l$ is convex, while $\mathcal{L}_t$ and $\mathcal{L}_c$ are quasi-concave, in the flow allocation configuration.*

*Proof.* We prove the corresponding properties of the three functions sequentially. First of all, according to Defination 6, we shall prove a function $f$ is quasi-concave if $f(\lambda x + (1 - \lambda)x') \geq \min(f(x), f(x'))$ for any $\lambda \in [0, 1]$.

• **$\mathcal{L}_l$ is convex.** The function $f_e(\pi)$ is, by definition, linear in the splitting ratios $\pi$. Since the $\ell_l$ is expressed as the maximum of functions that are linear in $\pi$, and the maximum of linear (or convex) functions is convex, it follows that $\mathcal{L}_l(\pi) = \max_{e \in \mathcal{E}} \frac{f_e(\pi)}{c_e}$ is convex with respect to $\pi$.

---

[4]Note that we only assume the attention inputs after token embedding are approximately linear, not that the original problem is linear. Indeed, a linear representation model (e.g., followed by an MLP) can express any nonlinear function (Ahn et al., 2023).

• $\mathcal{L}_t$ **is quasi-concave.** We denote by $\phi$ the normalization constant $1/\min\left(\min_e \frac{c_e}{x_e(\pi)}, 1\right) = \max\left(\max_e \frac{x_e(\pi)}{c_e}, 1\right)$, then $\mathcal{L}_t = \sum_{s,t} \sum_{p \in P_{s,t}} \frac{x_p(\pi)}{\phi}$. We also denote $x_p(\pi'') = \lambda x_p(\pi) + (1 - \lambda) x_p(\pi')$, and let $\phi''$ denote its corresponding normalization constant, respectively. Therefore, by using Lemma 2,

$$
\begin{aligned}
\min\left(\mathcal{L}_t(\pi), \mathcal{L}_t(\pi')\right) &= \min\left(\sum_{s,t} \sum_{p \in P_{s,t}} \frac{x_p(\pi)}{\phi}, \sum_{s,t} \sum_{p \in P_{s,t}} \frac{x_p(\pi')}{\phi'}\right) \\
&\leq \frac{\lambda \sum_{s,t} \sum_{p \in P_{s,t}} x_p(\pi) + (1 - \lambda) \sum_{s,t} \sum_{p \in P_{s,t}} x_p(\pi')}{\lambda\phi + (1 - \lambda)\phi'} \\
&= \sum_{s,t} \frac{\sum_{p \in P_{s,t}} (\lambda x_p(\pi) + (1 - \lambda) x_p(\pi'))}{\lambda\phi + (1 - \lambda)\phi'} \\
&\leq \sum_{s,t} \sum_{p \in P_{s,t}} \frac{x_p(\pi'')}{\phi''} = \mathcal{L}_t(\pi''),
\end{aligned}
$$

where the second inequality is because

$$
\begin{aligned}
\phi'' &= \max\left(\max_e \frac{x_e(\pi'')}{c_e}, 1\right) = \max\left(\max_e \frac{\lambda x_e(\pi) + (1 - \lambda)x_e(\pi')}{c_e}, 1\right) \\
&\leq \max\left(\max_e \frac{\lambda x_e(\pi)}{c_e}, \lambda\right) + \max\left(\max_e \frac{(1 - \lambda)x_e(\pi')}{c_e}, 1 - \lambda\right) \\
&= \lambda \max\left(\max_e \frac{x_e(\pi)}{c_e}, 1\right) + (1 - \lambda)\max\left(\max_e \frac{x_e(\pi')}{c_e}, 1\right) = \lambda\phi + (1 - \lambda)\phi'.
\end{aligned}
\tag{4}
$$

• $\mathcal{L}_c$ **is quasi-concave.** To analyze $\mathcal{L}_c$, in the following proof, we reuse some definitions from the proof of $\mathcal{L}_t$. Similarly, we have

$$
\begin{aligned}
\min\left(\mathcal{L}_c(\pi), \mathcal{L}_c(\pi')\right) &= \min\left(\min\left\{\frac{\sum_{p \in P_{s,t}} \frac{x_p(\pi)}{\phi}}{\mathcal{D}_{s,t}}\right\}, \min\left\{\frac{\sum_{p \in P_{s,t}} \frac{x_p(\pi')}{\phi'}}{\mathcal{D}_{s,t}}\right\}\right) \\
&\leq \frac{\lambda \min\left\{\frac{\sum_{p \in P_{s,t}} \frac{x_p(\pi)}{\phi}}{\mathcal{D}_{s,t}}\right\} + (1 - \lambda)\min\left\{\frac{\sum_{p \in P_{s,t}} \frac{x_p(\pi')}{\phi'}}{\mathcal{D}_{s,t}}\right\}}{\lambda\phi + (1 - \lambda)\phi'} \\
&= \min\left\{\frac{\sum_{p \in P_{s,t}} \frac{\lambda x_p(\pi) + (1 - \lambda)x_p(\pi')}{\lambda\phi + (1 - \lambda)\phi'}}{\mathcal{D}_{s,t}}\right\} \\
&\leq \min\left\{\frac{\sum_{p \in P_{s,t}} \frac{x_p(\pi'')}{\phi''}}{\mathcal{D}_{s,t}}\right\} = \mathcal{L}_c\pi''),
\end{aligned}
$$

where the first inequality is by Lemma 2, and the second inequality is by Ineqn (4). $\qquad\square$

**Lemma 3.** *Let $A$ be a nonempty open convex set, and let $f : A \to \mathbb{R}$ be a convex function. Let $\rho > 0$. Then the following are equivalent:* ① *$f$ is $\rho$-Lipschitz continuous on $A$.* ② *At every point $w \in A$, every subgradient $v \in \partial f(w)$ satisfies $\|v\| \leq \rho$.*

*Proof.* First, we assume that $f$ is $\rho$-Lipschitz. Now choose $w \in A$ and $v \in \partial f(w)$. Since $A$ is open, we have $u = w + \frac{v}{\|v\|}\epsilon \in A$ by choosing appropriate $\epsilon > 0$. We can write $\langle u - w, v \rangle = \epsilon\|v\|$ and $\|u - v\| = \epsilon$. Using Definition 3, i.e., $f(u) \geq f(w) + \langle u - w, v \rangle$, we obtain $f(u) - f(w) \geq \epsilon\|v\|$. Recall that $f(u) - f(w) \leq \rho\|u - w\| = \rho\epsilon$, which means $\|v\| \leq \rho$.

Second, we assume that $\|v\| \leq \rho$. From the definition of the subgradient, $f(w) - f(u) \leq \langle v, w - u \rangle$. Now using Cauchy-Schwartz inequality we have $f(w) - f(u) \leq \|v\|\|w - u\| \leq \rho\|w - u\|$. By

repeating similar steps, also we have $f(u) - f(w) \leq \rho \|w - u\|$. Therefore, $\|f(u) - f(w)\| \leq \rho \|w - u\|$ — $f$ is $\rho$-Lipschitz. $\square$

**Lemma 4.** *Consider a convex, $\rho$-Lipschitz function $f$. Let $w^*$ be a vector minimizing $f(w)$, in which $\|w\|$ is bounded by $B$, and GD algorithm with an update rule of the form $w^{(t+1)} = w^{(t)} - \eta v_t$ where $v_t \in \partial f(w)$, $w^{(1)} = 0$ and $\bar{w} = \frac{1}{T} \sum_{t=1}^{T} w^{(t)}$. If we run the algorithm on $f$ for $T \geq \frac{B^2 \rho^2}{\epsilon^2}$ steps with $\eta = \sqrt{\frac{B^2}{\rho^2 T}}$, then the output vector $\bar{w}$ satisfies $f(\bar{w}) - f(w^*) \leq \epsilon, \forall \epsilon > 0$.*

*Proof.* We start by Jensen's inequality,

$$f(\bar{w}) - f(w^*) = f\left(\frac{1}{T} \sum_{t=1}^{T} w^{(t)}\right) - f(w^*) \leq \frac{1}{T} \sum_{t=1}^{T} f\left(w^{(t)}\right) - f(w^*).$$

Thus we have $f(\bar{w}) - f(w^*) \leq \frac{1}{T} \sum_{t=1}^{T} \left(f\left(w^{(t)}\right) - f(w^*)\right)$. using Definition 3, we can write

$$f(\bar{w}) - f(w^*) \leq \frac{1}{T} \sum_{t=1}^{T} \left(f\left(w^{(t)}\right) - f(w^*)\right) \leq \frac{1}{T} \sum_{t=1}^{T} \left\langle w^{(t)} - w^*, v_t \right\rangle.$$

To bound the right-hand side, we focus on $\left\langle w^{(t)} - w^*, v_t \right\rangle$. Given the positive strength $\eta$,

$$\left\langle w^{(t)} - w^*, v_t \right\rangle = \frac{1}{\eta} \left\langle w^{(t)} - w^*, \eta v_t \right\rangle = \frac{1}{2\eta} \left( \left\| w^{(t)} - w^* \right\|^2 - \left\| w^{(t)} - w^* - \eta v_t \right\|^2 + \eta^2 \|v_t\|^2 \right)$$

$$= \frac{1}{2\eta} \left( \left\| w^{(t)} - w^* \right\|^2 - \left\| w^{(t+1)} - w^* \right\|^2 + \eta^2 \|v_t\|^2 \right).$$

Then we have that

$$\sum_{t=1}^{T} \left\langle w^{(t)} - w^*, v_t \right\rangle = \frac{1}{2\eta} \sum_{t=1}^{T} \left( \left\| w^{(t)} - w^* \right\|^2 - \left\| w^{(t+1)} - w^* \right\|^2 + \eta^2 \|v_t\|^2 \right)$$

$$= \frac{1}{2\eta} \underbrace{\left\| w^{(1)} - w^* \right\|^2}_{w^{(1)} = 0} - \frac{1}{2\eta} \underbrace{\left\| w^{(t+1)} - w^* \right\|^2}_{>0} + \frac{\eta}{2} \sum_{t=1}^{T} \|v_t\|^2$$

$$\leq \frac{1}{2\eta} \|w^*\|^2 + \frac{\eta}{2} \sum_{t=1}^{T} \|v_t\|^2$$

Now according to Lemma 3, $\|v_t\| \leq \rho$, and $\|w^*\| \leq B$. Plugging them along with $\eta = \sqrt{\frac{B^2}{\rho^2 T}}$ in the above inequality, we have $\frac{1}{T} \sum_{t=1}^{T} \left\langle w^{(t)} - w^*, v_t \right\rangle \leq \frac{B\rho}{\sqrt{T}}$. Therefore, for all $\epsilon > 0$, after $T \geq \frac{B^2 \rho^2}{\epsilon^2}$ iterations of gradient descent, the objective function converges, i.e., $f(\bar{w}) - f(w^*) \leq \epsilon$, which concludes the proof. $\square$

**Lemma 5.** *(Policy Gradient Theorem, Sutton et al. (1999)) For any MDP $\mathcal{M} = (\mathcal{S}, \mathcal{A}, P, R, \gamma)$,*

$$\frac{\partial \bar{R}}{\partial \theta} = \sum_s d^\pi(s) \sum_a \frac{\partial \pi(s, a)}{\partial \theta} Q^\pi(s, a).$$

*Proof.* We begin with the following observation:

$$Q^\pi(s, a) = \mathbb{E}\left[ r_{t+1} + \gamma \sum_{k=2}^{\infty} \gamma^{k-2} r_{t+k} \,\middle|\, s_t = s, a_t = a, \pi \right]$$

$$= \mathbb{E}(r_{t+1} \mid s_t = s, a_t = a) + \sum_{s'} \gamma \Pr\{s_{t+1} = s' \mid s_t = s, a_t = a\} \mathbb{E}\left[ \sum_{k=2}^{\infty} \gamma^{k-2} r_{t+k} \,\middle|\, s_{t+1} = s', \pi \right]$$

$$= R(s, a) + \sum_{s'} \gamma P(s', s, a) \sum_a \pi(s', a) Q^\pi(s', a).$$

Defining the value function as $V^\pi(s) = \sum_a \pi(s,a)Q^\pi(s,a), \forall s \in \mathcal{S}$, we have that

$$
\begin{aligned}
\frac{\partial V^\pi(s)}{\partial \theta} &= \frac{\partial}{\partial \theta} \sum_a \pi(s,a)Q^\pi(s,a) \\
&= \sum_a \left[ \frac{\partial \pi(s,a)}{\partial \theta} Q^\pi(s,a) + \pi(s,a) \frac{\partial}{\partial \theta} Q^\pi(s,a) \right] \\
&= \sum_a \left[ \frac{\partial \pi(s,a)}{\partial \theta} Q^\pi(s,a) + \pi(s,a) \frac{\partial}{\partial \theta} \left[ R(s,a) + \sum_{s'} \gamma P(s',s,a)V^\pi(s') \right] \right] \\
&= \sum_a \left[ \frac{\partial \pi(s,a)}{\partial \theta} Q^\pi(s,a) + \pi(s,a) \sum_{s'} \gamma P(s',s,a) \frac{\partial V^\pi(s')}{\partial \theta} \right],
\end{aligned}
$$

By expanding the recursion iteratively, i.e., $\frac{\partial V^\pi(s)}{\partial \theta} \to \frac{\partial V^\pi(s')}{\partial \theta} \to \frac{\partial V^\pi(s'')}{\partial \theta} \to \dots$, we obtain

$$
\frac{\partial V^\pi(s)}{\partial \theta} = \sum_x \sum_{k=0}^{\infty} \gamma^k Pr(s \to x, k, \pi) \sum_a \frac{\partial \pi(x,a)}{\partial \theta} Q^\pi(x,a),
$$

where $Pr(s \to x, k, \pi)$ denotes the probability of reaching state $x$ from state $s$ in exactly $k$ steps under policy $\pi$. Also we have

$$
V^\pi(s) = \sum_a \pi(s,a) \mathbb{E} \left[ \sum_{k=1}^{\infty} \gamma^{k-1} r_{t+k} \mid s_t = s, a_t = a, \pi \right] = \mathbb{E} \left[ \sum_{k=1}^{\infty} \gamma^{k-1} r_{t+k} \mid s_t = s, \pi \right].
$$

Finally, the gradient of the expected discounted return $\bar{R}$ follows directly:

$$
\begin{aligned}
\frac{\partial \bar{R}}{\partial \theta} &= \frac{\partial}{\partial \theta} \mathbb{E} \left[ \sum_{t=1}^{\infty} \gamma^{t-1} r_t \mid s_0, \pi \right] = \frac{\partial}{\partial \theta} V^\pi(s_0) \\
&= \sum_s \sum_{k=0}^{\infty} \gamma^k Pr(s_0 \to s, k, \pi) \sum_a \frac{\partial \pi(s,a)}{\partial \theta} Q^\pi(s,a) \\
&= \sum_s d^\pi(s) \sum_a \frac{\partial \pi(s,a)}{\partial \theta} Q^\pi(s,a).
\end{aligned}
$$

Hence, the proof is complete. $\qquad\square$

**Lemma 6.** *Let $f_w : \mathcal{S} \times \mathcal{A} \to \mathbb{R}$ be a parametric approximation to $Q^\pi(s,a)$ with parameter $w$. If $f_w$ is locally optimal and satisfies $\frac{\partial f_w(s,a)}{\partial w} = \frac{1}{\pi(s,a)} \frac{\partial \pi(s,a)}{\partial \theta}$, then $\frac{\partial \bar{R}}{\partial \theta} = \sum_s d^\pi(s) \sum_a \frac{\partial \pi(s,a)}{\partial \theta} f_w(s,a)$.*

*Proof.* We consider learning $f_w$ by sampling trajectories under policy $\pi$ and updating $w$ via a squared-error loss: $\Delta w_t \propto \frac{\partial}{\partial w} [\hat{Q}^\pi(s_t, a_t) - f_w(s_t, a_t)]^2 \propto [\hat{Q}^\pi(s_t, a_t) - f_w(s_t, a_t)] \frac{\partial f_w(s_t, a_t)}{\partial w}$, where $\hat{Q}^\pi(s_t, a_t)$ denotes an unbiased estimator of $Q^\pi(s_t, a_t)$. At convergence to a local optimum, the update direction vanishes in expectation, yielding

$$
\sum_s d^\pi(s) \sum_a \pi(s,a) [Q^\pi(s,a) - f_w(s,a)] \frac{\partial f_w(s,a)}{\partial w} = 0. \tag{5}
$$

Substituting the assumption $\frac{\partial f_w(s,a)}{\partial w} = \frac{1}{\pi(s,a)} \frac{\partial \pi(s,a)}{\partial \theta}$, we obtain $\sum_s d^\pi(s) \sum_a \frac{\partial \pi(s,a)}{\partial \theta} [Q^\pi(s,a) - f_w(s,a)] = 0$. This shows that the approximation error of $f_w$ is orthogonal to the gradient of the policy parameterization. Now recall the policy gradient theorem (Lemma 5) and subtract the orthogonality condition above, we obtain

$$
\begin{aligned}
\frac{\partial \bar{R}}{\partial \theta} &= \sum_s d^\pi(s) \sum_a \frac{\partial \pi(s,a)}{\partial \theta} Q^\pi(s,a) - \sum_s d^\pi(s) \sum_a \frac{\partial \pi(s,a)}{\partial \theta} [Q^\pi(s,a) - f_w(s,a)] \\
&= \sum_s d^\pi(s) \sum_a \frac{\partial \pi(s,a)}{\partial \theta} f_w(s,a).
\end{aligned}
$$

This establishes the claim. $\qquad\square$

**Lemma 7.** *Let $f$ be any differentiable function, and $w_t$ be a sequence generated by a GD-style update rule $w_{t+1} = w_t + \eta_t v_t$, where for all $t$, $v_t$ satisfies ① $c_1 \|\nabla f(w_t)\|^2 \leq -\nabla f(w_t) v_t$; ② $\|v_t\| \leq c_2 \|\nabla f(w_t)\|$, where $c_1$ and $c_2$ are some positive scalars. Assume that for some constant $L > 0$, for all $w, \bar{w} \in \mathbb{R}^n$, we have $\|\nabla f(w) - \nabla f(\bar{w})\| \leq L \|w - \bar{w}\|$, and that $\eta_t \to 0$, $\sum_{t=0}^{\infty} \eta_t = \infty$. Then either $f(w_t) \to -\infty$ or else $f(w_t)$ converges to a finite value and $\lim_{t \to \infty} \nabla f(w_t) = 0$.*

*Proof.* We begin by establishing a standard descent inequality. Fix $w, z \in \mathbb{R}^n$, and define $g(\xi) = f(w + \xi z)$ for $\xi \in [0, 1]$. By the chain rule, $g'(\xi) = \langle z, \nabla f(w + \xi z) \rangle$. Applying the fundamental theorem of calculus,

$$f(w + z) - f(w) = g(1) - g(0) = \int_0^1 g'(\xi)\, d\xi = \int_0^1 \langle z, \nabla f(w + \xi z) \rangle\, d\xi$$

$$= \langle z, \nabla f(w) \rangle + \int_0^1 \langle z, \nabla f(w + \xi z) - \nabla f(w) \rangle\, d\xi$$

$$\leq \langle z, \nabla f(w) \rangle + \int_0^1 \|z\| \cdot \|\nabla f(w + \xi z) - \nabla f(w)\|\, d\xi$$

$$\leq \langle z, \nabla f(w) \rangle + \|z\| \int_0^1 L\xi \|z\|\, d\xi$$

$$= \langle z, \nabla f(w) \rangle + \frac{L}{2} \|z\|^2.$$

Now set $z = \eta_t v_t$. Using also condition ① and ②, we can write

$$f(w_{t+1}) \leq f(w_t) + \langle \eta_t v_t, \nabla f(w_t) \rangle + \frac{L}{2} \|\eta_t v_t\|^2 = f(w_t) + \eta_t \langle v_t, \nabla f(w_t) \rangle + \frac{L}{2} \eta_t^2 \|v_t\|^2$$

$$\leq f(w_t) - \eta_t c_1 \|\nabla f(w_t)\|^2 + \frac{L}{2} \eta_t^2 c_2^2 \|\nabla f(w_t)\|^2$$

$$= f(w_t) - \eta_t \left( c_1 - \frac{Lc_2^2}{2} \eta_t \right) \|\nabla f(w_t)\|^2.$$

Since $\eta_t \to 0$, there exists a positive constant $c$ such that for all $t$ beyond some index $\bar{t}$, $\bar{t}$, $f(w_{t+1}) \leq f(w_t) - \eta_t c \|\nabla f(w_t)\|^2$. This inequality implies that for $t \geq \bar{t}$, the sequence $f(w_t)$ is monotonically non-increasing. Consequently, either $f(w_t) \to -\infty$, in which case the proof is complete, or $f(w_t)$ converges to a finite limit. We now proceed under the latter.

To prove that $\lim_{t \to \infty} \nabla f(w_t) = 0$, suppose the contrary, namely that $\limsup_{t \to \infty} |\nabla f(w_t)| > 0$. Then there exists $\epsilon > 0$ such that $|\nabla f(w_t)| < \epsilon/2$ for infinitely many $t$, while $|\nabla f(w_t)| > \epsilon$ for infinitely many $t$. Hence, we can extract an infinite subset of indices $\mathcal{T}$ such that for each $t \in \mathcal{T}$, there exists $i(t) > t$ with

$$\|\nabla f(w_t)\| < \epsilon/2, \quad \|\nabla f(w_{i(t)})\| > \epsilon, \quad \epsilon/2 \leq \|\nabla f(w_i)\| \leq \epsilon, \quad \text{if } t < i < i(t).$$

Meanwhile, because

$$\|\nabla f(w_{t+1})\| - \|\nabla f(w_t)\| \leq \|\nabla f(w_{t+1}) - \nabla f(w_t)\| \leq L \|\underbrace{w_{t+1} - w_t}_{=\eta_t v_t}\| \leq \eta_t L c_2 \|\nabla f(w_t)\|,$$

it follows that for all $t \in \mathcal{T}$ that are sufficiently large so that $\gamma_t L c_2 < 1$, we have $\frac{\epsilon}{4} \leq \|\nabla f(w_t)\|$; otherwise the condition $\epsilon/2 \leq |\nabla f(w_{t+1})|$ would be contradicted. Without loss of generality, we assume that this relation holds for all $t \in \mathcal{T}$.

We have for all $t \in \mathcal{T}$, using the condition $\|s_t\| \leq c_2 \|\nabla f(w_t)\|$ and the Lipschitz condition:

$$\frac{\epsilon}{2} \leq \|\nabla f(w_{i(t)})\| - \|\nabla f(w_t)\| \leq \|\nabla f(w_{i(t)}) - \nabla f(w_t)\|$$

$$\leq L \|w_{i(t)} - w_t\| \leq L \sum_{i=t}^{i(t)-1} \eta_i \|v_i\|$$

$$\leq L c_2 \sum_{i=t}^{i(t)-1} \eta_i \|\nabla f(w_i)\| \leq L c_2 \epsilon \sum_{i=t}^{i(t)-1} \eta_i,$$

which gives $\sum_{i=t}^{i(t)-1} \eta_i \geq \frac{1}{2Lc_2}$. Now using $\bar{t}$, $f(w_{t+1}) \leq f(w_t) - \eta_t c \|\nabla f(w_t)\|^2$, for sufficiently large $t \in \mathcal{T}$, and the relation $\|\nabla f(w_i)\| \geq \epsilon/4$ for $i = t, t+1, \ldots, i(t) - 1$, we can write

$$f(w_{i(t)}) \leq f(w_t) - c \left(\frac{\epsilon}{4}\right)^2 \sum_{i=t}^{i(t)-1} \eta_i, \quad \forall\, t \in \mathcal{T}.$$

Since $f(r_t)$ converges to a finite value, the preceding relation implies that $\lim_{t \to \infty,\ t \in \mathcal{T}} \sum_{i=t}^{i(t)-1} \eta_i = 0 < \frac{1}{2Lc_2}$. Therefore, the previous assumption is false, and we conclude that $\lim_{t \to \infty} \nabla f(w_t) = 0$. This concludes the proof. $\qquad\square$

### D.3   PROOF OF THEOREM 1

*Proof.* The basic idea of this proof is to show that the configuration vector is upper bounded and that all three objective functions are $\rho$-Lipschitz. Then, since we assume the problem is convex, we can directly obtain the final result via Lemma 4. First, according to the definition of $\pi$, configuration $\|\pi\|$ is clearly bounded by $\sqrt{n^2 \times p} \doteq B$. Below, we discuss the Lipschitz properties of the functions.

**Function $\mathcal{L}_l$'s Lipschitz**  We can observe that $f_e$ is linear in the path weights and so in $\pi$: $f_e = \sum_{s,t \in V} \sum_{p \in P_{s,t}} \sum_{e \ni p} \mathcal{D}_{s,t} \cdot r_p \leq \sum_{s,t \in V} \sum_{p \in P_{s,t}} \sum_{e \ni p} \mathcal{D}_{max} \cdot r_p$. So we have

$$\left\| \mathcal{L}_l\left(\pi^{(i)}\right) - \mathcal{L}_l\left(\pi^{(j)}\right) \right\| = \left\| \max_e \frac{f_e\left(\pi^{(i)} - \pi^{(j)}\right)}{c_e} \right\| \leq \frac{\mathcal{D}_{max}}{c_{min}} \left\| \pi^{(i)} - \pi^{(j)} \right\|.$$

Therefore, $\mathcal{L}_l$ is $\rho$-Lipschitz, with $\rho = \frac{\mathcal{D}_{max}}{c_{min}}$.

**Function $\mathcal{L}_t$'s Lipschitz**  From the definition of the maximum total flow, one should notice that $\mathcal{L}_t$ is the sum of $f_p$. Therefore, it suffices to show that $f_p$ is $\rho$-Lipschitz, in which case $\mathcal{L}_t$ is simply $\sum_p \rho$-Lipschitz. To focus on $\left\| f_p\left(\pi^{(i)}\right) - f_p\left(\pi^{(j)}\right) \right\|$, we distinguish between two cases according to the value of $\min\left(\min_e \frac{c_e}{x_e(\pi)}, 1\right) \doteq \psi$ and make the following observation: $\psi \leq \frac{c_e}{x_e(\pi)} \leq \frac{c_{max}}{x_p(\pi)}$.

• $\underline{\psi\left(\pi^{(j)}\right) = 1.}$ We have that

$$\left\| f_p\left(\pi^{(i)}\right) - f_p\left(\pi^{(j)}\right) \right\| = \left\| x_p\left(\pi^{(i)}\right) \psi\left(\pi^{(i)}\right) - x_p\left(\pi^{(j)}\right) \right\|.$$

Without loss of generality, we assume that $f_p\left(\pi^{(i)}\right) \geq f_p\left(\pi^{(j)}\right)$. Then

$$\left\| f_p\left(\pi^{(i)}\right) - f_p\left(\pi^{(j)}\right) \right\| = x_p\left(\pi^{(i)}\right) \psi\left(\pi^{(i)}\right) - x_p\left(\pi^{(j)}\right) \leq x_p\left(\pi^{(i)}\right) - x_p\left(\pi^{(j)}\right).$$

Also note that $x_p(\pi) = \mathcal{D}_{s,t} \cdot r_p$, where $r_p \in \pi$. Therefore,

$$\left\| f_p\left(\pi^{(i)}\right) - f_p\left(\pi^{(j)}\right) \right\| \leq \mathcal{D}_{max} \left\| \pi^{(i)} - \pi^{(j)} \right\| \leq 2 \cdot \frac{c_{max} \cdot \mathcal{D}_{max}}{c_{min}} \left\| \pi^{(i)} - \pi^{(j)} \right\|.$$

- $\psi\left(\pi^{(j)}\right) < 1.$ Under the same assumption as above, we additionally set $\psi\left(\pi^{(j)}\right) = \frac{c_{e_j}}{x_{e_j}\left(\pi^{(j)}\right)}.$

$$\left\|f_p\left(\pi^{(i)}\right) - f_p\left(\pi^{(j)}\right)\right\|$$

$$= x_p\left(\pi^{(i)}\right) \cdot \psi\left(\pi^{(i)}\right) - x_p\left(\pi^{(j)}\right) \cdot \psi\left(\pi^{(j)}\right)$$

$$= \psi\left(\pi^{(i)}\right) \cdot x_p\left(\pi^{(j)}\right) - \psi\left(\pi^{(j)}\right) \cdot x_p\left(\pi^{(j)}\right) + \psi\left(\pi^{(i)}\right) \cdot x_p\left(\pi^{(i)}\right) - \psi\left(\pi^{(i)}\right) \cdot x_p\left(\pi^{(j)}\right)$$

$$= \psi\left(\pi^{(i)}\right) \cdot \psi\left(\pi^{(j)}\right) \cdot x_p\left(\pi^{(j)}\right) \cdot \left[\frac{1}{\psi\left(\pi^{(j)}\right)} - \frac{1}{\psi\left(\pi^{(i)}\right)}\right] + \psi\left(\pi^{(i)}\right) \cdot \left[x_p\left(\pi^{(i)}\right) - x_p\left(\pi^{(j)}\right)\right]$$

$$\leq \psi\left(\pi^{(i)}\right) \cdot \psi\left(\pi^{(j)}\right) \cdot x_p\left(\pi^{(j)}\right) \cdot \left[\frac{x_{e_j}\left(\pi^{(j)}\right)}{c_{e_j}} - \frac{x_{e_j}\left(\pi^{(i)}\right)}{c_{e_j}}\right] + \psi\left(\pi^{(i)}\right) \cdot \left[x_p\left(\pi^{(i)}\right) - x_p\left(\pi^{(j)}\right)\right]$$

$$\leq \psi\left(\pi^{(i)}\right) \cdot \psi\left(\pi^{(j)}\right) \cdot x_p\left(\pi^{(j)}\right) \cdot \left\|\frac{x_{e_j}\left(\pi^{(j)}\right)}{c_{e_j}} - \frac{x_{e_j}\left(\pi^{(i)}\right)}{c_{e_j}}\right\| + \psi\left(\pi^{(i)}\right) \cdot \left\|x_p\left(\pi^{(i)}\right) - x_p\left(\pi^{(j)}\right)\right\|$$

$$\leq \psi\left(\pi^{(j)}\right) \cdot x_p\left(\pi^{(j)}\right) \cdot \left\|\frac{x_{e_j}\left(\pi^{(j)}\right)}{c_{e_j}} - \frac{x_{e_j}\left(\pi^{(i)}\right)}{c_{e_j}}\right\| + \left\|x_p\left(\pi^{(i)}\right) - x_p\left(\pi^{(j)}\right)\right\|$$

$$\leq \frac{c_{max}}{x_p\left(\pi^{(j)}\right)} \cdot x_p\left(\pi^{(j)}\right) \cdot \left\|\frac{x_{e_j}\left(\pi^{(j)}\right)}{c_{e_j}} - \frac{x_{e_j}\left(\pi^{(i)}\right)}{c_{e_j}}\right\| + \left\|x_p\left(\pi^{(i)}\right) - x_p\left(\pi^{(j)}\right)\right\|$$

$$\leq \frac{c_{max}}{c_{min}} \cdot \left\|x_{e_j}\left(\pi^{(j)}\right) - x_{e_j}\left(\pi^{(i)}\right)\right\| + \left\|x_p\left(\pi^{(i)}\right) - x_p\left(\pi^{(j)}\right)\right\|,$$

where the second inequality is derived using $|a + b| \leq |a| + |b|$. Recall that $x_e(\pi) = \sum_{e \ni p} x_p(\pi) = \sum_{e \ni p} \mathcal{D}_{s,t} \cdot r_p$, and $x_p\left(\pi^{(i)}\right) - x_p\left(\pi^{(j)}\right) \leq \mathcal{D}_{max}\left\|\pi^{(i)} - \pi^{(j)}\right\|$. Therefore,

$$\left\|f_p\left(\pi^{(i)}\right) - f_p\left(\pi^{(j)}\right)\right\| \leq \frac{c_{max}}{c_{min}} \cdot \mathcal{D}_{max}\left\|\pi^{(i)} - \pi^{(j)}\right\| + \mathcal{D}_{max}\left\|\pi^{(i)} - \pi^{(j)}\right\|$$

$$\leq 2 \cdot \frac{c_{max} \cdot \mathcal{D}_{max}}{c_{min}}\left\|\pi^{(i)} - \pi^{(j)}\right\|.$$

By combining two cases, we can conclude that the function $f_p$ is $\rho$-Lipschitz, where $\rho$ is at most $\frac{2 \cdot c_{max} \cdot \mathcal{D}_{max}}{c_{min}}$. As a result, $\mathcal{L}_t = \sum_{s,t} \sum_{p \in P_{s,t}} f_p$ is Lipschitz, and its Lipschitz constant is at most $\sum_{s,t} \sum_{p \in P_{s,t}} \frac{2 \cdot c_{max} \cdot \mathcal{D}_{max}}{c_{min}}$.

**Function $\mathcal{L}_c$'s Lipschitz** The proof for $\mathcal{L}_c$ follows essentially the same idea as for $\mathcal{L}_t$, as both focus on the Lipschitz property of $f_p$. Hence, we can directly reuse the result established in the proof of $\mathcal{L}_t$ for $f_p$. And $\mathcal{L}_c$ is also guaranteed to be $\rho$-Lipschitz, with $\rho$ at most $\sum_{p \in P_{s,t}} \frac{2 \cdot \mathcal{D}_{max} \cdot c_{max}}{\mathcal{D}_{min} \cdot c_{min}}$.

We have now established that all three objective functions $\mathcal{L}$ are $\rho$-Lipschitz and convex, and that $\|\pi\|$ is bounded by $B$. Finally, by Lemma 4, we can write: For every $\epsilon > 0$, if we run the algorithm on $\mathcal{L}$ for finite $T$ (at least $\frac{B^2 \rho^2}{\epsilon^2}$) steps with $\eta = \sqrt{\frac{B^2}{\rho^2 T}}$, then the output vector $\bar{\pi}$ satisfies $f(\bar{\pi}) - f(\pi^*) \leq \epsilon$, which completes the proof. □

### D.4  PROOF OF LEMMA 1

*Proof.* We consider the following expected policy gradient (Foerster et al., 2018) of PRAM:

$$g = \mathbb{E}_\pi\left[\sum_i A_i(s, a)\, \nabla_\theta \log \pi_\theta(a_i \mid s_i)\right], \quad \text{where} \quad A_i(s, a) = R(s, a) - \sum_{a_i'} \pi_\theta(a_i' \mid s_i)\, R\big(s, (a_{-i}, a_i')\big).$$

Therefore,

$$g = -\underbrace{\sum_{s,a} p(s)\pi_\theta(a \mid s) \sum_i \left(\sum_{a_i'} \pi_\theta(a_i' \mid s_i) R(s, (a_{-i}, a_i'))\right) \nabla_\theta \log \pi_\theta(a_i \mid s_i)}_{g_1}$$

$$+ \underbrace{\sum_{s,a} p(s)\pi_\theta(a \mid s) \sum_i R(s,a) \nabla_\theta \log \pi_\theta(a_i \mid s_i)}_{g_2}.$$

**Vanishing of $g_1$.** For each agent $i$, the corresponding component $g_1^{(i)}$ can be written as

$$g_1^{(i)} = \sum_s p(s) \sum_{a_{-i}} \pi_\theta(a_{-i} \mid s_{-i}) \sum_{a_i} \pi_\theta(a_i \mid s_i) \left(\sum_{a_i'} \pi_\theta(a_i' \mid s_i) R(s, (a_{-i}, a_i'))\right) \nabla_\theta \log \pi_\theta(a_i \mid s_i)$$

$$= \sum_{s,a_{-i}} p(s)\pi_\theta(a_{-i} \mid s_{-i}) B_i(s, a_{-i}) \left(\sum_{a_i} \pi_\theta(a_i \mid s_i) \nabla_\theta \log \pi_\theta(a_i \mid s_i)\right),$$

where the inner term $B_i(s, a_{-i}) := \sum_{a_i'} \pi_\theta(a_i' \mid s_i) R(s, (a_{-i}, a_i'))$ is independent of $a_i$. Since

$$\sum_{a_i} \pi_\theta(a_i \mid s_i) \nabla_\theta \log \pi_\theta(a_i \mid s_i) = \sum_{a_i} \nabla_\theta \pi_\theta(a_i \mid s_i) = \nabla_\theta \left(\sum_{a_i} \pi_\theta(a_i \mid s_i)\right) = \nabla_\theta(1) = 0,$$

it follows that $g_1^{(i)} = 0$ for all $i$, and hence $g_1 = \sum_i g_1^{(i)} = 0$.

**Analysis of $g_2$.** As discussed in § 4, under our MCF setting (decoupling of rewards and states), the long-term return satisfies $Q^\pi(s, a) = R(s, a)$. Thus, $g_2^{(i)}$ can be rewritten as $g_2^{(i)} = \mathbb{E}_\pi[Q^\pi(s, a) \nabla_\theta \log \pi_\theta(a_i \mid s_i)]$, which coincides with the standard policy gradient formulation (Williams, 1992). Moreover, by Lemma 5, this admits the equivalent expression $g_2^{(i)} = \sum_s d^\pi(s) \sum_a \frac{\partial \pi(s,a)}{\partial \theta} Q^\pi(s, a)$.

Suppose $f_w$ is a function approximator of $Q^\pi$, parameterized by $w$ and locally optimal. Without loss of generality, assume $\frac{\partial f_w(s,a)}{\partial w} = \frac{1}{\pi(s,a)} \frac{\partial \pi(s,a)}{\partial \theta}$. Under this construction, Lemma 6 implies $g_2^{(i)} = \frac{\partial \bar{R}}{\partial \theta} = \sum_s d^\pi(s) \sum_a \frac{\partial \pi(s,a)}{\partial \theta} f_w(s, a)$.

We may define the update direction $v_k = \sum_s d^{\pi^{(k)}}(s) \sum_a \frac{\partial \pi_{(k)}(s,a)}{\partial \theta} f_{w_{(k)}}(s, a)$, since the $\theta^{(k)}$-update follows the policy gradient direction. The boundedness of $\nabla_\theta^2 R(s, a)$ ensures, via the mean value theorem, that $\nabla_\theta \bar{R}(s, a)$ is Lipschitz. Furthermore, $v_k$ satisfies the required conditions: $\|\nabla_{\theta^{(k)}} \bar{R}(s,a)\|^2 \le -\nabla_{\theta^{(k)}} \bar{R}(s, a) v_k$ and $\|v_k\| \le \|\nabla_{\theta^{(k)}} \bar{R}(s, a)\|$, since $v_k$ is an unbiased estimator of the policy gradient. Together with the step-size requirements and bounded function, these are precisely the conditions needed to invoke Lemma 7, which guarantees convergence to a local optimum, i.e., $\lim_{k\to\infty} g_2 = \lim_{k\to\infty} \sum_i \nabla_{\theta^{(k)}} \bar{R}(s, a) = 0$.

Finally, we have $\lim_{k\to\infty} g = 0 + \lim_{k\to\infty} g_2 = 0$, which completes the proof. $\qquad\square$

### D.5 PROOF OF THEOREM 2

*Proof.* Under Assumptions 2 and 3, we first analyze the dynamics induced by gradient descent. Consider the reference linear model $y(x) = Wx$ and a fine-tuning dataset consisting of input embeddings $x_i$ and corresponding decision embeddings $y_i$. The training objective is the squared-error loss $\mathcal{L}(W) = \frac{1}{2N} \sum_{i=1}^N \|Wx_i - y_i\|^2$. One step of gradient descent with learning rate $\eta$ updates the weights as $\Delta W = -\eta \nabla_W \mathcal{L}(W) = -\frac{\eta}{N} \sum_{i=1}^N (Wx_i - y_i)x_i^\top$. Substituting into the loss gives

$$\mathcal{L}(W + \Delta W) = \frac{1}{2N} \sum_{i=1}^N \|(W + \Delta W)x_i - y_i\|^2 = \frac{1}{2N} \sum_{i=1}^N \|Wx_i - (y_i - \Delta y_i)\|^2.$$

Now consider the linear self-attention update on the token $e_j$,

$$e_j \leftarrow e_j + \text{ATT}(\{e_1, \ldots, e_N\}) = e_j + \sum_h P_h V_h K_h^\top q_{h,j}.$$

The key idea is to construct $W_K, W_Q, W_V$ and the projection $F$ such that a single Transformer update on token $e_j$ replicates the gradient-descent dynamics: $e_j \leftarrow (x_j, y_j) + (0, -\Delta W x_j) = (x_j, y_j - \Delta y_j)$.

Specifically, consider the block-structured weight matrices for one attention head:

$$W_K = W_Q = \begin{pmatrix} I_x & 0 \\ 0 & 0 \end{pmatrix}, \qquad W_V = \begin{pmatrix} 0 & 0 \\ W_0 & -I_y \end{pmatrix},$$

where $I_x$ and $I_y$ are identity matrices of size $N_x$ and $N_y$, and $W_0 \in \mathbb{R}^{N_y \times N_x}$ is the reference model. We also set $P = \frac{\eta}{N} I$, where $I$ is the identity matrix of dimension $N_x + N_y$. With this construction (Ahn et al., 2023),

$$\text{ATT}(\{e_1, \ldots, e_N\}) = \sum_h P_h \sum_i v_{h,i} \otimes k_{h,i} q_{h,j} = \sum_h P_h W_{h,V} \sum_i e_{h,i} \otimes e_{h,i} W_{h,K}^\top W_{h,Q} e_j,$$

and we obtain the following update dynamics:

$$\begin{pmatrix} x_j \\ y_j \end{pmatrix} \leftarrow \begin{pmatrix} x_j \\ y_j \end{pmatrix} + \frac{\eta}{N} I \sum_{i=1}^N \left( \begin{pmatrix} 0 & 0 \\ W_0 & -I_y \end{pmatrix} \begin{pmatrix} x_i \\ y_i \end{pmatrix} \right) \otimes \left( \begin{pmatrix} I_x & 0 \\ 0 & 0 \end{pmatrix} \begin{pmatrix} x_i \\ y_i \end{pmatrix} \right) \begin{pmatrix} I_x & 0 \\ 0 & 0 \end{pmatrix} \begin{pmatrix} x_j \\ y_j \end{pmatrix}$$

$$= \begin{pmatrix} x_j \\ y_j \end{pmatrix} + \frac{\eta}{N} I \sum_{i=1}^N \left( \begin{pmatrix} 0 \\ W_0 x_i - y_i \end{pmatrix} \otimes \begin{pmatrix} x_i \\ 0 \end{pmatrix} \begin{pmatrix} x_j \\ 0 \end{pmatrix} \right)$$

$$= \begin{pmatrix} x_j \\ y_j \end{pmatrix} + \begin{pmatrix} 0 \\ -\Delta W x_j \end{pmatrix}.$$

Thus, each token $e_j = (x_j, y_j)$, including the query token $e_{N+1} = e_{\text{test}} = (x_{\text{test}}, -W_0 x_{\text{test}})$, follows exactly the gradient-descent update. By stacking multiple attention blocks and heads, we can therefore simulate multiple steps of gradient descent, completing the proof. $\square$

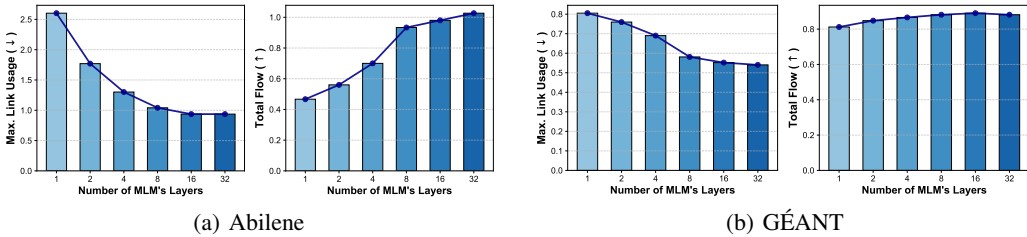

(a) Abilene  (b) GÉANT

Figure 20: (normalized) Performance comparison with different number of layers in the MLM.

# E  ADDITIONAL EMPIRICAL RESULTS

In this section, we present supplementary experiments to further explore the effectiveness of PRAM. We first analysis the impact of multimodal language model's layers and backbones. Then, we investigate the performance of PRAM with different synthetic data distribution.

## E.1  IMPACT OF MLM'S BACKBONE MODEL

During our main experiments, we kept the LM backbone fixed to simplify presentation and ensure fair comparison. In this subsection, however, besides Qwen2.5-VL-7B-Instruct, we further report PRAM's performance with alternative pretrained backbones of different scales, including the larger Llama-3.2-11B-Vision-Instruct (Meta, 2024), and the smaller Qwen2.5-VL-3B-Instruct (Bai et al., 2025). As described in our experimental setup, all MLMs are truncated to 8 layers. Notably, there exists a substantial architectural difference between Llama and Qwen: while Qwen encodes visual features as part of the input "prompt," Llama integrates them across multiple layers via cross-attention. To maintain

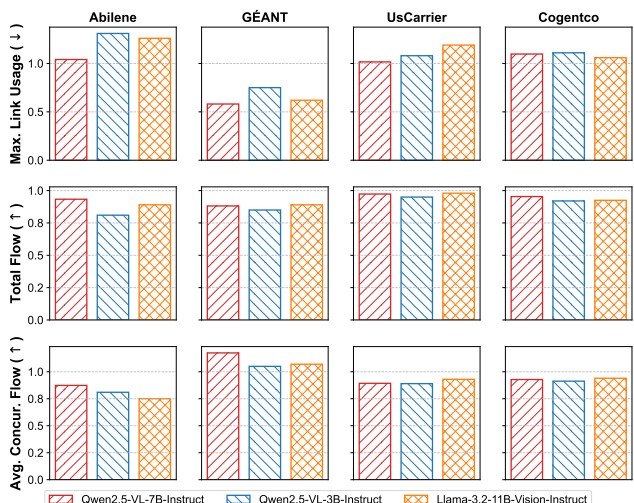

Figure 19: (normalized) Performance comparison with different MLMs' backbone models.

comparability, we truncate only those layers of Llama-3.2-11B-Vision-Instruct's language model that do not involve cross-attention.

The results, presented in Figure 19, show that most MLMs perform well on the target MCF problems. Qwen2.5-VL-7B-Instruct consistently performs no worse than Qwen2.5-VL-3B-Instruct, which aligns with our expectations, as more parameters often lead to stronger understanding and reasoning capabilities. But interestingly, despite having fewer parameters, Qwen2.5-VL-7B-Instruct demonstrates stronger planning ability than the larger Llama-3.2-11B-Vision-Instruct. We attribute this to Qwen's training strategy, which emphasizes numerical reasoning and step-by-step problem solving, whereas Llama, although powerful in language generation, is less specialized in mathematical domains. This suggests that model scale alone does not guarantee superior performance on the MCF problem. Therefore, we choose Qwen2.5-VL-7B-Instruct as the default backbone for PRAM.

## E.2  IMPACT OF NUMBER OF MLM'S LAYERS

In Figure 20, we show how PRAM's link utilization and throughput (total flow) change with the number of Transformer layers across different datasets. The results clearly indicate that performance improves as the number of layers increases, with the effect being particularly pronounced on the Abilene dataset.

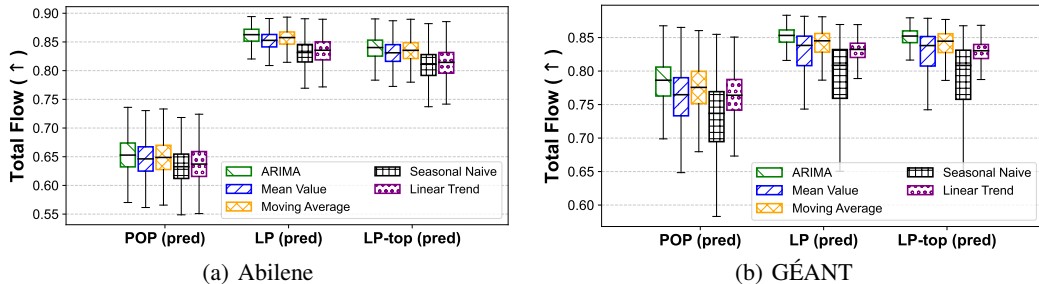

(a) Abilene         (b) GÉANT

Figure 21: (normalized) Performance comparison with different prediction methods.

This finding is consistent with the sharp deterioration observed in our main experiments when the backbone is entirely removed (i.e., # of Transformer layers = 0). We attribute this to the heterogeneous allocation patterns in Abilene, which demand stronger representation capacity (i.e., deeper models) to capture effectively. Nonetheless, the performance gain diminishes beyond 8 layers. To strike a balance between fine-tuning efficiency and performance, we adopt 8 layers as the default setting in the main experiments.

### E.3 BASELINE RESULTS W.R.T. PREDICTION ALGORITHMS

Our evaluation in § 5 focused on demand-prediction-based schemes using a simple moving average. We now compare the moving average against other prediction algorithms on real-world datasets (Abilene and G'EANT). We restrict our comparison to non-parametric methods, since parametric methods such as regression incur higher computational cost and suffer from significant errors due to denormalization of DNN outputs. Specifically, we consider the following prediction methods: ① Mean Value: predicts the average of the entire history; ② Seasonal Naive: repeats the last observed season into the future; ③ Linear Trend: performs linear extrapolation based

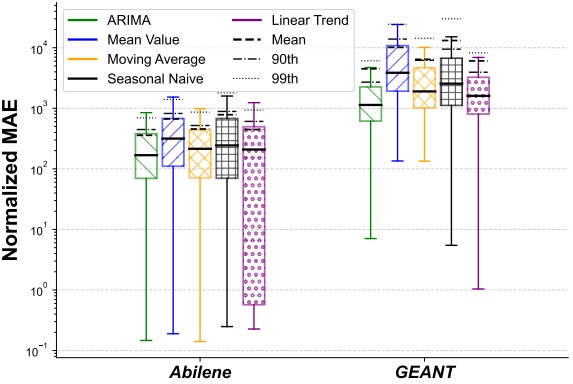

Figure 22: Accuracy of different prediction methods.

on the last two observations; and ④ AutoRegressive Integrated Moving Average (ARIMA, Shumway & Stoffer (2017)): a widely used statistical model for time series forecasting. Each method predicts the next demand for every source-destination pair using only the 12 most recent observed demands of that pair.

Figure 22 presents the accuracy of the different predictors, measured by the normalized mean absolute error (NMAE), i.e., $\frac{1}{K}\sum_{i}^{K}\frac{|x_i - \hat{x}_i|}{x_i}$. As shown, ARIMA and the moving average achieve the lowest errors on average across both datasets, although ARIMA requires more time to collect sufficient information. In contrast, the linear trend exhibits high instability, and the remaining predictors are clearly less accurate. Figure 21 further illustrates the impact of different predictors on multi-commodity flow (MCF) allocation, quantified by total flow relative to an optimal LP solver. The trends mirror the accuracy results: ARIMA performs best, followed by the moving average, while linear trend underperforms compared to mean value despite its seemingly aggressive predictions. Overall, considering all evaluation metrics and computational cost, we adopt the moving average as the default predictor for our baseline methods.

### E.4 RESULTS W.R.T. SYNTHETIC DATA DISTRIBUTIONS

In this subsection, we provide a supplementary study to our evaluation in § 5.2, where we synthesize demand matrices for large-scale topologies using a broader set of distributions, including the Gravity,

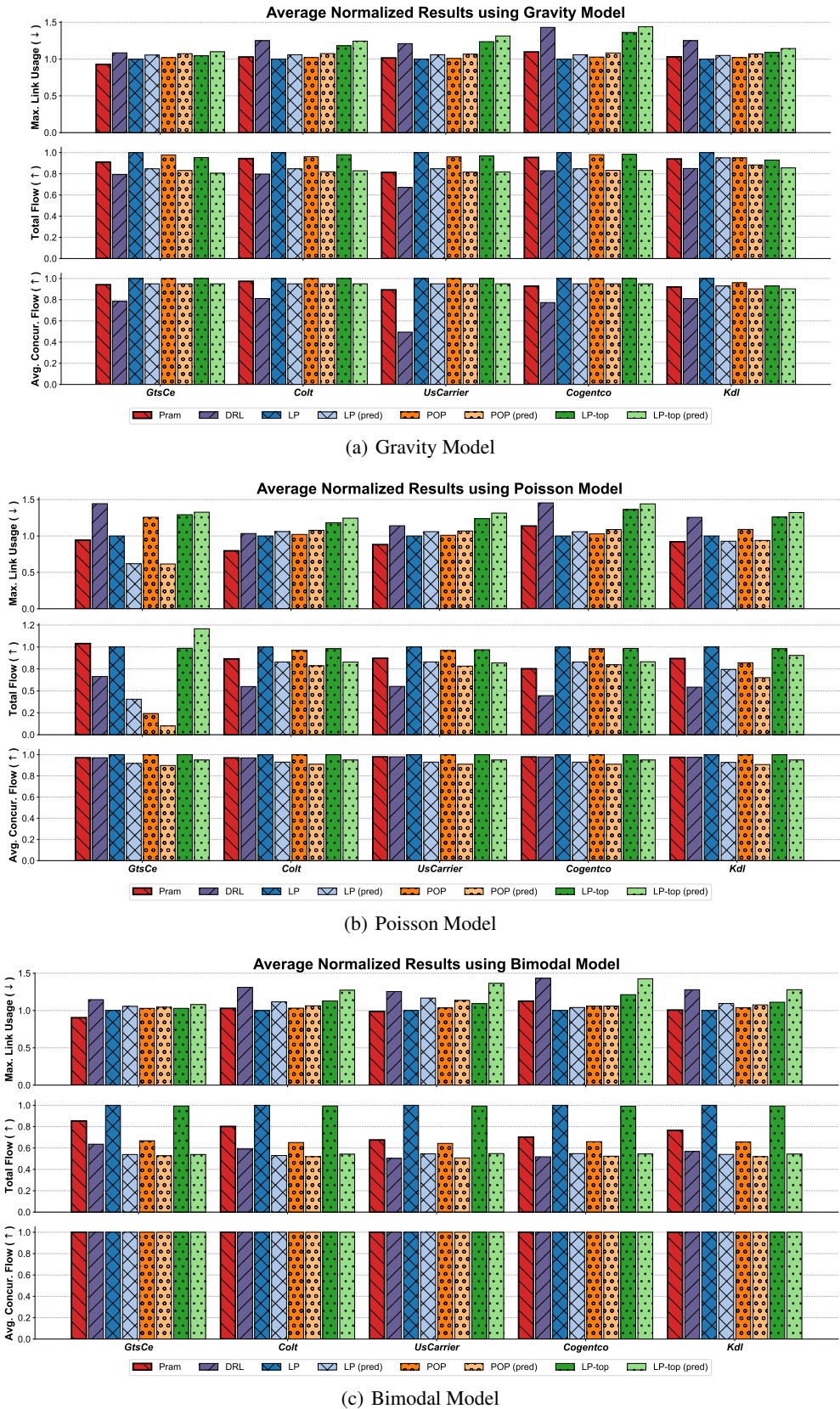

(a) Gravity Model

(b) Poisson Model

(c) Bimodal Model

Figure 23: Performance comparison on large-scale topologies with different data distribution.

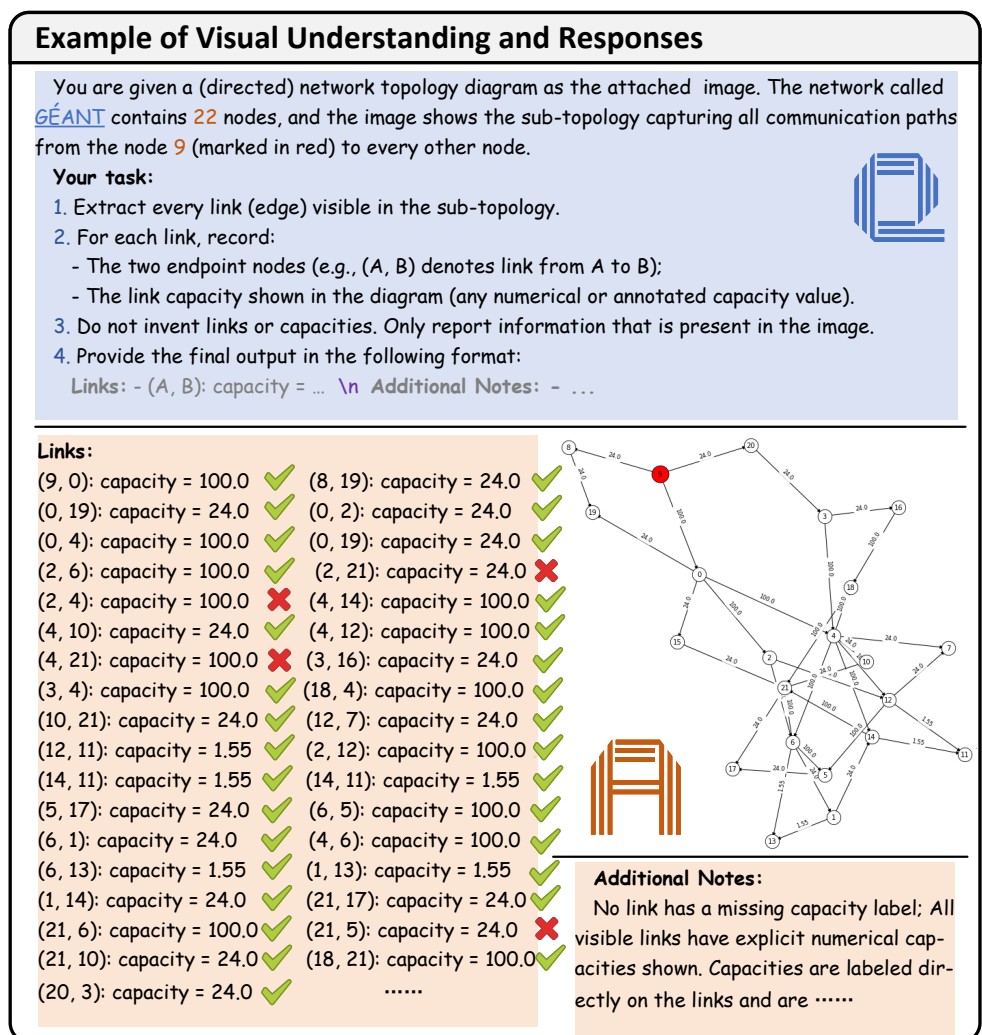

**Example of Visual Understanding and Responses**

You are given a (directed) network topology diagram as the attached image. The network called GÉANT contains 22 nodes, and the image shows the sub-topology capturing all communication paths from the node 9 (marked in red) to every other node.

**Your task:**
1. Extract every link (edge) visible in the sub-topology.
2. For each link, record:
   - The two endpoint nodes (e.g., (A, B) denotes link from A to B);
   - The link capacity shown in the diagram (any numerical or annotated capacity value).
3. Do not invent links or capacities. Only report information that is present in the image.
4. Provide the final output in the following format:
   Links: - (A, B): capacity = … \n Additional Notes: - ...

**Links:**
| | |
|---|---|
| (9, 0): capacity = 100.0 ✓ | (8, 19): capacity = 24.0 ✓ |
| (0, 19): capacity = 24.0 ✓ | (0, 2): capacity = 24.0 ✓ |
| (0, 4): capacity = 100.0 ✓ | (0, 19): capacity = 24.0 ✓ |
| (2, 6): capacity = 100.0 ✓ | (2, 21): capacity = 24.0 ✗ |
| (2, 4): capacity = 100.0 ✗ | (4, 14): capacity = 100.0 ✓ |
| (4, 10): capacity = 24.0 ✓ | (4, 12): capacity = 100.0 ✓ |
| (4, 21): capacity = 100.0 ✗ | (3, 16): capacity = 24.0 ✓ |
| (3, 4): capacity = 100.0 ✓ | (18, 4): capacity = 100.0 ✓ |
| (10, 21): capacity = 24.0 ✓ | (12, 7): capacity = 24.0 ✓ |
| (12, 11): capacity = 1.55 ✓ | (2, 12): capacity = 100.0 ✓ |
| (14, 11): capacity = 1.55 ✓ | (14, 11): capacity = 1.55 ✓ |
| (5, 17): capacity = 24.0 ✓ | (6, 5): capacity = 100.0 ✓ |
| (6, 1): capacity = 24.0 ✓ | (4, 6): capacity = 100.0 ✓ |
| (6, 13): capacity = 1.55 ✓ | (1, 13): capacity = 1.55 ✓ |
| (1, 14): capacity = 24.0 ✓ | (21, 17): capacity = 24.0 ✓ |
| (21, 6): capacity = 100.0 ✓ | (21, 5): capacity = 24.0 ✗ |
| (21, 10): capacity = 24.0 ✓ | (18, 21): capacity = 100.0 ✓ |
| (20, 3): capacity = 24.0 ✓ | ······ |

**Additional Notes:**
No link has a missing capacity label; All visible links have explicit numerical capacities shown. Capacities are labeled directly on the links and are ······

Table 3: Pretrained MLM's visual comprehension example on sub-topology with overlapping elements. The task involves extracting every visible link (edge) within the specified sub-topology and recording the two endpoint nodes $(A, B)$ and the associated numerical link capacity. The (✓) and (✗) indicate the model's success or failure in identifying the corresponding link and its capacity value, demonstrating the visual comprehension level of technical diagrams.

Poisson, and Bimodal models (see Appendix B.2.2 for details). Our results are in Figure 23. Note that the results of the gravity model in Figure 23(a) have already been reported in the main text, we refrain from elaborating on it here as a benchmark. Since the std is very small for different samples (see Figure 9(b) (right)), we omit the drawing of error bars here.

First, consider the Poisson model in Figure 23(b), where we observe pronounced performance fluctuations across methods as the topology changes. Particularly on the GtsCe topology, POP exhibits poor maximum flow preservation. Somewhat counterintuitively, almost all LP-based methods perform better in link utilization prediction than the exact method. We attribute this to the randomness of the Poisson distribution, which selects aggregation points independently at each step, leading to increasingly chaotic commodity flows. Among the learning-based methods, DRL consistently underperforms, while PRAM demonstrates robustness with only negligible fluctuations.

Next, as shown in Figure 23(c), our first observation under the Bimodal model is that all methods achieve near-optimal performance in terms of concurrent flow. This is expected, since the synthetic

traffic is highly sparse: fewer than 10% of the flows are relatively large, while the rest are close to zero, making average parallel flow optimization straightforward. However, prediction errors become more pronounced due to the increased magnitude of the dominant flows. While PRAM is largely resilient to such errors, it performs slightly sub-optimally on the maximum flow objective, ranking just behind LP and LP-top.

To sum up, across diverse demand distributions, PRAM consistently demonstrates robustness and adaptability. Unlike DRL, which struggles with instability, or LP-based methods, whose performance is sensitive to distributional assumptions, PRAM maintains stable results across objectives and topologies. This suggests that PRAM not only generalizes beyond the traffic models used for training but also avoids overfitting to specific flow structures, thereby providing a more reliable solution in practical large-scale settings.

### E.5 EXAMPLES OF MLM'S VISUAL UNDERSTANDING

To ensure the used off-the-shelf MLM's visual understanding capabilities, we conduct some specially designed question-answering small experiments with *Qwen* in the subsection. Our pipeline is very simple and consists mainly of two stages: a) To simulate the iterative process during fine-tuning, meaning the MLM should understand the overall topology information, we first input all subgraphs to the model once; b) We then perform the formal question-answering (QA) experiment, instructing it to provide as much of the observed image information as possible.

As illustrated in Table 3, we start by a challenging test case for evaluating the ability of MLMs to accurately parse and extract structured information in the presence of visual occlusion. The specified GÉANT sub-topology (communication paths from node 9, marked in red) contains several indistinguishable elements resulting from the intersection and overlap of edges and nodes (e.g., $2 \rightarrow 12$ and $2 \rightarrow 6$). For view and report links, MLMs can achieve good performance with an accuracy of nearly 90%. A important finding here is that the majority of erroneous answers are not the occluded ones. This can be attributed to a) the global information previously presented to the multimodal model, which, to some extent, provides the MLM with overlapping structural information from multiple perspectives. Next, Table 4 presents results on two topologies of different sizes: Abilene and UsCarrier. Note that since the MLM refused to answer based on observations from the ultra-large topology, we segmented the corresponding diagram into multiple smaller images for recognition. The accuracy on Abilene is almost zero error, and the results on UsCarrier did not degrade significantly due to node density. Finally, we conduct QA on GÉANT with different image quality. In Table 5, image resolutions of $512 \times 512$ and $128 \times 128$ are used and compared. The performance is reduced by ∼10%; however, we consider it reasonable to appropriately compress the resolution to accelerate model inference.

It is noteworthy that the MLM achieves the above performance without fine-tuning, showcasing its powerful and general vision understanding capabilities nowadays. In real-world scenarios, such as cloud environments or data centers, we anticipate having sufficient time and resources to adapt the MLM (via our lightweight method), allowing it to further correct and refine its information extraction from these network sub-images.

### E.6 RESULTS W.R.T. SUBGRAPH PROCESSING SCHEMES

In this part, we conduct an investigation into various methodologies for representing subgraph as content that can be understood by MLMs. Beyond the Vision-based scheme used by PRAM, we also consider two variants, namely the Text-based scheme and the GNN-based scheme. Following the best practice of Fatemi et al. (2024), Text-based scheme encodes graphs as text in an *Incident* style. For example, we can add "`Here is a graph among nodes 0∼8. In this graph: Node 0 is connected to nodes 1, 2. Node 1 is connected to nodes 0, 2. Node 2 is connected to nodes 0, 1, 3, 4, 5, 7. ... Node 8 is connected to nodes 3, 7.`" into the original prompt for encoding any subgraphs. Regrading GNN-based scheme, although this contradicts our intention to avoid manual design, we separately design a encoder, which is 3-layer Graph Transformer architecture using PyTorch Geometric library, for processing graph-structured inputs, where the node features consist of each node's degree and the sum of the capacities of its adjacent links. Analogous to the aforementioned

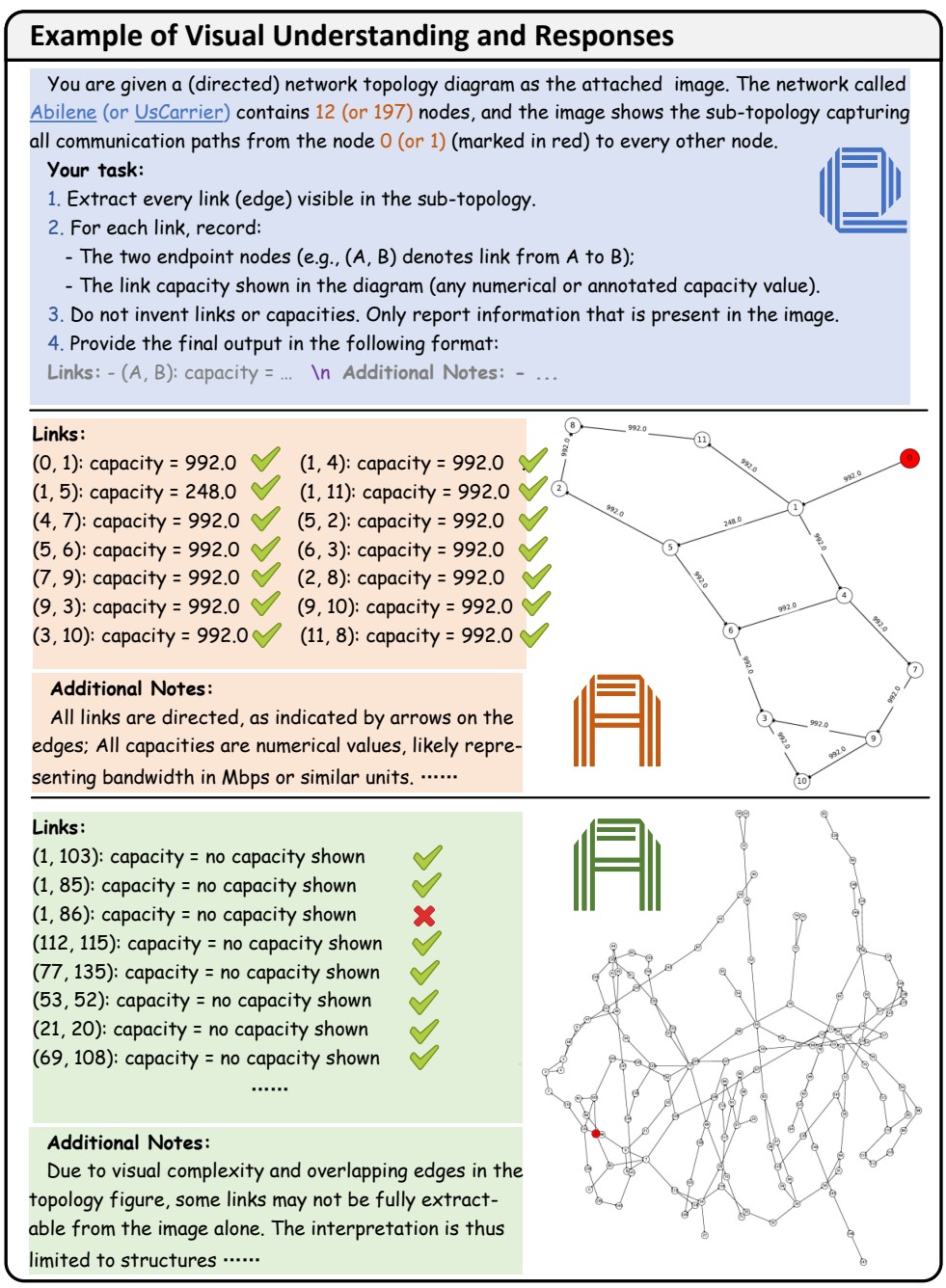

**Example of Visual Understanding and Responses**

You are given a (directed) network topology diagram as the attached image. The network called Abilene (or UsCarrier) contains 12 (or 197) nodes, and the image shows the sub-topology capturing all communication paths from the node 0 (or 1) (marked in red) to every other node.

**Your task:**

1. Extract every link (edge) visible in the sub-topology.
2. For each link, record:
   - The two endpoint nodes (e.g., (A, B) denotes link from A to B);
   - The link capacity shown in the diagram (any numerical or annotated capacity value).
3. Do not invent links or capacities. Only report information that is present in the image.
4. Provide the final output in the following format:

Links: - (A, B): capacity = …   \n   Additional Notes: - ...

**Links:**

(0, 1): capacity = 992.0 ✓    (1, 4): capacity = 992.0 ✓
(1, 5): capacity = 248.0 ✓    (1, 11): capacity = 992.0 ✓
(4, 7): capacity = 992.0 ✓    (5, 2): capacity = 992.0 ✓
(5, 6): capacity = 992.0 ✓    (6, 3): capacity = 992.0 ✓
(7, 9): capacity = 992.0 ✓    (2, 8): capacity = 992.0 ✓
(9, 3): capacity = 992.0 ✓    (9, 10): capacity = 992.0 ✓
(3, 10): capacity = 992.0 ✓   (11, 8): capacity = 992.0 ✓

**Additional Notes:**

All links are directed, as indicated by arrows on the edges; All capacities are numerical values, likely representing bandwidth in Mbps or similar units. ······

**Links:**

(1, 103): capacity = no capacity shown       ✓
(1, 85): capacity = no capacity shown         ✓
(1, 86): capacity = no capacity shown         ✗
(112, 115): capacity = no capacity shown      ✓
(77, 135): capacity = no capacity shown       ✓
(53, 52): capacity = no capacity shown        ✓
(21, 20): capacity = no capacity shown        ✓
(69, 108): capacity = no capacity shown       ✓
······

**Additional Notes:**

Due to visual complexity and overlapping edges in the topology figure, some links may not be fully extract-able from the image alone. The interpretation is thus limited to structures ······

Table 4: Pretrained MLM's visual comprehension example on sub-topology with different scale (density). The task involves extracting every visible link (edge) within the specified sub-topology and recording the two endpoint nodes $(A, B)$ and the associated numerical link capacity. The (✓) and (✗) indicate the model's success or failure in identifying the corresponding link and its capacity value, demonstrating the visual comprehension level of technical diagrams.

**Example of Visual Understanding and Responses**

You are given a (directed) network topology diagram as the attached image. The network called GÉANT contains 22 nodes, and the image shows the sub-topology capturing all communication paths from the node 0 (marked in red) to every other node.

**Your task:**

1. Extract every link (edge) visible in the sub-topology.
2. For each link, record:
    - The two endpoint nodes (e.g., (A, B) denotes link from A to B);
    - The link capacity shown in the diagram (any numerical or annotated capacity value).
3. Do not invent links or capacities. Only report information that is present in the image.
4. Provide the final output in the following format:

**Links:** - (A, B): capacity = ... - (C, D): capacity = ...

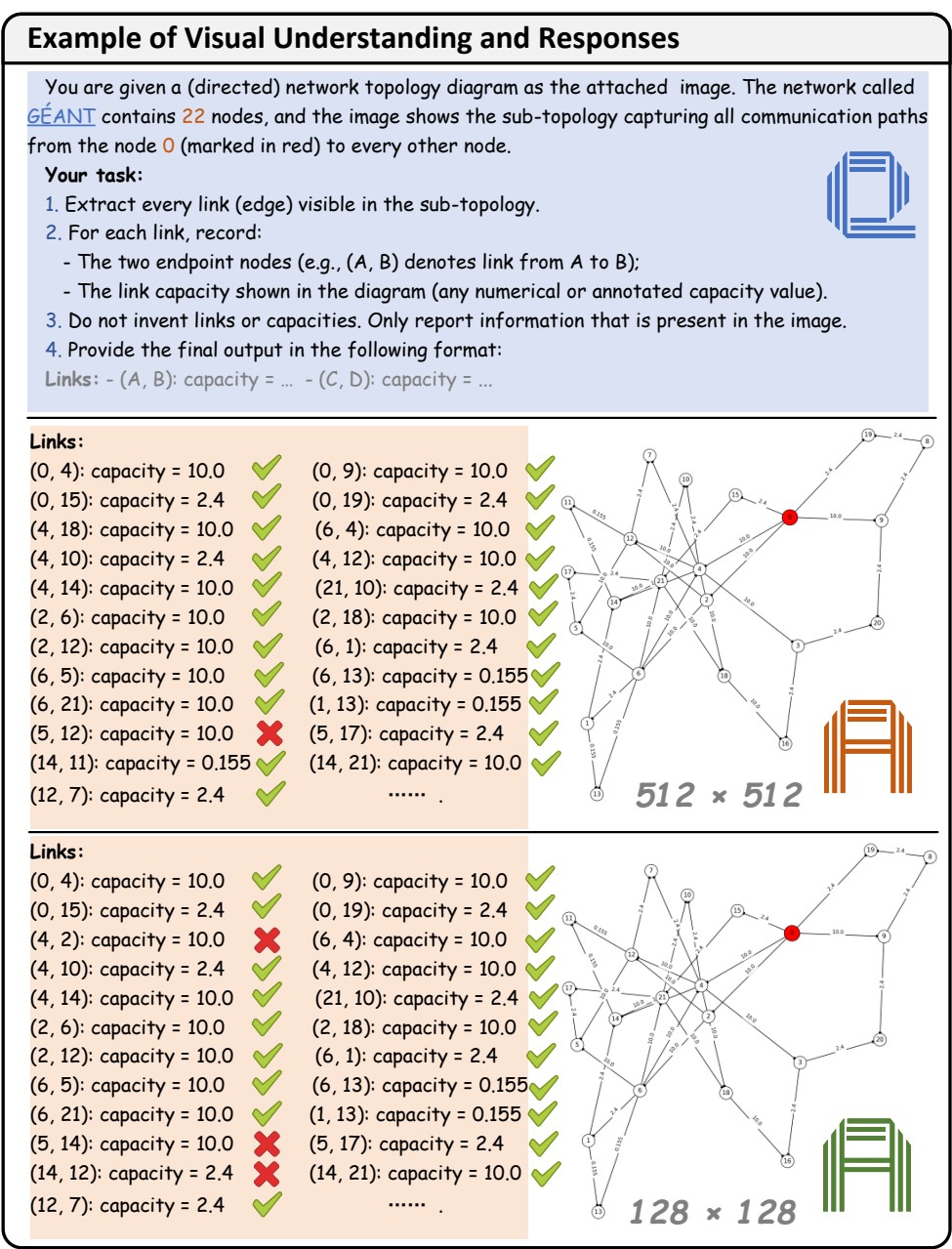

**Links:**

| | |
|---|---|
| (0, 4): capacity = 10.0 ✅ | (0, 9): capacity = 10.0 ✅ |
| (0, 15): capacity = 2.4 ✅ | (0, 19): capacity = 2.4 ✅ |
| (4, 18): capacity = 10.0 ✅ | (6, 4): capacity = 10.0 ✅ |
| (4, 10): capacity = 2.4 ✅ | (4, 12): capacity = 10.0 ✅ |
| (4, 14): capacity = 10.0 ✅ | (21, 10): capacity = 2.4 ✅ |
| (2, 6): capacity = 10.0 ✅ | (2, 18): capacity = 10.0 ✅ |
| (2, 12): capacity = 10.0 ✅ | (6, 1): capacity = 2.4 ✅ |
| (6, 5): capacity = 10.0 ✅ | (6, 13): capacity = 0.155 ✅ |
| (6, 21): capacity = 10.0 ✅ | (1, 13): capacity = 0.155 ✅ |
| (5, 12): capacity = 10.0 ❌ | (5, 17): capacity = 2.4 ✅ |
| (14, 11): capacity = 0.155 ✅ | (14, 21): capacity = 10.0 ✅ |
| (12, 7): capacity = 2.4 ✅ | ...... . |

**512 × 512**

**Links:**

| | |
|---|---|
| (0, 4): capacity = 10.0 ✅ | (0, 9): capacity = 10.0 ✅ |
| (0, 15): capacity = 2.4 ✅ | (0, 19): capacity = 2.4 ✅ |
| (4, 2): capacity = 10.0 ❌ | (6, 4): capacity = 10.0 ✅ |
| (4, 10): capacity = 2.4 ✅ | (4, 12): capacity = 10.0 ✅ |
| (4, 14): capacity = 10.0 ✅ | (21, 10): capacity = 2.4 ✅ |
| (2, 6): capacity = 10.0 ✅ | (2, 18): capacity = 10.0 ✅ |
| (2, 12): capacity = 10.0 ✅ | (6, 1): capacity = 2.4 ✅ |
| (6, 5): capacity = 10.0 ✅ | (6, 13): capacity = 0.155 ✅ |
| (6, 21): capacity = 10.0 ✅ | (1, 13): capacity = 0.155 ✅ |
| (5, 14): capacity = 10.0 ❌ | (5, 17): capacity = 2.4 ✅ |
| (14, 12): capacity = 2.4 ❌ | (14, 21): capacity = 10.0 ✅ |
| (12, 7): capacity = 2.4 ✅ | ...... . |

**128 × 128**

Table 5: Pretrained MLM's visual comprehension example on sub-topology with different resolution. The task involves extracting every visible link (edge) within the specified sub-topology and recording the two endpoint nodes $(A, B)$ and the associated numerical link capacity. The (✓) and (×) indicate the model's success or failure in identifying the corresponding link and its capacity value, demonstrating the visual comprehension level of technical diagrams.

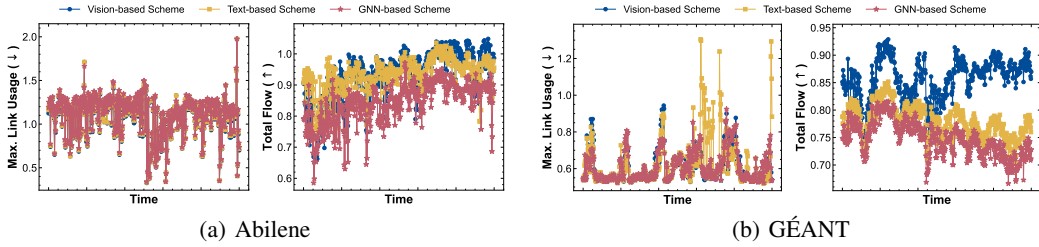

(a) Abilene          (b) GÉANT

Figure 24: (normalized) Performance comparison with different subgraph processing scheme.

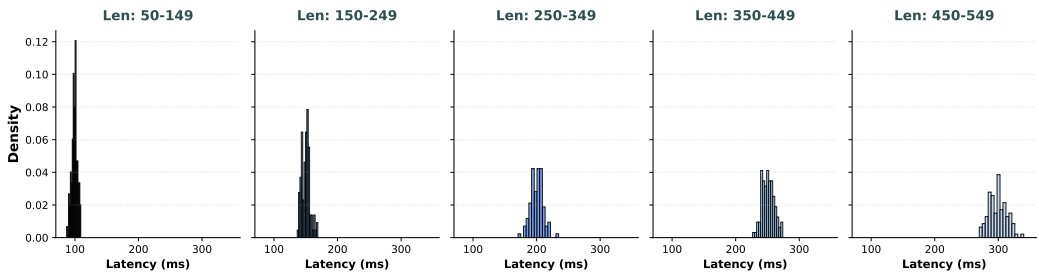

Figure 25: Inference latency distribution vs. input token length of language models.

Qwen architecture, we concatenate the GNN output features with the textual embeddings to form the input for the backbone model.

As depicted in Figure 24, all the adapted MLMs outperform the learning-based baselines presented in the main text. This suggests that the reasoning capabilities of MLMs indeed contribute to solving the multi-commodity flow problem. However, it can be seen that there is a clear ranking among the three schemes for maximizing total flow on GÉANT, where the Vision-based scheme consistently yield improvement over other two schemes. Although the performance of textual encoding exhibits high variance, it ranks second overall, surpassing the average results of GNN encoding, due to better understanding of the modality. It is also noteworthy that as the topology scales, the increasing complexity of the description leads to a rapid growth in prompt length. To investigate the inference overhead of Text-based scheme associated with this length, we specifically conducted simulation experiments. The sampled latency results are presented in Figure 25. We observe a pronounced linear increase in inference latency with respect to the input token length. This implies that, in practice, on larger and more complex topologies, Text-based schemes are likely to suffer a precipitous drop in efficiency owing to their significantly inferior compression ratio compared to visual approaches.

## F   LIMITATIONS & FUTURE WORKS

Several limitations of PRAM warrant consideration. First, its effectiveness relies heavily on the underlying pretrained MLMs, which were originally developed for natural language and image processing rather than MCF problem solving. When these pretrained models face resource constraints, as discussed in § 6, their ability to capture global patterns may be impaired. Second, PRAM currently adopts a non-autoregressive generation paradigm for solving general MCF problems. While recent studies have shown promising results with autoregressive formulations for temporal sequences (Liu et al., 2024), incorporating such techniques into PRAM represents a valuable avenue for future exploration. Third, although frozen MLMs are employed to improve computational efficiency, the training process remains time-consuming, which may limit applicability in time-sensitive settings. In addition, processing identical sub-images can still incur significant memory overhead.

Despite these challenges, PRAM introduces an innovative foundation model architecture specifically tailored for MCF problem solving, delivering gains in both efficiency and solution quality. This work lays the groundwork for future advances in scalability and resource optimization. Moving forward, we plan to explore more sophisticated fine-tuning strategies to further enhance performance while reducing computational demands, drawing on recent progress in parameter-efficient adaptation as well as memory storage and retrieval mechanisms.

## G   USAGE OF LLMS

We acknowledge the use of LLMs as a general-purpose assist tool during the preparation of this paper. The LLM was used to help with text refinement (e.g., improving clarity, grammar, and style in § 3 and § 4) and with programming assistance (e.g., generating plotting scripts and formatting code for § 5). All ideas, research methodology, analyses, and conclusions are entirely those of the authors.

