# OpenReview forum: "Divide, Harmonize, Then Conquer It: Shooting Multi-Commodity Flow Problems with Multimodal Language Models"
_ICLR.cc/2026/Conference — ICLR 2026 Poster_

### Official Review · Reviewer_Zyiq · 2025-10-24

**Soundness:** 2
**Presentation:** 2
**Contribution:** 3
**Rating:** 4
**Confidence:** 4

**Summary:**

This paper proposed a LLM-driven solver for the multi-commondity flow problem. The problem is firstly decomposed as $|\mathcal{V}|$ sub-problems, then the sub-problems are transformed into images, and a MLM is prompted to solve these sub-problems. A multi-agent reinforcement learning algorithm with counterfactual objective is applied to fine-tune the MLM, so that the sub-problems can be solved in a cooperative way that maximize the global objective. Experimental results show that its performance is comparable to traditional baselines.

**Strengths:**

- This paper shows the potential of multimodal language models (MLMs) on the multi-commodity flow problem
- Proper techniques applied to the MLM to improve the efficiency/quality, including LoRA, global context, and the reprogramming technique
- Successful application of multi-agent reinforcement learning on MLMs with a counterfactual objective
- Good ablation studies
- Sufficient details in the appendix

**Weaknesses:**

1. The main claim of the paper, the necessity of MLMs, is not very sufficiently supported by empirical results. For example:
- The paper mentioned that "(MLMs) exhibit emergent abilities that were not explicitly programmed in to them during pre-training on massive data, such as mathematical problem reasoning and generalization to unforeseen conditions" (line 66). I generally agree with this. However, there is no empirical result shown in this paper that MLMs generalize better than other DNNs on unforeseen conditions. In Figure 8, no other ML-based baselines are presented.
- The paper mentioned that "(MLMs) mitigate the costs of retraining and handcrafting specialized DNNs for complex inputs" (line 69). However, even with LoRA, retraining an MLM is still typically considered more costly than traditional DNNs (especially considering the GPU memory requirements). I generally agree that MLMs can be a more "universal" solution than specialized DNNs, which would be easier to adapt between different type of optimization problems (ideally only with a change of the prompt) while other approaches may require a major change. However, this paper is scoped only on the MCF problem with very specific designs (e.g., the decomposion approach). In such a way, this advantage is not very relavant to this paper.

2. The only ML-related baseline approach (Valadarsky et al., 2017) seems to be very old. While some more recent approaches like GDDR (Hope & Yoneki, 2021) and MARL-GNN (Bernárdez et al., 2021) are mentioned in the paper, they are not included in the baselines.

3. Some claims in the paper are not very clear to me, including:
- In the abstract, the paper says "dividing the original problem into local subproblems via an MLM-powered agent" (line 20). But if I understand correctly, the division is not done by an MLM agent but rather "rule-based" ("treating commodities from the same source as one subset", line 157).
- The convexity of the problem is illustrated in a case study and described as an assumption (and all the proof rely on this assumption). Given the closed-form formulation of the problem (appendix B.1), can we make it clear whether it is convex or not?
- The proof is based on the fact that "gradient descent can effectively solve the problem" (line 253). Unfortunately, gradient descent cannot effectively solve the problem. Gradient descent typically works for unconstrained problems or soft-constrained problems (so that the constraints can be transformed to a penalty term in the objective). However, this is a hard-constrained optimization problem, in which all the constraints in equation (1,2,3) should never be violated. In such a way, specific constrained optimization approaches should be applied (typically with KKT condition and lagrange multiplier). Or, empirically, if gradient descent is truely an effective choice, why not solve the problem simply with gradient descent?

4. Using bitmap images to represent graphs can be pretty lossy, especially when the graph is complicated. For example, in the examples provided in the paper, some edge capacities cannot be clearly identified due to heavily crossed links (e.g., row 3, column 1 of Figure 15, the capacity between node 2 and 12 cannot be identified)

**Questions:**

See the questions in "weaknesses".

---

> ### Author Response · Authors · 2025-11-23
> **Official Response to Reviewer Zyiq (Part I)**
>
> Many thanks to you for providing thorough very insightful comments. We have carefully reviewed all your comments and revised our manuscript. Due to space constraints, the new and revised contents have been integrated not only into the relevant sections of the main body, but also extensively into the appendix. We offer detailed responses to your comments below.
>
> ---
>
> > **Comment 1**: The only ML-related baseline approach (Valadarsky et al., 2017) seems to be very old. While some more recent approaches like GDDR (Hope & Yoneki, 2021) and MARL-GNN (Bernárdez et al., 2021) are mentioned in the paper, they are not included in the baselines.
>
> &nbsp;&nbsp;&nbsp;&nbsp;**Response**: We agree that comparing against updated learning-based baselines further highlights the superiority of our proposed method. Unfortunately, we could not reproduce GDDR [1] (ICDCS 2021) or MARL-GNN [2] (ICNP 2021) because their source code is either unreleased or lacks essential parameter files. Instead, we selected two newer and better open-source baselines: HARP [3] (SIGCOMM 2024) and Aether [4] (INFOCOM 2025). Both methods incorporate topological information via GNNs and employ state-of-the-art optimization techniques such as iterative fine-tuning and Graph Transformers. More details about their implementations can be found in `Appendix B.3`. We have updated the revised manuscript to include these two methods in our performance comparison. Specifically, `Figure 6 (Page 7)` illustrates the results on real-world datasets, and `Figure 7 (Page 8)` presents the results on large-scale topologies. The results demonstrate that our method consistently outperforms these SOTA baselines across all tested scenarios (on average ~15\% better than HARP and ~12\% better than Aether). This validates the effectiveness of our design even when compared against the latest advancements in the field.
>
> ---
>
> > **Comment 2**: The main claim of the paper, the necessity of MLMs, is not very sufficiently supported by empirical results. For example:
> The paper mentioned that "(MLMs) exhibit emergent abilities that were not explicitly programmed in to them during pre-training on massive data, such as mathematical problem reasoning and generalization to unforeseen conditions" (line 66). I generally agree with this. However, there is no empirical result shown in this paper that MLMs generalize better than other DNNs on unforeseen conditions. In Figure 8, no other ML-based baselines are presented.
>
> &nbsp;&nbsp;&nbsp;&nbsp;**Response**: Thanks for your scientific rigor. To this end, we refine and provide the additional experiments in our revised paper:
> - **More comprehensive robustness comparison.** In the experimental section, `Figure 8 (Page 9)` now incorporates all three learning-based baseline methods. Figure 8(a) demonstrates that Pram consistently outperforms mainstream machine learning approaches, achieving a lead of over 10\%. Furthermore, the performance degradation exhibited by the learning-based baselines is more pronounced than that observed in the optimization methods. Figure 8(b) reveals that the newly incorporated comparison methods possess better robustness compared to prediction-based approaches. However, their overall stability remains inferior to that of the MLM-powered Pram.
> - **Deeper exploration of MLM's abilities.** We introduced a fine-grained single-link failure experiment to clearly illustrate the generalization benefits of the MLM backbone. As shown in `Figure 10 (Page 10)`, Pram without the MLM backbone suffers far more severe congestion in certain critical failure scenarios than the complete model, despite the application of flow redistribution.
>
> Given the above two sets of experiments, we believe there is now a clear empirical demonstration of how the MLM component confers superior stability and generalization power than other DNNs on unforeseen conditions.
>
> ---
> (continue)

---

> ### Author Response · Authors · 2025-11-23
> **Official Response to Reviewer Zyiq (Part II)**
>
> > **Comment 3**: The paper mentioned that "(MLMs) mitigate the costs of retraining and handcrafting specialized DNNs for complex inputs" (line 69). However, even with LoRA, retraining an MLM is still typically considered more costly than traditional DNNs (especially considering the GPU memory requirements). I generally agree that MLMs can be a more "universal" solution than specialized DNNs, which would be easier to adapt between different type of optimization problems (ideally only with a change of the prompt) while other approaches may require a major change. However, this paper is scoped only on the MCF problem with very specific designs (e.g., the decomposion approach). In such a way, this advantage is not very relavant to this paper.
>
> &nbsp;&nbsp;&nbsp;&nbsp;**Response**: We thank the reviewer for the clarification. Our statement was not intended to argue that fine-tuning an MLM (even with LoRA) is computationally cheaper than training a small DNN. Instead, the point concerns engineering cost, not GPU cost. Prior work [5] has shown that deep learning based architectures incur substantial model engineering overhead. Our claim is also specifically about this aspect: MLMs reduce the model-engineering cost because the same pretrained backbone can be adapted to different MCF variants without designing a new architecture. Even though our paper focuses on MCF, the practical benefit still appears in this setting. Prior DNN-based MCF solvers require bespoke architectures: (manunally) custom graph encoders, task-specific message passing structures, or topology-dependent architectural constraints. Changing the input representation or decomposition often necessitates modifying the architecture and retraining from scratch. In contrast, our proposal keeps the backbone fixed and adapts only via prompting and lightweight fine-tuning. To avoid misunderstanding, we have revised the text in `introduction` Section to emphasize our contributions. We hope this better conveys our intent and addresses the reviewer’s concern.
>
> ---
>
> > **Comment 4**: Some claims in the paper are not very clear to me, including: In the abstract, the paper says "dividing the original problem into local subproblems via an MLM-powered agent" (line 20). But if I understand correctly, the division is not done by an MLM agent but rather "rule-based" ("treating commodities from the same source as one subset", line 157).
>
> &nbsp;&nbsp;&nbsp;&nbsp;**Response**: We appreciate your careful reading and identifying the ambiguity in the abstract. The reviewer is correct: the initial division of the original problem into local subproblems is indeed rule-based. The MLM component is leveraged to resolve these resulting subproblems, not to perform the initial division itself. We have revised the `abstract` in the updated manuscript (line 20) to clarify this distinction. The corrected sentence now reads: "...quickly computes high-quality allocations by dividing the original problem into local subproblems, *which are then resolved by* an MLM-powered 'agent'".
>
> ---
>
> > **Comment 5**: The convexity of the problem is illustrated in a case study and described as an assumption (and all the proof rely on this assumption). Given the closed-form formulation of the problem (appendix B.1), can we make it clear whether it is convex or not?
>
> &nbsp;&nbsp;&nbsp;&nbsp;**Response**: We thank the reviewer for raising this critical question regarding the convexity assumption, which is indeed fundamental to our convergence guarantees. We have addressed this by adding `Proposition 1 (Page 33-34)` to the `Appendix D.2` of the revised manuscript, which strictly proves the properties of all the three objective functions. Our analysis confirms that only the Maximum Link Utilization (MLU) objective is truly convex. The remaining two objective functions are quasi-convex. Crucially, this distinction does not invalidate our convergence proofs relying on gradient descent. Equivalent theoretical guarantees exist for gradient-based convergence involving quasi-convex functions, as detailed in established literature (see Theorem 5.1 in [6]). Therefore, the core conclusions of our theoretical analysis remain sound.
>
> ---
> (continue)

---

> ### Author Response · Authors · 2025-11-23
> **Official Response to Reviewer Zyiq (Part III)**
>
> > **Comment 6**: The proof is based on the fact that "gradient descent can effectively solve the problem" (line 253). Unfortunately, gradient descent cannot effectively solve the problem. Gradient descent typically works for unconstrained problems or soft-constrained problems (so that the constraints can be transformed to a penalty term in the objective). However, this is a hard-constrained optimization problem, in which all the constraints in equation (1,2,3) should never be violated. In such a way, specific constrained optimization approaches should be applied (typically with KKT condition and lagrange multiplier).
>
> &nbsp;&nbsp;&nbsp;&nbsp;**Response**: We thank the reviewer for raising this important point regarding the applicability of GD to hard-constrained optimization problems. While the reviewer is correct that GD is typically unsuitable for solving general hard-constrained problems, the design of our *end-to-end* machine learning framework ensures that the model's output inherently satisfies these constraints. This is detailed across multiple sections of our appendix through both narrative and pseudocode. Specifically, we utilize the following mechanisms to enforce constraint satisfaction:
> - **Minimizing max link usage.** The non-negativity and sum-to-one constraints are maintained by applying a Sigmod or Softmax layer at the network's output (see `Algorithm 8 (Page 29)`), which guarantees the final allocation weights are non-negative and sum to one.
> - **Maximizing (concurrent) flow.** We can also scale down any flow that exceeds link capacity (by max congestion constant $\frac{c_e}{x_e(\pi)}$) to ensure the constraints are never violated. This procedure is explicitly detailed within the theoretical proof `notation (Page 32)` and its practical implementation can be seen in `Algorithm 9 (Page 29)`.
>
> Therefore, from the perspective of the end-to-end machine learning optimization, it makes sense in our paper since the need for traditional KKT conditions or Lagrange multipliers is circumvented, allowing standard GD to effectively optimize the objective function within the constrained output space.
>
> ---
>
> > **Comment 7**: Or, empirically, if gradient descent is truely an effective choice, why not solve the problem simply with gradient descent?
>
> &nbsp;&nbsp;&nbsp;&nbsp;**Response**: Good question! In our case study, the GD algorithm would indeed be a simple and effective solution if we had perfect knowledge of the demand distribution for the upcoming interval. However, as stated in the problem formulation Section, all we have is just *historical* demand information, specifically the demand recorded over the previous 12 intervals in our experiment. Consequently, the objective function for the next interval is unknown at the time of decision-making, rendering a direct calculation of the true gradient impossible. The gradient can only be approximated based on historical patterns. This is precisely why we state in the theoretical analysis Section (lines 279-281) that "a more practical approach is to implicitly model the probability distribution using a DNN trained on extensive empirical data (like Pram)."
>
> ---
>
> > **Comment 8**: Using bitmap images to represent graphs can be pretty lossy, especially when the graph is complicated. For example, in the examples provided in the paper, some edge capacities cannot be clearly identified due to heavily crossed links (e.g., row 3, column 1 of Figure 15, the capacity between node 2 and 12 cannot be identified).
>
> &nbsp;&nbsp;&nbsp;&nbsp;**Response**: We need to emphasize that, although accurately interpreting occluded diagrams is genuinely challenging from a inference perspective (even for humans), our adaptation procedure is performed over all sub-figures. This means that Pram is able to leverage information from other sub-images to ‘fill in’ the missing details of a partially occluded one. To verify this hypothesis, we have added new toy experiments in the `Appendix E.5 (Page 46)`, where we feed all images to a off-the-shelf MLM (Qwen) in two rounds. In the first round, the model sees all images once; in the second round, we query it about specific occluded images. The results (especially in `Table 3`) show that, although the model still makes mistakes (since it is not carefully fine-tuned), it can output correct answers in several clearly ambiguous cases. Therefore, we argue that in our setting, such element overlap does not introduce a significant impact. Furthermore, while the focus of this paper is on general MCF optimization, we believe that modern multimodal language models should now be capable of handling this type of mathematical-diagram recognition task (as explored in [7] and [8]), and that extending vision models toward stronger topological understanding is a promising direction for future work.
>
> ---
> (continue)

---

> ### Author Response · Authors · 2025-11-23
> **Official Response to Reviewer Zyiq (Part IV)**
>
> _**References:**_
> &nbsp;&nbsp;&nbsp;&nbsp;[1] Hope et al. "*GDDR: GNN-based data-driven routing.*" 2021 IEEE 41st International Conference on Distributed Computing Systems (ICDCS). IEEE, 2021.
> &nbsp;&nbsp;&nbsp;&nbsp;[2] Bernárdez et al. "*Is machine learning ready for traffic engineering optimization?*" 2021 IEEE 29th International Conference on Network Protocols (ICNP). IEEE, 2021.
> &nbsp;&nbsp;&nbsp;&nbsp;[3] AlQiam et al. "*Transferable neural wan te for changing topologies.*" Proceedings of the ACM SIGCOMM 2024 Conference. 2024.
> &nbsp;&nbsp;&nbsp;&nbsp;[4] Fan et al. "*Aether: Toward Generalized Traffic Engineering with Elastic Multi-agent Graph Transformers.*" IEEE INFOCOM 2025-IEEE Conference on Computer Communications. IEEE, 2025.
> &nbsp;&nbsp;&nbsp;&nbsp;[5] Wu et al. "*Netllm: Adapting large language models for networking.*" Proceedings of the ACM SIGCOMM 2024 Conference. 2024.
> &nbsp;&nbsp;&nbsp;&nbsp;[6] Hazan et al. "*Beyond convexity: Stochastic quasi-convex optimization.*" Advances in neural information processing systems 28 (2015).
> &nbsp;&nbsp;&nbsp;&nbsp;[7] Zhang et al. "*Mathverse: Does your multi-modal llm truly see the diagrams in visual math problems?.*" European Conference on Computer Vision. 2024.
> &nbsp;&nbsp;&nbsp;&nbsp;[8] Wei et al. "*DeepSeek-OCR: Contexts Optical Compression.*" arXiv preprint arXiv:2510.18234 (2025).

---

> ### Author Response · Authors · 2025-11-27
>
> Dear Reviewer Zyiq,
>
> We hope this message finds you well. We wanted to let you know that the new experiments and proofs have been added to our updated rebuttal. Whenever you have a moment to review it, we would greatly appreciate your feedback.
>
> Thank you for your time and consideration.
>
> Best regards,
> The Authors

---

> > ### Comment · Reviewer_Zyiq · 2025-11-27
> >
> > Since the authors have provided more information regarding my concerns (especially the new baselines), I have raised my score.

---

> > > ### Author Response · Authors · 2025-11-27
> > >
> > > Thank the reviewer very much for the updated assessment. We appreciate that our additional explanations and the inclusion of new baselines addressed several of your concerns. If there are any remaining issues that we can further clarify or strengthen during the rebuttal phase, we would be glad to do so.

---

### Official Review · Reviewer_QZnd · 2025-10-31

**Soundness:** 3
**Presentation:** 4
**Contribution:** 3
**Rating:** 8
**Confidence:** 2

**Summary:**

This paper presents PRAM a reinforcement pipeline architecture to solve
multi-commodity flow (MCF). PRAM works by decomposing MCF problems, compute the
representations of subproblems and recombine the subproblem representations to
return a complete solution. The context-free representation computation is
performed by a vision encoder and tokenizer (enriched with a learned global
context). Then graph and text vectors are passed to a multimodal language
model to combine modalities and compute MCF subproblem representations. The
global recombination is performed by multiagent reinforcement learning using
counterfactual policy gradient.

Communication between subproblems is implemented through LoRA on the multimodal
language model, and through the global context vectors which are the query of a
cross-attention mechanism.

Experimentally, the model shows strong results on real-world datasets and public
topologies, outperforming previous reinforcement learning approaches.
Interestingly, this model exhibits robustness as it is able to solve
problems with link failures and flow bursts with performance degradation under
10%.


The paper is very well written, and the method is novel (to the best of my
knowledge). The reason for using an off-the-shelf LM is clearly motivated, and
the adaptation mechanism is thoroughly described (although quite convoluted) and
justified in ablation studies.

**Strengths:**

1. Well written and clear
2. A novel method based on decomposition
3. Strong empirical results, with intersting robust predictions

**Weaknesses:**

- the neural architecture is quite complex

**Questions:**

1. the use of a MLM is justified, but the model w/o MLM process the input in a
  puzzling way. I can understand how the topology is processed by a GNN, but I
  don't understand how the textual descriptions are handled by a feedforward
  layer (size may change for instance). Can you elaborate? Is there a global
  context vector in this case?

---

> ### Author Response · Authors · 2025-11-23
> **Official Response to Reviewer QZnd**
>
> We sincerely thank the reviewer for the thoughtful feedback. We are encouraged by the comments that our paper is "very well written", while our proposal is "novel" and "clearly motivated". Next, let us address each weakness and question in detail.
>
> ---
>
> > **Comment 1**: The neural architecture is quite complex.
>
> &nbsp;&nbsp;&nbsp;&nbsp;**Response**: We really appreciate the reviewer’s observation. The architecture is designed to remain as simple as possible while still supporting effective multimodal reasoning for MCF. Most components follow standard and widely adopted designs, and the overall structure is modular and lightweight in practice. More importantly, our experiments show that this level of complexity is necessary to achieve strong performance on real topologies. Exploring even simpler alternatives is a valuable direction for future work.
>
> ---
>
> > **Comment 2**: The use of a MLM is justified, but the model w/o MLM process the input in a puzzling way. I can understand how the topology is processed by a GNN, but I don't understand how the textual descriptions are handled by a feedforward layer (size may change for instance). Can you elaborate? Is there a global context vector in this case?
>
> &nbsp;&nbsp;&nbsp;&nbsp;**Response**: Great question! We apologize for the unclear phrasing. In fact, we are not feeding raw text into the model w/o MLM here, as textual input is a modality specific to language models. For the multi-commodity flow problem, the system input consists only of the graph and demand information. Apart from using a GNN to process the topology, we simply input the demand data into the model. Since the historical and statistical features of the demand have fixed dimensions, they can be directly combined into a tensor and processed through fully connected layers. Similarly, the global context vector is resized (via a MLP) and fused with this representation to get final results. We believe this explanation addresses the reviewer’s concern. The corresponding section in the main text has also been updated for clarity. It is worth noting that we also include a GNN+MLM comparison in `Appendix E.6`, and the results further show that the current vision-based version performs better.

---

> > ### Comment · Reviewer_QZnd · 2025-11-28
> >
> > I have read the author's response. Thank you for improving the explanation of the model w/o MLM

---

> > > ### Author Response · Authors · 2025-11-28
> > >
> > > We truly appreciate your positive feedback and continued support.

---

### Official Review · Reviewer_X9xi · 2025-11-01

**Soundness:** 3
**Presentation:** 3
**Contribution:** 3
**Rating:** 4
**Confidence:** 3

**Summary:**

The paper showcases an agentic AI application leveraging multimodal large language models (MLLMs) to solve the multi-commodity flow (MCF) problem through a multi-agent Reinforcement Learning (RL) framework. The method aims to resolve the scalability bottleneck of the classic solver by first dividing the network into subnetworks, then applying MLLM agents to solve each one, and finally ensuring global consistency through the harmonization of the subproblems.

**Strengths:**

1. **Novelty.** To the best of my knowledge, this paper is among the first to leverage the reasoning power of modern Multimodal Language Models (MLMs) for a classic combinatorial optimization problem (MCF). It bridges the gap between large-scale AI models and network optimization.
2. **Significant Performance (Speed and Scalability).** Based on the results, PRAM is significantly faster on large-scale topologies. The ablation study confirms that the partitioning is critical to this scalability.
3. **Near-Optimal Allocation Quality.** Unlike previous ML methods (like DRL) that sacrifice quality for speed , PRAM achieves near-optimal performance, often ranking just behind an LP solver with perfect future knowledge. In some cases (minimizing link usage), it even surpassed the LP solver's results.
4. **Strong Generalization and Robustness.** PRAM demonstrates a key advantage of ML-based approaches: it generalizes well to unforeseen conditions. It shows minimal performance degradation (<10-16%) during simulated link failures and demand fluctuations, scenarios where traditional prediction-based solvers struggle.
5. **Solid Theoretical Foundation.** The paper provides a convincing theoretical justification for why PRAM works. It connects the problem's structure (convexity of MCF objectives, Theorem 1) to the model's learned capability (MLMs can simulate gradient descent, Theorem 2).
6. **Thorough and Convincing Evaluation.** The experimental validation is comprehensive. It leverages multiple real-world and large-scale synthetic datasets , tests three different MCF objectives , and includes a detailed ablation study that validates the importance of every major component (partitioning, the MLM backbone, LoRA, context, and MARL).

**Weaknesses:**

1. **Cost in Training**. From my understanding, the superior scalability of PRAM is achieved by truncation of backbone layers and parameter-efficient adaptation. This can potentially be a problem during deployment.
2. **Memory overhead**. Input to the model are images of the subnetworks, which are sensitive to the partition parameters. I wonder if this can lead to significant memory overhead, especially when different subproblems (nodes) have similar network views that are processed redundantly.
3. **Dependency on Pretrained Backbone**. The success of PRAM is "heavily" dependent on the quality of the underlying pretrained MLM. I wonder how performance varies with different backbones.

**Questions:**

1. **Convergence**. My understanding of Theorem 1, Theorem 2, and Lemma 1 is that they only guarantee convergence for PRAM to locally optimal agents with a limited number of training steps. How can we guarantee that PRAM can strictly learn a global optimal set of agents?
2. **MARL Adaptation**. The MARL algorithm uses counterfactual reasoning, asking how the global objective would change if one agent acted differently. Does the computational cost of this "counterfactual baseline" scale linearly with the number of agents (i.e., the number of nodes )? If so, could this harmonization step become a new bottleneck for extremely large topologies, even if the problem is partitioned?
3. **Comparison to LP Solvers**. The paper notes that PRAM's results for minimizing link usage were "even better than those of LPs". Based on my understanding, since LP solvers are guaranteed to find a near-optimal solution given a specific demand matrix, how is this possible? Does this imply the LP solver was using predicted demands, while PRAM (using historical data) generalized better to the actual future demands used for evaluation?
4. **Role of _Reasoning_**. The paper claims the MLM "_...learns to perform gradient descent in context..._". Is this an emergent, implicit behavior, or is the model explicitly structured to perform GD steps? The proof for Theorem 2 constructs specific attention weights to demonstrate that this is possible, but does the model actually learn this specific structure during fine-tuning?

---

> ### Author Response · Authors · 2025-11-23
> **Official Response to Reviewer X9xi (Part I)**
>
> We appreciate the reviewer’s recognition of our work’s main strengths: bridging a critical research gap, delivering strong empirical results, and building on a solid theoretical foundation. We now address your concerns with additional experiments and the discussion below.
>
> ---
>
> > **Comment 1**: Cost in Training. From my understanding, the superior scalability of PRAM is achieved by truncation of backbone layers and parameter-efficient adaptation. This can potentially be a problem during deployment.
>
> &nbsp;&nbsp;&nbsp;&nbsp;**Response**: Thanks for pointing this out. We would like to clarify that the scalability of Pram comes from our divide-and-optimize strategy rather than from the backbone model itself. We understand that the reviewer may be referring to the lightweight training and testing setup enabled by truncation and parameter-efficient adaptation, but this design does not hinder deployment. These modifications are applied only during the offline training stage. Once training is completed, the resulting model is fully self-contained and does not require any additional adaptation mechanisms. In practice, the deployed model is smaller, faster, and more stable than the full backbone, and it incurs no extra overhead during inference. The training-time reductions therefore do not translate into deployment-time limitations.
>
> ---
>
> > **Comment 2**: Memory overhead. Input to the model are images of the subnetworks, which are sensitive to the partition parameters. I wonder if this can lead to significant memory overhead, especially when different subproblems (nodes) have similar network views that are processed redundantly.
>
> &nbsp;&nbsp;&nbsp;&nbsp;**Response**: We acknowledge this limitation in the manuscript (see `Appendix F`). First, all subproblems share the same source node, and in our problem formulation the candidate paths between sources and destinations are predetermined. As a result, the partitioning procedure is fixed rather than varying across instances. We have added additional details and pseudocode in `Appendix C.2`. Besides, as shown in `Algorithm 7 (Page 27)`, the partitioning of nodes and edges is performed only once at the beginning of the task. All subsequent training and testing steps directly load the pre-generated images, so the cost introduced by the partitioning itself is minimal.
> &nbsp;&nbsp;&nbsp;&nbsp;Regarding redundancy, the concern primarily arises from repeatedly feeding identical images to the frozen vision encoder to obtain embeddings. A more efficient solution would be to store the embeddings directly instead of the images, which would eliminate the need to run the (32-layer) vision encoder for identical inputs. Unfortunately, current mainstream MLM interfaces do not support such caching. We expect future versions with more flexible interfaces to substantially alleviate this overhead.
>
> ---
>
> > **Comment 3**: Dependency on Pretrained Backbone. The success of PRAM is "heavily" dependent on the quality of the underlying pretrained MLM. I wonder how performance varies with different backbones.
>
> &nbsp;&nbsp;&nbsp;&nbsp;**Response**: The concern about dependence on the pretrained backbone is reasonable. However, our initial submission already includes an evaluation of this aspect. Specifically, `Appendix E.1` reports results on four datasets using three open-source multimodal backbones with different sizes and architectures (*Llama-3.2-11B-Vision-Instruct*, *Qwen2.5-VL-3B-Instruct*, and *Qwen2.5-VL-7B-Instruct*). The overall trend is consistent: while stronger backbones generally yield better performance, Pram remains effective across all three variants. This suggests that the method is not tied to any single backbone and can transfer across different pretrained multimodal models.
>
> ---
>
> > **Comment 4**: Convergence. My understanding of Theorem 1, 2, and Lemma 1 is that they only guarantee convergence for PRAM to locally optimal with a limited number of training steps. How can we guarantee that PRAM can strictly learn a global optimal set of agents?
>
> &nbsp;&nbsp;&nbsp;&nbsp;**Response**: Thank you for the question. We want to clarify that although Lemma 1 provide local convergence guarantees, it is among the strong form of theoretical guarantee attainable in general multi-agent reinforcement learning with function approximation. The joint policy space in MARL is inherently nonconvex, making global optimality provably impossible to guarantee even under much stronger assumptions than ours. This limitation is shared by existing MARL approaches such as MADDPG [1] and QMIX [2], none of which provide global-optimality guarantees. Moreover, our goal in this paper is not to solve the global RL optimum, but to enable scalable distributed optimization for MCF. Empirically, our evaluations show Pram consistently finds high-quality solutions that match or surpass state-of-the-art baselines, indicating that local convergence is sufficient for strong real-world performance.
>
> ---
> (continue)

---

> ### Author Response · Authors · 2025-11-23
> **Official Response to Reviewer X9xi (Part II)**
>
> > **Comment 5**: MARL Adaptation. The MARL algorithm uses counterfactual reasoning, asking how the global objective would change if one agent acted differently. Does the computational cost of this "counterfactual baseline" scale linearly with the number of agents (i.e., the number of nodes )? If so, could this harmonization step become a new bottleneck for extremely large topologies, even if the problem is partitioned?
>
> &nbsp;&nbsp;&nbsp;&nbsp;**Response**: It is true that, in a typical representation [3], the number of output nodes of such a $n$-agent policy would equal $|U|^n$, the size of the joint action space, making it impractical to train. But as discussed in the main text, our scheme capitalizes on the fact that the flow allocations in MCF computed for one time interval do not affect future intervals (e.g., demand matrices). This domain-specific insight allows us to improve training by reducing the long-term return to an immediate one-step return. Because the temporal components are independent, samples drawn at any time are equivalent, so repeated fine-tuning iterations are sufficient to realize counterfactual reasoning. This keeps the baseline computation lightweight by drawing a small number ($K \ll |U|$) of random samples each forward (and backward) process. It makes sense for sufficient number of training epoches. More importantly, by the same principle, different agents can also be sampled and assessed at the same time, meaning that the exponent also decreases with the batch size, i.e., $n_a < n$. Therefore, we can say that the computational cost of the counterfactual baseline does not grow linearly with the number of nodes. Now, after partitioning in the node-level (i.e., at most thousands of agents), it does not become a bottleneck even on the largest topologies we tested, which is still cheaper than solving LPs.
>
> ---
>
> > **Comment 6**: Comparison to LP Solvers. The paper notes that PRAM's results for minimizing link usage were "even better than those of LPs". Based on my understanding, since LP solvers are guaranteed to find a near-optimal solution given a specific demand matrix, how is this possible? Does this imply the LP solver was using predicted demands, while PRAM (using historical data) generalized better to the actual future demands used for evaluation?
>
> &nbsp;&nbsp;&nbsp;&nbsp;**Response**: Your comment reflects a very precise understanding of the problem, and we truly appreciate that level of technical accuracy. For this case, we were also surprised by the outcome. Although Gurobi, using either simplex or interior-point methods, should in principle find the global optimum if run sufficiently long, it may also be affected by highly degenerate or ill-conditioned constraints, as well as accumulated floating-point numerical errors in practice [4]. We prepared two LP formulations for MLU (see Python-style code blocks `Algorithm 4 ~ Algorithm 6 (Page 25-26)` added in `Appendix B.3`): LP-w directly optimizes the weights, and the other LP-f explicitly optimizes per-path flows. Unfortunately, as shown in the table below, neither consistently produced strong results. The weight-based formulation performed well on small real-world datasets but broke down on larger topologies, whereas the flow-based version was more stable but consistently suboptimal. We ultimately chose the latter as discussed in `Appendix B.3`.
>
> | **Approach** | Abilene | GÉANT | UsCarrier | Cogentco |
> | :----: | :----: | :----: | :----: | :----: |
> | **LP-w** | 19.67 ± 8.26 | 1.60 ± 0.54 | 67.88 ± 7.25 | 73.39 ± 2.87 |
> | **LP-f** | 22.56 ± 6.47 | 2.87 ± 1.09 | 10.08 ± 0.03 | 17.08 ± 0.06 |
> | **Pram** | 23.87 ± 12.44 | 1.62 ± 0.51 | 9.97 ± 0.03 | 15.79 ± 0.05 |
>
> &nbsp;&nbsp;&nbsp;&nbsp;Regarding the experiment setup, we can confirm that Pram and DRL only uses historical flow (recent flow sizes and statistics). As discussed in the evaluation, we attribute the advantage to the superior gradient-descent compatibility of MLU, the only objective function among the three with a provably convex structure (see `Proposition 1 (Page 33-34)`).
> >*Simple Proof*: flow $f_e$ on eage is, by definition, linear in the flow splitting ratios $\pi$ and so in. Since the $\max$ is a convex function, we have that $\mathcal{L}_l = \max \frac{f_e}{c(e)}$ is convex in $\pi$.
> >
> &nbsp;&nbsp;&nbsp;&nbsp;It aligns with our theoretical analysis. This interpretation also resonates with points raised by next question about the optimization landscape of MLU.
>
> ---
> (continue)

---

> ### Author Response · Authors · 2025-11-23
> **Official Response to Reviewer X9xi (Part III)**
>
> > **Comment 7**: Role of Reasoning. The paper claims the MLM "...learns to perform gradient descent in context...". Is this an emergent, implicit behavior, or is the model explicitly structured to perform GD steps? The proof for Theorem 2 constructs specific attention weights to demonstrate that this is possible, but does the model actually learn this specific structure during fine-tuning?
>
> &nbsp;&nbsp;&nbsp;&nbsp;**Response**: Thanks for the interesting question from the theoretical perspective! The short answer is that this behavior is not explicitly hard-coded. Our theoretical construction relies on the assumption that “for some input–decision pairs in the natural training set, their embeddings align with the learned in-context structure.” In practice, we cannot directly extract these latent label pairs, nor can we identify the exact blockwise linear relations or the precise matrices $W$ that would mirror the construction in Theorem 2. This makes an explicit, theory-faithful visualization extremely difficult. That said, this does not mean the behavior is fully implicit or mysterious. As summarized in our related work section, several prior studies conducted controlled toy-setting experiments on simpler linear problems, and their visualizations provide strong evidence that Transformers *do* learn GD-like update dynamics:
> - Figure 9 in [5] and Figure 2 in [7], showing the similarity between ideal GD updates and learned weights.
> - Figure 2 in [6], visualizing GD-style forward passes of a trained Transformer on two causal-graph tasks.
>
> &nbsp;&nbsp;&nbsp;&nbsp;We expect these results to partially address the reviewer’s question: while we cannot extract a clean one-to-one structure from our full model, the broader evidence strongly suggests that such GD-style dynamics *can* and *do* emerge during fine-tuning. Looking ahead, we plan to incorporate more explicitly interpretable design choices to further clarify and scrutinize these emergent update behaviors in the future.
>
> ---
>
> _**References**_:
> &nbsp;&nbsp;&nbsp;&nbsp;[1] Lowe et al. "*Multi-agent actor-critic for mixed cooperative-competitive environments.*" Advances in neural information processing systems 30 (2017).
> &nbsp;&nbsp;&nbsp;&nbsp;[2] Rashid et al. "*Monotonic value function factorisation for deep multi-agent reinforcement learning.*" Journal of Machine Learning Research 21.178 (2020): 1-51.
> &nbsp;&nbsp;&nbsp;&nbsp;[3] Foerster et al. "*Counterfactual multi-agent policy gradients.*" Proceedings of the AAAI conference on artificial intelligence. Vol. 32. No. 1. 2018.
> &nbsp;&nbsp;&nbsp;&nbsp;[4] Bertsimas et al. "*Online mixed-integer optimization in milliseconds.*" INFORMS Journal on Computing 34.4 (2022): 2229-2248.
> &nbsp;&nbsp;&nbsp;&nbsp;[5] Von Oswald et al. "*Transformers learn in-context by gradient descent.*" International Conference on Machine Learning. PMLR, 2023.
> &nbsp;&nbsp;&nbsp;&nbsp;[6] Nichani et al. "*How Transformers Learn Causal Structure with Gradient Descent.*" International Conference on Machine Learning. PMLR, 2024.
> &nbsp;&nbsp;&nbsp;&nbsp;[7] Ahn et al. "*Transformers learn to implement preconditioned gradient descent for in-context learning.*" Advances in Neural Information Processing Systems 36 (2023): 45614-45650.

---

> ### Author Response · Authors · 2025-11-27
>
> Dear Reviewer X9xi:
>
> Hi! I hope you’re doing well. We're writing to kindly check in regarding our rebuttal responses. Whenever you have a moment to take a look, we would really appreciate it. Thanks again for your time and help.
>
> Best Regards,
> The Authors

---

### Official Review · Reviewer_j7pq · 2025-11-01

**Soundness:** 4
**Presentation:** 4
**Contribution:** 3
**Rating:** 8
**Confidence:** 3

**Summary:**

The paper proposes PRAM, a novel framework that applies multimodal large language models (MLMs) to solve the multi-commodity flow (MCF) problem. Instead of handling large-scale MCFs as a single optimization, PRAM decomposes the process into a structured Divide–Harmonize–Conquer pipeline.

1. Divide:
   The global network is partitioned into node-based subproblems. Each subproblem is encoded as a multimodal input, with network topology represented as images and demand profiles represented as text.

2. Harmonize:
   A collection of MLM-based agents is coordinated via a lightweight multi-agent reinforcement learning framework, enhanced by LoRA (Low-Rank Adaptation) and learned context embeddings for efficient cross-agent communication.

3. Conquer:
   The framework achieves global convergence through the implicit gradient-descent-in-context property of MLMs, allowing distributed subproblem solutions to align toward an optimal global flow.


Empirically, PRAM demonstrates near-optimal allocations across ten real and synthetic network topologies, often outperforming classical LP-based solvers by 1–2 orders of magnitude in runtime while maintaining <8% optimality gap. Theoretically, the paper establishes convergence guarantees under convexity assumptions and provides an interpretability analysis of the learned context embeddings.

**Strengths:**

1. Using multimodal LLMs as distributed solvers for MCF is genuinely new. Prior work on neural optimization largely relied on GNNs or RL; this paper reframes the task as multimodal reasoning over partitioned problems, which feels like a meaningful step forward.

2. The paper’s theoretical discussion—particularly the connection between convexity in MCF and the ability of MLMs to simulate gradient descent is well argued and gives the approach more credibility than most ML-for-optimization papers.

3. The experiments are comprehensive, covering several datasets and objectives. PRAM consistently performs close to LP while running 10–100× faster. The robustness tests (link failures, demand fluctuations) also support the generalization claim.

4. The choice to use LoRA and shared context embeddings keeps the model efficient. The MARL setup is simple but effective for ensuring coordination between agents.

**Weaknesses:**

1. The multimodal input design (topology as image + demand as text) is acknowledged to introduce layout sensitivity and encoding bias.
2. Although faster than LP in inference, fine-tuning large MLMs still requires considerable resources.

**Questions:**

1. How is the visual encoding of topology constructed, and how sensitive are results to the layout or resolution of the generated subgraphs?

2. Would representing the topology as a matrix or sequence (instead of an image) yield comparable results?

---

> ### Author Response · Authors · 2025-11-23
> **Official Response to Reviewer j7pq (Part I)**
>
> Thank you very much for your careful reading and very positive feedback! We are grateful for your encouraging comments including calling our key idea is "genuinely new" and "well argued", as well as our experiments are "comprehensive". Below we respond to your questions and concerns.
>
> ---
>
> > **Comment 1**: The multimodal input design (topology as image + demand as text) is acknowledged to introduce layout sensitivity and encoding bias.
>
> &nbsp;&nbsp;&nbsp;&nbsp;**Response**: We appreciate this insightful comment. First of all, we would like to clarify that our work does not aim to introduce a new foundation model capable of perfectly capturing every detail in images or/and text.  Instead, our aim is to leverage the human-like visual and textual perception abilities of multimodal LLMs to simplify and improve the solution quality of existing approaches. The model only needs to ingest the information relevant to solving the multicommodity flow (MCF) problem (i.e., links and flows). This requirement is not demanding for multimodal LLMs nowadays. We argue that MLMs are pretrained on massive and diverse data, which gives them a basic understanding of structural patterns. This becomes even more pronounced once we further fine-tune them on the observed information in our task. As shown in the examples added to the `Appendix E.5` (see `Table 3 ~ Table 5`), even an off-the-shelf model already performs the key information extraction with high reliability (nearly 90\% accuracy for link identification in subgraphs).
> &nbsp;&nbsp;&nbsp;&nbsp;In fact, recent studies increasingly explore multimodal approaches for mathematical or regression-style problem solving [1], where information is provided to a foundation model in both textual and visual form. For example, G-LLaVA [2] extracts visual features from geometric figures and jointly models them with textual descriptions, enabling the model to understand key geometric elements (e.g., points, lines, angles) and their relationships to the text. For graph-related tasks, Dai et al. [3] showed that LLMs can already recognize graph patterns and even perform simple graph classification on top of them. Moreover, [4] and [5] demonstrate that encoding sequential (historical) information as visual charts significantly boosts downstream performance in anomaly detection tasks, which further supports the rationale behind our design.
>
> ---
>
> > **Comment 2**: How is the visual encoding of topology constructed, and how sensitive are results to the layout or resolution of the generated subgraphs?
>
> &nbsp;&nbsp;&nbsp;&nbsp;**Response**: We would like to gently remind the reviewer that `Appendix C.2` of the original submission briefly described the construction of the visual encoding of the sub-image, however, we apologize for not providing a more detailed explanation. In the revised version, we have added Python-style pseudocode to more clearly illustrate the process. In brief, the procedure iterates through all nodes and edges across the pre-partitioned candidate paths for each source node, and dynamically adjusts the figure size as well as node and edge scales according to the subgraph’s size. The layout is selected to ensure that nodes and edges remain sufficiently separated. This also addresses the reviewer’s engineering concern regarding subgraph clarity: the rendering preserves visual distinguishability to human observers. Of course, we additionally provide results in `Appendix E.5` comparing MLM outputs under different input resolutions (see `Table 5 (Page 48)`). The results show that our current choice of generated resolution is already adequate for reliable recognition and model performance.
>
> ---
>
> (continue)

---

> ### Author Response · Authors · 2025-11-23
> **Official Response to Reviewer j7pq (Part II)**
>
> > **Comment 3**: Would representing the topology as a matrix or sequence (instead of an image) yield comparable results?
>
> &nbsp;&nbsp;&nbsp;&nbsp;**Response**: This is an excellent question. We first note that representing different subgraphs as a fixed-size matrix is not feasible because their structures vary significantly. Following the method in [6], we encoded each subgraph in textual form within the prompt. We conducted experiments on two real-world datasets, and the details are provided in `Appendix E.6`. The results in `Figure 24 (Page 49)` show that the Text-based strategy performs comparably on Abilene in most cases, but experiences a substantial degradation on GÉANT. We suspect this is due to the model’s insufficient global understanding of larger and more complex topologies. Additionally, providing more context increases inference latency. For example, increasing the input size from 1K to 32K tokens can raise the time to first token by a factor of 5 to 10 or even more (see `Figure 25 (Page 50)`). Therefore, we opt for the visual encoding approach, which offers a higher compression ratio and more stable performance across different topologies. For a more comprehensive comparison, we additionally include a GNN-based processing variant, where the vision encoder is replaced with a GNN encoder. The results show that the pretrained native module in MLMs has an advantage in topological understanding, since the GNN struggles to properly align the embedding features.
>
> ---
>
> > **Comment 4**: Although faster than LP in inference, fine-tuning large MLMs still requires considerable resources.
>
> &nbsp;&nbsp;&nbsp;&nbsp;**Response**: We acknowledge the reviewer’s concern. We have already reduced the fine-tuning cost as much as possible by using compact visual encodings and a lightweight parameter-efficient setup. Fine-tuning is a one-time offline step, and our main goal, namely accelerating inference-time MCF solving, has been fully achieved. Developing even more lightweight designs is a natural direction for future work.
>
> ---
>
> _**References**_:
> &nbsp;&nbsp;&nbsp;&nbsp;[1] Yan et al. "*A Survey of Mathematical Reasoning in the Era of Multimodal Large Language Model: Benchmark, Method & Challenges.*" Findings of the Association for Computational Linguistics: ACL 2025.
> &nbsp;&nbsp;&nbsp;&nbsp;[2] Gao et al. "*G-LLaVA: Solving Geometric Problem with Multi-Modal Large Language Model.*" The Thirteenth International Conference on Learning Representations. 2025.
> &nbsp;&nbsp;&nbsp;&nbsp;[3] Dai et al. "*How Do Large Language Models Understand Graph Patterns? A Benchmark for Graph Pattern Comprehension.*" The Thirteenth International Conference on Learning Representations. 2025.
> &nbsp;&nbsp;&nbsp;&nbsp;[4] Zhou et al. "*Can LLMs Understand Time Series Anomalies?*" The Thirteenth International Conference on Learning Representations. 2025.
> &nbsp;&nbsp;&nbsp;&nbsp;[5] Park et al. "*Delving into Large Language Models for Effective Time-Series Anomaly Detection.*" The Thirty-ninth Annual Conference on Neural Information Processing Systems. 2025.
> &nbsp;&nbsp;&nbsp;&nbsp;[6] Fatemi et al. "*Talk like a Graph: Encoding Graphs for Large Language Models.*" The Twelfth International Conference on Learning Representations. 2024.

---

### Author Response · Authors · 2025-11-23
**General Responses and Summary of Revisions**

Dear Reviewers/AC/SAC/PC,

We are grateful to the reviewers (**R1**=**j7pq**, **R2**=**X9xi**, **R3**=**QZnd**, **R4**=**Zyiq**) for their detailed and constructive feedback on our submission. In particular, we are pleased that all reviewers unanimously recognized the contribution of our work. First, all reviewers agreed that our proposed framework, namely Pram, is novel and fills a notable gap in the existing literature. Additionally, most of them found our work to be interesting and appreciated the practical importance of the problem addressed by the paper. We are also glad the reviewers liked our presentation and extensive theoretical \& empirical evaluation.

We summarize major changes in the revision during the rebuttal process.

- **Newer Baselines** (**R4**). In the original submission, we only compared against a reinforcement learning algorithm, which may not sufficiently represent the SOTA performance at the intersection of this area and machine learning. As of now, two most recent ML-based baselines from 2024 and 2025 have been added to `Section 5.1` and `Section 5.2` to strengthen the overall persuasiveness.
- **Deeper Exploration** (**R1**, **R2**, **R3**, **R4**). We conduct a series of experiments supporting our proposal, including: in `Section 5.3` and `Section 5.4`, more comprehensive robustness tests to highlight the generalization advantages of MLMs over standard DNNs; in the QA experiments of `Appendix E.5`, multiple examples demonstrating the visual understanding capabilities of multimodal models; and in `Appendix E.6`, comparisons of three different topology-processing schemes to show the efficiency of vision-based encoding.
- **Rigorous Analysis** (**R4**): Our convexity assumptions are well-motivated from case studies, but there is a lack of a rigorous proof previously. To solidify our theoretical analysis, we add Proposition 1 in `Appendix D.2`, which strictly proves the (quasi-)convexity for all objective functions.
- **Implementation Details** (**R2**, **R3**, **R4**). We provide additional algorithmic details: `Appendix B.3` lists all Gurobi optimization pseudocode used in our experiment, and `Appendix C.1` presents the pseudocode for sub-image rendering.
- **Additional Clarifications** (**R2**, **R4**). For parts that were unclear or potentially confusing, we rewrote or supplemented the text, with revisions affecting `Abstract`, `Section 1`, `Section 5` and `Section 6`.

All updates are now highlighted in *blue*. Compared with the original submissions, the revised manuscript has additional **10** pages (40 $\rightarrow$ 50).

The valuable suggestions from reviewers are very helpful for us to revise the paper to a better shape. We'd be very happy to answer any further questions. Looking forward to the reviewer's feedback.

Best Regards,
The Authors

---

### Meta-Review · Area_Chair_7uw4 · 2026-01-13

**Summary:**

This paper proposes a vision-based LLM for addressing multi-commodity flow (MCF) problem. The original problem is first decomposed into subproblems. These subproblems are then transformed into image representations and solved by prompting a multimodal language model (MLM). To enable cooperative optimization, the MLM is fine-tuned using a multi-agent reinforcement learning algorithm with a counterfactual objective, allowing the subproblems to be solved collaboratively in a way that maximizes the global objective.

In the rebuttal, authors explained that 1) current LLMs have strong performance in recognizing graph patterns, so the design in this paper is free from encoding bias and layout sensitivity; 2) LLMs are not sensitive to different input resolutions; 3) image encoding is better than textual and graph encodings; 4) the global optimality is hard to achieve; 5) computational cost of the counterfactual baseline is low. Some new results were added by implementing recent baselines and authors also clarified the division rule, the robustness and the GD soundness for the specific studied problem.

Some concerns from reviewers were not totally addressed: 1) fine-tuning LLMs are still expensive. While authors argue that the fine-tuning is generally applicable to different problems, this kind of advantage was not present in current paper; 2) redundant processing of the same subnetworks, limited by LLM itself; 3) the interpretability is insufficient; 4) the GD seems limited to a very specific problem without much generalization potential.

**Reviewer Concerns:**

Most questions and concerns have been addressed in the rebuttal. The remaining concern is that this work applies a very general, powerful LLM to solve a very specific problem without showing the potential of it being applied to similar variants. The interpretability is critical by using language and images but missing in this work.

**Reviewer Scores:**

Two reviewers assigned scores of 8 and maintained these scores during the rebuttal phase. However, one of them expressed low confidence and did not provide substantive feedback. The remaining two reviewers initially gave scores of 4, with one reviewer acknowledging the newly added baselines and indicating a willingness to raise the score.

---

### Decision · Program_Chairs · 2026-01-26

Accept (Poster)